# HETEROFOR 1.0: a spatially explicit model for exploring the response of structurally complex forests to uncertain future conditions. II. Phenology and water cycle.

Louis de Wergifosse[1], Frédéric André[1], Nicolas Beudez[2], François de Coligny[2], Hugues Goosse[1], François Jonard[1], Quentin Ponette[1], Hugues Titeux[1], Caroline Vincke[1], Mathieu Jonard[1]

[1]Earth and Life Institute, Université catholique de Louvain, Louvain-la-Neuve, 1348, Belgium
[2]AMAP, Univ Montpellier, CIRAD, CNRS, INRAE, IRD, 34000 Montpellier, France
*Correspondence to*: Louis de Wergifosse (louis.dewergifosse@uclouvain.be)

**Abstract**

Climate change affects forest growth in numerous and sometimes opposite ways and the resulting trend is often difficult to predict for a given site. Integrating and structuring the knowledge gained from the monitoring and experimental studies into process-based models is an interesting approach to predict the response of forest ecosystems to climate change. While the first generation of models operates at stand level, one needs now spatially explicit individual-based approaches to account for individual variability, local environment modification and tree adaptive behaviour in mixed and uneven-aged forests supposed to be more resilient under stressful conditions. The local environment of a tree is strongly influenced by the neighbouring trees which modify the resource level through positive and negative interactions with the target tree. Among others, drought stress and vegetation period length vary with tree size and crown position within the canopy.

In this paper, we describe the phenology and water balance modules integrated in the tree growth model HETEROFOR (HETEROgenous FORest) and evaluate them on six heterogeneous sessile oak and European beech stands with different levels of mixing and development stages and installed on various soil types. More precisely, we assess the ability of the model to reproduce key phenological processes (budburst, leaf development, yellowing and fall) as well as water fluxes.

Three variants are used to predict budburst (Uniforc, Unichill and Sequential), which differ regarding the inclusion of chilling and/or forcing periods and the calculation of the coldness or heat accumulation. Among the three, the Sequential approach is the least biased (overestimation of 2.46 days) while Uniforc (chilling not considered) best accounts for the interannual variability (Pearson's r = 0.68). For the leaf development, yellowing and fall, predictions and observation are in accordance. Regarding the water balance module, the predicted throughfall is also in close agreement with the measurements (Pearson's r = 0.856, bias = -1.3%) and the soil water dynamics across the year is well-reproduced for all the study sites (Pearson's r comprised between 0.893 and 0.950, and bias between -1.81 and -9.33%). The model also well reproduced individual transpiration for sessile oak and European beech with similar performances at the tree and stand scale (Pearson's r of 0.84 – 0.85 for sessile oak and 0.88 – 0.89 for European beech). The good results of the model assessment will allow us to use it reliably in projection studies to evaluate the impact of climate change on tree growth in structurally complex stands and test various management strategies to improve forest resilience.

## 1 Introduction

Climate projections for the future indicate a substantial increase in air temperature all over Europe (between 1.0 and 5.5°C depending on the greenhouse gas emission scenario) and a change in precipitation regime variable according to the region (Jacob et al., 2014; Kovats et al., 2014). Climate extremes (*e.g.* heat waves and droughts) are also predicted to increase in intensity and frequency (Dai, 2013; Jacob et al., 2018). These changing climate conditions affect forest growth and mortality (Allen et al., 2015; Teskey et al., 2015; Charru et al., 2017; Kornhuber et al., 2019) and have an impact on the provision of ecosystem services (Hassan et al., 2005; Shvidenko et al., 2005; Rasche et al., 2013). Among others, forests play an important role in regulating the climate system by sequestering carbon in biomass and soil (Myhre et al., 2013; Le Quéré et al., 2018) and by determining water and energy exchanges with the atmosphere through their evapotranspiration and land surface properties (e.g. albedo, roughness) (Bonan, 2008; Stocker et al., 2013).

Since climate change affects some tree growth processes positively and others negatively and given the interactions among factors as well as the feedback and acclimation mechanisms, it is not easy to predict the resulting effect on tree growth at a given site (Lindner et al., 2014; Herr et al., 2016). Knowledge about climate change has been acquired based on long-term monitoring studies that are limited to the observed changes (Bussotti and Pollastrini, 2017; Etzold et al., 2019) and on experiments of environment manipulation generally analysing one or two factors at a time on a limited period (Ainsworth and Long, 2005; Norby et al., 2010; Wolkovich et al., 2012; Meir et al., 2015). In order to apprehend the complex functioning of forest ecosystems, the use of process-based modelling is a complementary approach that allows integrating and structuring the existing knowledge and making extrapolations for unprecedented conditions like those projected for the coming decades.

Process-based models were originally built to predict forest growth response to environmental changes at stand level without accounting for management operations and canopy heterogeneity. Such models were therefore suitable for pure even-aged stands but hardly manage to simulate mixed and structurally-complex stands (Dufrêne et al., 2005; Pretzsch et al., 2007). Yet, nowadays, a promising way to adapt forests to climate change is to progressively turn them into uneven-aged and mixed stands using continuous cover forestry and natural-disturbance based management to improve their stress resistance and resilience (DeRose and Long, 2014; Messier et al., 2015; Anderegg et al., 2018). To account for the spatial heterogeneity, some process-based models were designed or adapted to simulate various tree cohorts (Collalti et al., 2016). However, this approach only considers the vertical dimension of spatial heterogeneity while implementing innovative forestry practices in structurally complex stands requires to account for the horizontal dimension through a spatially explicit approach at tree level (Pacala and Deutschman, 1995; Pretzsch et al., 2007; Berger et al., 2008; Bravo et al., 2019).

To reproduce the complexity of forest ecosystem functioning in mixed and structured forests, models must take individual variability, local environment and tree adaptive behaviour into account (Berger et al., 2008). Tree size and species influence physiological and morphological properties that in turn affect the main growth processes (Binkley et al., 2013). Considering average individuals is therefore a rough approximation and does not allow accounting for all the variability within a heterogeneous forest (Berger et al., 2008). Even in cohort-based approaches, tree grouping can only be done on a limited

number of criteria that are not necessarily representative of the whole tree diversity. The local environment of a tree is strongly influenced by the neighbouring trees which modify the resource level through positive and negative interactions with the target tree (Grossiord et al., 2014). As trees compete for limited resources, neighbouring trees can decrease light, water and nutrient availability. Tree species can however develop strategies to avoid competition by using different temporal and spatial niches (complementarity, Grossiord, 2018). Positive interactions may also occur when the neighbouring trees improve the growing conditions of the target trees (facilitation, Pretzsch et al., 2013). Finally, trees adapt their morphology and physiological behaviour to the local environmental conditions by optimizing carbon allocation in order to maximise the acquisition of the limiting resource (Petritan et al., 2009; Yuang et al., 2019).

As this study focus on phenology and water cycling, we briefly review how these processes are influenced by tree characteristics and local environment. Phenology timing varies among tree species, which favours early-leafing species but can also expose them to late frosts (Lopez et al., 2008; Liu et al., 2018). Many studies report that leaf development starts earlier and leaf senescence occurs later in the understory compared to the overstory (Gill et al., 1998; Seiwa 1999a; Augspurger and Bartlett, 2003; Schieber, 2006; Vitasse, 2013; Gressler et al., 2015) which allows the understory trees to benefit from a longer growing period and consequently, to increase their productivity (Jolly et al., 2004). Warmer temperatures in the understory is one of the hypotheses advanced to explain this difference in budburst between under- and overstory (Augspurger and Bartlett, 2003; Schieber, 2006). Using a construction crane, Vitasse (2013) tested this hypothesis by transplanting seedlings of 5 tree species at 30 and 35 m height in the canopy. He observed that the budburst of the seedling growing at these heights was much earlier than that of the dominant trees. He concluded that the main factor to explain this difference in budburst is driven by ontogeny (tree age and height) as stated by Seiwa (1999b) and that the vertical profile in temperature within the canopy only plays a secondary role. To capture the differences in budburst between understorey and dominant trees, ontogeny must be taken into account in priority.

Drought stress occurs when trees cannot anymore adjust their water use to soil water availability, which reduces growth and can even lead to mortality at short or medium term due to hydraulic failure or progressive carbon starvation (McDowell and Allen, 2015; Meir et al., 2015; Greenwood et al., 2017). The stomatal control of water use varies among tree species and depends on tree size (Martínez-Vilalta and Lloret, 2016). In general, stomatal conductance decreases with tree height which can be related to the fact that taller trees experience higher hydraulic resistance, higher soil-to-leaf water potential differences and are more vulnerable to cavitation (Grote et al., 2016). For the same climate conditions above the forest canopy, water demand vary with the degree of crown shading (local microclimate) which depends on the crown position within the canopy (Bennett et al., 2015). All in all, dominant trees are more susceptible to drought stress and mortality since they are more exposed to stressful conditions (excessive radiation, high vapour pressure deficit and elevated temperature) and present a higher risk of cavitation (Grote et al., 2016; Rötzer et al., 2017). In addition, as dominant trees have higher evapotranspiration rate, the soil water reserves in their surroundings is more rapidly depleted which is however partly compensated by deeper rooting and horizontal water redistribution. These dominant trees reduced water availability for suppressed ones but, at the

same time, protect them from stressful conditions. Complementarity in water use can occur when trees of different size and species take up water from different soil layers (Schwendenmann et al., 2015). This can also result in facilitation through hydraulic lift (Zapater et al., 2011). Mixed and structured stands promote facilitation and complementarity in water use but can also lead to faster exploitation of soil water reserves (Schäfer et al., 2018).

Modelling the complex functioning of heterogeneous forests is rather challenging. A more detailed representation of tree interactions comes at the price of a higher complexity, eventually lower robustness and longer computing times. One needs however spatially explicit individual-based models for gaining a mechanistic and comprehensive understanding of tree interactions and for comparing various spatial representations of stand structure in order to select the best one for the considered process (Berger et al., 2008; Bravo *et al.*, 2019). Among others, such models allow to take tree spatial configuration into

account and distinguish between stands composed of the same trees but with a contrasted spatial aggregation (*e.g.* intimate *vs* patch-wise mixture). We decided therefore to develop a spatially explicit individual-based model called HETEROFOR for HETEROgeneous FOrest.

The processes regulating the carbon fluxes and the dimensional growth constitute the core of the HETEROFOR model and are described in Jonard et al. (accepted with major revisions, 2019). Here, we focus on the description of two modules essential

for predicting the impact of climate change on tree growth: phenology and water balance (Park et al., 2016; Choat et al., 2018). Phenology is described at the species level with the possibility to make it dependent on tree size. Water balance can be achieved at the tree level or at the stand level by aggregation of individual tree properties. We used data from long-term forest monitoring to evaluate the capacity of the model to reproduce key phenological phases (budburst, leaf development, yellowing and fall) and the soil water content dynamics as well as to estimate individual transpiration, stand throughfall and deep drainage.

Evaluating each module separately is necessary to ensure the consistency of the whole model (Soares et al., 1995).

## 2 Material and Methods

### 2.1 Model description

#### 2.1.1 Overall model

HETEROFOR is a model hosted in CAPSIS (Computer-Aided Projections of Strategies In Silviculture), a software platform
for forest growth simulations (Dufour-Kowalski et al., 2012) that provides the execution system and procedures to run
simulations and display the outputs. Still, apart from these data structures and operative methods, all initialisation and evolution
procedures are specific to HETEROFOR. The initialisation phase of the model consists in loading different files (tree species
parameters, tree and stand characteristics, chemical and physical soil properties, meteorological data and fruit production data)
in order to create trees and soil horizons. Then, tree growth is calculated yearly according to the HETEROFOR methods
presented in Jonard et al. (accepted with major revisions, 2019). So far, HETEROFOR is adapted and calibrated only for
deciduous species but the adaptation to evergreen species is under progress.

Once the initialisation is completed, the first routine called is the calculation of phenological periods from meteorological data,
which is described is Sect. 2.1.2. This function provides key phenological dates and daily foliage state (foliage development
stage and green *vs* discoloured leaf proportion) during the year. These phenological outputs are notably used for the radiation
budget carried out using the SAMSARALIGHT library coupled to HETEROFOR (Courbaud et al., 2003). According to a ray
tracing approach and based on the solar radiation from the meteorological file, this library differentiates the direct and the
diffuse components of the global radiation and determines, for both, the part of energy absorbed by the crown and the trunk of
each tree and the part transmitted to the forest floor. The intercepted radiation is required to estimate evapotranspiration and
tree photosynthesis. All aboveground and belowground water fluxes are calculated according to the processes described in
Sect. 2.1.3, which allows to perform hourly a water balance for each soil horizon at the tree or stand scale.

For each tree, GPP is estimated either annually with a radiation use efficiency approach or daily using the photosynthesis
method implemented in the model CASTANEA of CAPSIS (Dufrêne et al., 2005). In the latter case, the daily GPP is cumulated
over the year. At the end of the year, a part of the annual GPP is used for growth and maintenance respiration, the remaining
part constituting the NPP. Maintenance respiration can be estimated as a fraction of the GPP or calculated for each tree
compartment by a method accounting for the living biomass, its nitrogen concentration and a $Q_{10}$ function that describes the
temperature dependence. Growth respiration corresponds to a fraction of the carbon used to build the new tissues. NPP is then
distributed to the different tree compartments (branches, trunk, roots, leaves) giving priority to the functional organs, namely,
leaves, fine roots and fruits. The carbon sharing between leaves and fine roots depends on the tree nutritional status, trees with
a poorer nutrient status allocating relatively more carbon to fine roots. After carbon allocation to leaves, fine roots and fruits,
the residual NPP is distributed to structural tree parts (stem, branches, coarse roots) based on biomass allometry relationships.
All these processes involving carbon fluxes are described in details in Jonard et al. (accepted with major revisions, 2019). The
HETEROFOR model also contains a tree nutrition and nutrient cycling module that will be described later.

### 2.1.2 Phenological module

The phenological module aims at predicting the temporal variation of the foliage status during the vegetation period. From budburst, leaf biomass progressively increases until a maximum value, then remains constant and finally decreases during leaf fall. This temporal evolution is characterized by the proportion of leaf biomass relatively to its maximum value at full leaf development. In addition, two types of leaves are distinguished: green and discoloured leaves. The green leaf proportion is the ratio between the green leaf and the maximum leaf biomass. These two foliage properties are key variables to simulate energy, water and carbon fluxes within the forest ecosystem. Photosynthesis and tree transpiration are dependent on the proportion of green leaves since they are not active anymore on discoloured leaves. When leaves start yellowing, they still intercept rainfall while their photosynthetic activity and transpiration are progressively reduced.

The following phenological phases are distinguished, in chronological order:

- Chilling period: accumulation of coldness that breaks the dormancy. It is initiated at the chilling starting date ($t_0$) and ends at the forcing starting date ($t_1$).
- Forcing period: accumulation of heat that initiates the leaf development in the bud and leads to the budburst (budburst date = $t_{2a}$).
- Leaf development: progressive growth of the leaves from budburst to the complete leaf development (leaf development date = $t_{2b}$).
- Ageing: accumulation of coldness that is initiated at the ageing starting date ($t_3$) and ends at the yellowing starting date ($t_{4a}$).
- Yellowing: loss of photosynthetic activity linked to the decrease of day length. This phase ends at the yellowing ending date ($t_{4b}$).
- Falling: the fall of the dead leaves starts ($t_{5a}$) when less than 60% of the leaves are still green and continues until the leaf fall ending date ($t_{5b}$).

Since the phenological timing can vary considerably between species, the phenology dates are calculated for each tree species separately. Intra-specific differences are also likely to occur according to the size or social status (Cole and Sheldon, 2017) and can be optionally accounted for as described later.

The phenological module is optional in HETEROFOR. Activating the phenology requires an hourly meteorological file. If not activated, the model uses the budburst and leaf fall dates provided by the user and identical for all years and tree species.

The principle behind the whole phenology module is similar for each phase. A *state* variable is increasing progressively growing at a *rate* depending on meteorological conditions (air temperature). When the phase *state* reaches a certain *threshold,* the start of a new phase is triggered, except for the leaf yellowing and fall that are partly simultaneous.

Three routines are implemented so far to calculate the average budburst date ($t_{2a}$): the Uniforc (Chuine, 2000), the Unichill (Chuine, 2000) and Sequential (Kramer, 1994) models. The first is a one-phase model that only considers forcing while the latter ones are two-phase models integrating both chilling and forcing.

The Unichill model starts to operate when the day of year corresponds to the chilling starting date ($t_0$). At this moment, the daily chilling rate ($R_c$) is calculated according to a sigmoïd function:

$$R_c = \begin{cases} \frac{1}{1+e^{Ca(T-Cc)^2+Cb(T-Cc)}}, & -5 \leq T \leq 10 \\ 0, & T > 10 \; or \; T < -5 \end{cases} \tag{1}$$

with

Ca, Cb and Cc (°C), chilling parameters

T, the daily average temperature (°C).

This rate is summed each day until reaching the chilling threshold ($C^*$) that triggers the forcing process and sets the forcing starting date ($t_1$) to the current day. For the Uniforc model, $t_1$ is fixed. Regarding the forcing period, the forcing rate ($R_f$) is calculated using the following equation in both models:

$$R_f = \begin{cases} \frac{1}{1+e^{Fb(T-Fc)}}, & T > 0 \\ 0, & T \leq 0 \end{cases} \tag{2}$$

with

Fb and Fc (°C), forcing parameters.

The budburst is activated when the sum of the daily forcing rates reaches the forcing threshold ($F^*$).

For the sequential model, the following equations are considered for $R_c$ and $R_f$:

$$R_c = \begin{cases} 0, & T \leq T_{min} \\ \frac{T-T_{min}}{T_{opt}-T_{min}}, & T_{min} < T \leq T_{opt} \\ \frac{T-T_{max}}{T_{opt}-T_{max}}, & T_{opt} < T \leq T_{max} \\ 0, & T \geq T_{max} \end{cases} \tag{3}$$

$$R_f = \frac{Fa}{1+e^{-Fb(T-Fc)}} \tag{4}$$

with

$T_{min}$, $T_{max}$ and $T_{opt}$, the minimum, maximum and optimal temperatures (°C), respectively,

Fa , Fb and Fc (°C), forcing parameters.

We implemented these three routines for modelling budburst to allow the user to compare them and to choose the most appropriate. We suggest to retain the budburst model that best reproduce the current observations in the region of interest. Using the three approaches could also be interesting to characterize the conceptual uncertainty.

As the module was calibrated based on observations carried out on trees representative of the stand, the predicted budburst starting date is expected to be that of an average tree. Since, at this date, the leaf expansion of some trees has already started in real conditions, the model shifts the budburst date to correspond to that of the earliest trees. This budburst shift, t2a_shift, is equal to half the period between the budburst of the first and the last tree and must be provided by the user for the various tree species. By doing so, leaf development starts early for all trees which follow a same average evolution when belonging to a same tree species.

Once the budburst starting date ($t_{2a}$) is calculated, the equations for the subsequent phenological variables are the same. The leaf development rate ($R_{ld}$) is cumulated daily until the leaf development threshold ($LD*$) is reached. It is computed according to:

$$R_{ld} = \begin{cases} T, & T > 0 \\ 0, & T \leq 0 \end{cases} \tag{5}$$

where $T$ is the daily average temperature of the current day (°C).

The leaf proportion (leafProp, g g$^{-1}$) is calculated daily for each tree species ($sp$) according to

$$leafProp_{sp\_t} = \frac{\sum_{t_{2a}}^{t} R_{ld}}{LD*} \tag{6}$$

with $t$, the current day.

As many studies have shown that budburst in the understory occurs earlier than in the overstory and ascribed this primarily to ontogeny (Gill et al., 1998; Seiwa 1999a; Seiwa, 1999b; Augspurger and Bartlett, 2003; Schieber, 2006; Vitasse, 2013), we implemented an option to make the phenology size-dependent (phenology at tree level). With this option, the leaf development is first triggered in the smallest trees of each tree species and then progressively in the tallest ones according to their height. At the stand level, the option 'phenology at tree level' provide exactly the same leaf development than the default option but the difference appear at the tree scale. The default option assumes that all trees of a same species initiate budburst at the same time and display the same progressive leaf development while the alternative one supposes that trees break down one after the other depending on their size.

With the option 'phenology at tree level', the leaf proportion of each tree ($leafProp_{tree\_t}$) is updated daily ($t$) between the budburst starting date ($t_{2a}$) and the budburst ending date ($t_{2b}$) based on the leaf proportion calculated at the stand scale for the corresponding tree species ($leafProp_{sp\_t}$):

$$leafProp_{tree\_t} = \begin{cases} 1, & \frac{\sum_{1}^{tree} a_{leaf}}{A_{leaf}} \leq leafProp_{sp\_t} \\ 0, & \frac{\sum_{1}^{tree} a_{leaf}}{A_{leaf}} > leafProp_{sp\_t} \end{cases} \tag{7}$$

With

$tree$, the tree of interest (note that the trees are sorted by ascending order based on their height),

$a_{leaf}$, the tree leaf area (m²),

$A_{leaf}$, the total stand leaf area (m²).

A fixed date, defined according to Dufrêne et al. (2005), is considered for the start of the ageing process ($t_3$). This process does not alter leaf quality but is a prerequisite for leaf yellowing ($t_{4a}$) that is initiated when the cumulated daily ageing rate ($R_{age}$) equals the ageing threshold ($A*$), with

$$R_{age} = \begin{cases} T_{b\_age} - T, & T < T_{b\_age} \\ 0, & T \geq T_{b\_age} \end{cases} \tag{8}$$

with

$T_{b\_age}$, the base temperature for ageing (°C).

The leaf yellowing calculation gives the green leaf proportion, $greenProp$ (g g$^{-1}$), which provides the fraction of remaining green leaves compared to the maximum green leaf amount for each tree species. It is set to 1 before the start of yellowing, and then decreases with day length according to the following equation:

$$greenProp_{sp\_t} = greenProp_{sp\_t-1} * \left(\frac{DL_t - DL_{min}}{DL_{t4a} - DL_{min}}\right)^y \tag{9}$$

with

$DL_t$ and $DL_{t4a}$, the day lengths (hours) for the current day and $t_{4a}$, respectively,

$DL_{min}$, the minimum day length (hours) value over the year, and

y, a leaf yellowing parameter.

The day length (hours) is calculated according to Teh (2006):

$$DL = \frac{24}{\pi} * acos\left(-\frac{\sin(\delta) * \sin(\lambda)}{\cos(\delta) * \cos(\lambda)}\right) \tag{10}$$

where $\lambda$ is the site latitude (rad) and $\delta$, the solar declination (rad) determined as $\delta = -\frac{23.45 * \pi}{180} * \cos\left(2\pi \frac{DOY+10}{365}\right)$

and $DOY$ is the day of year (i.e., Jan 1=1, Jan 2=2, Feb 1=32…).

The yellowing phase ends when the green leaf proportion drops below a threshold, called yellowing threshold, Y*, indicated
by the model user in the species file. The leaf fall ($t_5$) is set to start rapidly after yellowing initiation, namely, when $greenProp$ reaches 0.60, considering that leaves no longer photosynthetically active can quickly fall.

The falling rate ($R_{fall}$) is calculated daily and is used to update $leafProp$ for each tree species. It depends on the wind and frost episodes. While the frost weakens the leaf petiole, the wind can break it and take away the leaf. For this reason, $leafProp$ is determined as follows for each day $t$:

$$leafProp_{sp\_t} = leafProp_{sp\_t-1} - f_{ampl} * WS * R_{fall} \tag{11}$$

with

$f_{ampl}$, a frost amplifier coefficient fixed to 1 before the occurrence of five consecutive hours with air temperature below 0°C and is then set to 2 and 3 for oak and beech, respectively,

$WS$ is the daily average wind speed (m s$^{-1}$),

$R_{fall}$ is the falling rate (s m$^{-1}$ d$^{-1}$) calibrated as described in Sect. 2.2.

According to Eq. (11), $leafProp_{sp\_t}$ progressively decreases from 1 to 0 but it cannot take a value below $greenProp_{sp\_t}$, accounting for the fact that green leaves are not expected to fall. Finally, when all leaves have fallen, the trees enter in the leafless period until the budburst of the following year.

As for leaf development but with a reverse order, the option '*phenology at tree level*' first triggers the leaf yellowing and fall
in the taller trees and then in the smaller ones in order to reproduce the observations reported by Gressler et al. (2015). This options daily updates the green leaf and leaf proportions of each tree ($greenProp_{tree\_t}$, $leafProp_{tree\_t}$) between the yellowing

starting date ($t_{4a}$) and the falling ending date ($t_{5b}$) based on the green leaf and leaf proportions calculated at the stand scale for the corresponding tree species ($greenProp_{sp\_t}, leafProp_{sp\_t}$):

$$greenProp_{tree\_t} = \begin{cases} 1, & \frac{\sum_n^{tree} a_{leaf}}{A_{leaf}} \leq greenProp_{sp\_t} \\ 0, & \frac{\sum_n^{tree} a_{leaf}}{A_{leaf}} > greenProp_{sp\_t} \end{cases} \tag{12}$$

$$leafProp_{tree\_t} = \begin{cases} 1, & \frac{\sum_n^{tree} a_{leaf}}{A_{leaf}} \leq leafProp_{sp\_t} \\ 0, & \frac{\sum_n^{tree} a_{leaf}}{A_{leaf}} > leafProp_{sp\_t} \end{cases} \tag{13}$$

With

*tree*, the tree of interest (note that the trees are sorted by descending order based on their height),

$a_{leaf}$, the tree leaf area (m²),

$A_{leaf}$, the total stand leaf area (m²).

The option '*phenology at tree level*' gives the opportunity to compare two contrasted hypotheses regarding individual tree phenology and to evaluate to which extent it has an impact on tree growth.

**2.1.3 Water balance module**

The water balance module operates at an hourly time step and simulates the partitioning of incident rainfall into the main forest water fluxes and pools, namely, interception (i.e., water storage on foliage and bark, and evaporation), throughfall, stemflow, water movements between soil horizons and deep drainage, transpiration and soil water uptake in the different soil horizons, and soil evaporation (Fig. 1). Surface runoff and groundwater level rise are not yet considered in the current HETEROFOR version.

In a first step, the parameters considered as constant during the leaved and leafless periods are estimated. Then, the various water fluxes are calculated at an hourly time step. The default option for the water balance module calculates the water fluxes at the stand level by summing properties estimated at the tree level (maximum foliage and bark storage capacities, throughfall and stemflow proportions). For this option, tree transpiration is calculated at the tree level and summed at the stand scale. Stand transpiration is then used to estimate root water uptake in the different soil horizons assuming that all trees are taking up water in the same reservoirs in which soil water is redistributed homogeneously between two hourly time steps. This hypothesis can be justified by soil anisotropy, which induces a higher horizontal than vertical soil conductance. This is justified since water movements through the same horizon depend only on its own hydrological properties while the presence of one horizon with a low conductance can slow down vertical water movement in the upper horizons (Todd and Mays, 2005). Moreover, as sediments are preferentially deposited on their longest side, the vertical conductance is decreased with regards

to the horizontal one (Cristiano et al., 2016) so that the ratio of the horizontal *vs* vertical conductance ranges between 2 and 10 in alluvial soils and amounts to 100 in clay soils (Todd and Mays, 2005).

The user can select an alternative option '*activate fine spatial resolution*' to perform water balance on an individual scale. In this case, all the water fluxes (throughfall, stemflow, foliage, bark and soil evaporation, transpiration, water uptake, soil water movements and drainage) are calculated at the individual level. For this option, the model distributes the total soil volume in individual soil volumes (called pedon).

The pedon area ($a_{pedon}$) is determined proportionally to the leaf area of the associated tree (but is limited to two times its crown projection):

$$a_{pedon} = \frac{a_{leaf}}{A_{leaf}} . A_{stand} \tag{14}$$

with     $a_{leaf}$, the tree leaf area (m²)

$A_{leaf}$, the total stand leaf area (m²)

$A_{stand}$, the total stand area (m²)

In sparse stands, all the stand area is not allocated to the trees and the remaining area is considered as a pedon without any associated tree. With the fine spatial resolution, the model performs a water balance for each tree pedon and also for the remaining pedon (without tree). Contrary to the default option, the alternative option supposes no water redistribution among pedons. This hypothesis could become more appropriate than the perfect redistribution hypothesis when soil dries (Friedman and Jones, 2001), at least beyond the air entry value (Assouline and Or, 2006). The two options allow the user to test two contrasted hypotheses regarding soil water redistribution in the horizontal dimension. In the following description, variables calculated at the stand scale are represented with capital letters while lowercase letters are used for variables at the tree level. In some cases, when the equation is the same at the tree and the stand level, the variables are represented only with capital letters to avoid unnecessary duplications.

*Foliage and bark storage capacity*

The maximum foliage storage capacity of a tree ($c_{foliage\_max}$, l) is calculated by multiplying the foliage storage capacity of the corresponding tree species by the tree leaf area:

$$c_{foliage\_max} = a_{leaf} \cdot c_{foliage\_sp} \tag{15}$$

with

$c_{foliage\_sp}$, the foliage storage capacity for the species *sp* (mm or l per m² of leaf).

To obtain it at the stand level ($C_{foliage\_max}$, l), the model sums the maximum foliage storage capacity of all the trees.

Bark storage capacity depends on season (i.e., leafed and leafless periods) and on tree species. It is derived from a linear model proposed by André et al. (2008a) predicting the individual stemflow (*sf*, l) produced during a rain event as a function of tree girth (*C130*, cm) and rainfall amount (*R*, mm):

$$sf = a + b.C130 + c.R + d.C130.R + \tau + \delta + \varepsilon \tag{16}$$

where $a$ (l), $b$ (l cm$^{-1}$), $c$ (m²) and $d$ (m² cm$^{-1}$) are fixed effect parameters varying with tree species and season, $\tau$ and $\delta$ are random factors characterizing the tree and the rain event variability and $\varepsilon$ account for the residuals.

As it multiplies the rainfall amount in Eq. (16), the term "$c + d.C130$" may be interpreted as an estimate of the stemflow rate ($sf_{rate}$, l mm$^{-1}$). In parallel, André et al. (2008a) determined the rainfall threshold for stemflow appearance ($R_{min}$, mm), defined as the amount of rainfall required to produce stemflow at the base of the trunk. This threshold was found to be independent of tree size while it depends on both season and tree species. Multiplying the $sf_{rate}$ estimations by $R_{min}$ values for the corresponding species and season provides estimates of the tree bark storage capacity ($c_{bark}$, l), namely, the amount of water accumulated on branch and trunk bark before stemflow occurs at tree base:

$$c_{bark} = (c + d \cdot C130) \cdot R_{min} \tag{17}$$

The individual $c_{bark}$ estimates are then summed over all trees of a same species for each season to determine leafless (*ll*) and leaved (*ld*) stand bark storage capacity ($C_{bark\_sp\_ll}, C_{bark\_sp\_ld}$, l). As shown by André et al. (2008a), the seasonal variation of the bark storage capacity is not significant since the corresponding changes in the three parameters (c, d and Rmin) offset each other. We maintained, however, the distinction between seasons since the parameters of Eq. (16) were also used to estimate throughfall and stemflow proportions (described hereafter), which are clearly season-dependent.

*Throughfall and stemflow proportions*

For a given tree, the proportion of stand rainfall reaching the ground at the base of the trunk as stemflow may be calculated by dividing the stemflow rate (see above) by the pedon or stand area ($a_{pedon}$ or $A_{stand}$, m²) depending on the selected option (tree *vs* stand scale water balance):

$$\%sf = \frac{c + d \cdot C130}{a_{pedon}} \text{ or } \frac{c + d \cdot C130}{A_{stand}} \tag{18}$$

For the water balance at the stand scale, the stemflow proportion per tree species is then calculated separately for the leafless and the leaved periods ($\%SF_{sp\_ll}, \%SF_{sp\_ld}$) by summing the corresponding tree stemflow proportions. The stemflow proportion is also calculated at the stand scale for each period ($\%SF_{ll}, \%SF_{ld}$). Finally, tree and stand level throughfall proportions are obtained directly from the stemflow proportions:

$$\%tf_{ll} = 1 - \%sf_{ll} \text{ or } \%TF_{ll} = 1 - \%SF_{ll} \tag{19}$$

$$\%tf_{ld} = 1 - \%sf_{ld} \text{ or } \%TF_{ld} = 1 - \%SF_{ld} \tag{20}$$

*Absorbed radiation proportions*

During the leaved period, the radiation absorbed by the trees is provided by the SAMSARALIGHT library either for the whole period (simplified radiation balance, default option) or for every hour of key phenological dates (detailed radiation balance, alternative option). It may be determined either by considering absorption by tree crowns as a function of leaf area density and

ray path length through the crown by applying the Beer-Lambert law, or by specifying relative crown radiation absorption coefficients for each species. At the tree scale, the proportion of incident radiation absorbed per unit of leaf area during the vegetation period ($\%aRAD_{tree\_canopy\_m^2}$) is calculated as the ratio of the radiation absorbed by the crown over the whole vegetation period ($aRAD_{tree\_crown}$, $MJ$) divided by the corresponding incident radiation (*RAD,* MJ m⁻²) and the tree leaf area:

$$\%aRAD_{tree\_leaf\_m^2} = \frac{aRAD_{tree\_crown}}{RAD \cdot a_{leaf}} \tag{21}$$

At the stand scale, this proportion is obtained by summing the radiation absorbed by each crown and dividing it by the incident radiation and the leaf area of the whole stand:

$$\%aRAD_{stand\_leaf\_m^2} = \frac{\sum_{tree} aRAD_{tree\_crown}}{RAD \cdot A_{leaf}} \tag{22}$$

Similarly, the proportion of incident radiation absorbed per unit of bark area is obtained, at the tree and stand scales respectively, by

$$\%aRAD_{tree\_bark\_m^2} = \frac{aRAD_{tree\_trunk}}{RAD \cdot a_{bark}} \tag{23}$$

with

$aRAD_{tree\_trunk}$, the radiation absorbed by the trunk of a given tree (MJ),

$a_{bark}$, the tree bark area (m²)

$$\%aRAD_{stand\_bark\_m^2} = \frac{\sum_{tree} aRAD_{tree\_trunk}}{RAD \cdot A_{bark}} \tag{24}$$

with

$A_{bark}$, the stand bark area (m²)

At both scale (tree and stand), the proportion of incident radiation transmitted to the understorey is the transmitted radiation ($transRAD$, MJ $m^{-2}$), determined as the difference between the incident radiation and the radiation absorbed by the tree(s), divided by the incident radiation:

$$\%transRAD = \frac{transRAD}{RAD} \tag{25}$$

The radiation transmitted to the understory is then partitioned into the radiation intercepted by the ground vegetation and that reaching the soil by applying Beer-Lambert law considering the ground vegetation leaf area index (described later in *Ground vegetation transpiration and soil evaporation*).

In the following sections, all these proportions are used to estimate the hourly absorbed or transmitted radiations based on the hourly incident radiation.

For the leafless period, the proportions of incident radiation intercepted by the trunks and the branches and transmitted to the understory are obtained based on the Beer-Lambert law using the bark area index (i.e. bark surface divided by the stand or pedo area, *BAI*, m² ⁻²) calculated from the bark biomass, density and thickness:

$$\%aRAD_{bark\_m^2} = \frac{1-\exp(-k \cdot BAI)}{BAI} \tag{26}$$

$$\%transRAD = \frac{\exp(-k \cdot BAI)}{BAI} \tag{27}$$

*Interception and evaporation of water stored on foliage and bark*

Based on the preceding calculations, the water balance module starts updating the different water fluxes and pools for every hourly time step. First water evaporation from foliage and from bark is computed using the Penman Monteith (P-M) equation (Monteith, 1965) at the tree or stand scale. The latent heat flux density is calculated as follows:

$$\lambda.E = \frac{\Delta R + \frac{\rho.c_p.VPD}{r_a}}{\Delta + \gamma\left(\frac{r_a+r_s}{r_a}\right)} \tag{28}$$

with

$\lambda.E$: latent heat flux density (W m$^{-2}$),

$\lambda$: water latent heat of vaporization = 2454000 J kg$^{-1}$ (Teh, 2006),

$\gamma$: psychometric constant = 0.658 mbar K$^{-1}$ (Teh, 2006),

$\Delta$: slope of the saturated vapour pressure curve (mbar K$^{-1}$):

$$\Delta \approx \frac{de_s(T)}{dT} = \frac{25029.4 \cdot exp\left[\frac{17.269.T}{T+237.3}\right]}{(T+237.3)^2}, \tag{29}$$

$\rho$: moist air density = 1.209 kg m$^{-3}$,

$c_p$: moist air specific heat capacity = 1010 J kg$^{-1}$ K$^{-1}$,

$T$: air temperature (°C),

$R$: absorbed radiation per unit of leaf or bark area (Watt per m² of leaf/bark),

$r_a$: aerodynamic resistance (s m$^{-1}$), the inverse of aerodynamic conductance, ga:

$$r_a = \frac{1}{g_a} \tag{30}$$

$r_s$: surface resistance (s m$^{-1}$), the inverse of surface conductance, $g_s$:

$$r_s = \frac{1}{g_s} \tag{31}$$

$VPD$: the vapour pressure deficit (mbar or hPa) calculated as follows based on the air temperature and the relative humidity:

$$VPD = e_s(T) - e_r \tag{32}$$

with

$e_s$: saturated vapour pressure (mbar):

$$e_s(T) = 6.1078 . exp\left[\frac{17.269T}{T+237.3}\right] \tag{33}$$

$e_r$: air vapour pressure (mbar):

$$e_r = \frac{RH}{100} \cdot e_s(T_r) \tag{34}$$

where $RH$ is the relative humidity ($10^{-2}$ hPa hPa$^{-1}$)

The radiation absorbed hourly per unit of leaf area ($h\_aRAD_{leaf\_m^2}$, W.m$^{-2}$) is obtained by multiplying the proportion of incident radiation absorbed per leaf area unit by the hourly incident radiation ($h\_RAD$, W m$^{-2}$):

$$h\_aRAD_{leaf\_m^2} = \%aRAD_{leaf\_m^2} \cdot h\_RAD \tag{35}$$

Similarly, the hourly absorbed radiation per unit of bark area ($h\_aRAD_{bark\_m^2}$, W.m$^{-2}$) is obtained by multiplying the proportion of incident radiation absorbed by the bark by the hourly incident radiation:

$$h\_aRAD_{bark\_m^2} = \%aRAD_{bark\_m^2} \cdot h\_RAD \tag{36}$$

The aerodynamic resistance is defined as the inverse of the aerodynamic conductance, which represents the ease for a water vapour molecule to get away from its original location once it has been evaporated. Similarly, the surface resistance is the inverse of surface conductance that represents the ease for water molecules to migrate through the surface-air interface. The aerodynamic resistance depends mainly on wind speed and turbulence while the surface resistance is a function of the water diffusivity through the surface.

According to Teh (2006) and depending on the scale considered (tree or stand), the mean canopy air resistance may be obtained by integrating the canopy air conductance ($g_a$, m.s$^{-1}$) values estimated at 11 height levels between the mid-crown or mid-canopy height and the dominant height for the foliage and between half of the total or dominant height and the dominant height for the bark:

$$g_a = 0.006 \cdot \sqrt{\frac{WS}{l_{sp}}} \tag{37}$$

with

$l_{sp}$, the mean leaf width,

$WS$, the wind speed (m s$^{-1}$).

The mid-canopy height is determined as the mid-height between the dominant height of the stand ($hd$, m), defined as the mean total height of the 100 biggest trees per ha, and the canopy base height ($hcb$, m), defined as the mean height to crown base of the 100 smallest trees per ha. At the tree scale, the integration is done between the mid-crown height and the total height for the foliage and between half of the total height and the total height for the bark.

$WS$ is estimated at the different heights ($h$, m) based on the dominant height wind speed ($WS_{hd}$, m s$^{-1}$) and on the wind speed attenuation coefficient ($\alpha$):

$$WS = WS_{hd} \cdot e^{-\left[\alpha \cdot \left(1 - \frac{h}{hd}\right)\right]} \tag{38}$$

where $WS_{hd}$ is calculated according to Jetten (1996) based on the measured wind speed and its height of measurement:

$$WS(h) = WS(z_m) \cdot \frac{ln\left[(z_e - d_m)/z_{0m}\right]}{ln\left[(z_m - d_m)/z_{0m}\right]} \cdot \frac{ln\left[(h - d_f)/z_{0f}\right]}{ln\left[(z_e - d_f)/z_{0f}\right]} \tag{39}$$

with

$h$ is the height at which wind speed is estimated (in this case the dominant height),

$z_e$ is the reference height (m) fixed to 50 m,

$z_m$ is the wind speed measurement height (2.5 m),

$d_m$ is the surface roughness height (m) of the meteorological station fixed to 0.08 m,

$z_{0m}$ is the zero plane displacement (m) of the meteorological station fixed to 0.015 m,

$d_f$ is the surface roughness height (m) of the forest and estimated as $0.75 \cdot hd$ and

$z_{0f}$ is the zero plane displacement (m) of the meteorological station fixed to $0.1 \cdot hd$.

While no surface resistance is considered for the foliage evaporation (infinite conductance), the bark conductance (m s$^{-1}$) depends on the bark storage at the previous time step ($prevS_{bark\_sp}$, l) and the bark storage capacity ($C_{bark\_sp}$, l) according to

$$g_{s\_bark\_sp} = g_{s\_bark\_min} + (g_{s\_bark\_max} - g_{s\_bark\_min}) \cdot \frac{prevS_{bark\_sp}}{C_{bark\_sp}} \tag{40}$$

The latent heat flux density is then converted to hourly water evaporation ($EV$, l per hour per m² of leaf):

$$EV_{foliage\ or\ bark\_m^2} = \frac{\frac{\lambda.E}{\lambda}}{d_{H2O}} \cdot 1000 \cdot 60 \cdot 60 \tag{41}$$

with

$E$, the mass of water evaporated (kg m$^{-2}$ s$^{-1}$) and

$d_{H2O}$, the water density (998 kg m$^{-3}$)

Hourly tree or stand foliage evaporation ($EV_{foliage\_stand}$, l.h$^{-1}$) is obtained by multiplying $EV_{foliage}$ from Eq. (41) by the tree or stand leaf area:

$$EV_{foliage} = EV_{foliage\_m^2} \cdot (a_{leaf}\ or\ A_{leaf}) \tag{42}$$

Similarly, hourly evaporation from bark ($EV_{bark}$, l h$^{-1}$) is determined separately for each tree or tree species by

$$EV_{bark} = EV_{bark\_sp\_m^2} \cdot (a_{leaf}\ or\ A_{bark_{sp}}) \tag{43}$$

where $A_{bark\_sp}$ is the bark area for the tree species $sp$ (m²).

Evaporation from foliage and from bark cannot be larger than the corresponding amounts of water stored on these surfaces, namely, $S_{foliage}$(l) and $S_{bark\_sp}$ (l) (see next section). Therefore, the following conditions are set:

$$EV_{foliage} = \min(EV_{foliage}, S_{foliage}) \tag{44}$$

$$EV_{bark} = \min(EV_{bark}, S_{bark\_sp}) \tag{45}$$

*Partitioning of rainfall into interception, throughfall and stemflow*

Rainfall passing through the canopy can be intercepted by the foliage, the branches and the stems of the tree(s). These reservoirs saturate progressively and the water then flows along the trunks to the tree base(s) to produce stemflow or drips from the canopy to the ground as throughfall. For some of the parameters (i.e., storage capacities, stemflow proportions) showing contrasting values depending on the season, the leaved and the leafless periods are distinguished to describe these processes. In addition, several intermediate state variables are considered, namely:

- tree or stand rainfall $(R_{tree\ or\ stand}, l) = R \cdot (A_{pedon}\ or\ A_{stand})$; (46)

- foliage storage $(S_{foliage}, l)$ corresponding to the amount of water stored on the tree or stand foliage;

- previous stand foliage storage $(prevS_{foliage}, l)$ being the tree or stand foliage storage at the previous time step;

- remaining foliage storage capacity $(RemC_{foliage}, l)$, defined as

$$RemC_{foliage} = C_{foliage} - (prevS_{foliage} - EV_{foliage})$$ (47)

- non-intercepted rainfall $(unintR, l)$.

For the **leaved period**, the foliage storage and the non-intercepted rainfall are updated at every time step considering various cases:

if $(RemC_{foliage} > 0)$ {

    if $(RemC_{foliage} > R_{tree\ or\ stand})$ {

        $S_{foliage} = prevS_{foliage} - EV_{foliage} + R_{tree\ or\ stand}$

        $unintR = 0$ }

    else {

        $S_{foliage} = C_{foliage}$

        $unintR = R_{tree\ or\ stand} - RemC_{foliage}$ }

else {

    $S_{foliage} = C_{foliage}$

    $unintR = R_{tree\ or\ stand}$ }

For the **leafless period**, we have $C_{foliage} = 0$, which gives $unintR = R_{tree\ or\ stand}$)

Throughfall and stemflow fluxes are then calculated separately for the leaved and leafless periods. For both periods, tree or stand throughfall and pre-stemflow $(preSF, l)$ are considered as complementary fractions of the non-intercepted rainfall. Pre-stemflow is the amount of rain deviated towards the branches and the trunk but not necessarily reaching the base of the trunk due to storage and evaporation losses. At the stand level, pre-stemflow is estimated separately for each tree species.

$$TF_{tree\ or\ stand} = \%TF \cdot unintR$$ (48)

$$preSF_{tree\ or\ sp} = \%SF \cdot unintR$$ (49)

At this stage, the following state variables are used:

- the tree or species bark storage ($S_{bark}$, l) = amount of water stored in the bark of a given tree or in that of all the trees of a same tree species,

- the previous tree or species bark storage ($prevS_{bark}$, l) = tree or species bark storage at the previous time step;

- the remaining bark storage capacity of a given tree or species ($RemC_{bark}$, l):

$$RemC_{bark} = C_{bark} - (prevS_{bark} - EV_{bark}) \tag{50}$$

Similarly as above for foliage storage and non-intercepted rainfall, various cases are distinguished to hourly update the bark storage and the stemflow volume ($SF$, l) of each tree or species:

if ($RemC_{bark} > 0$) {

       if ($RemC_{bark} > preSF$) {

10               $S_{bark} = prevS_{bark} - EV_{bark} + preSF$

              $SF = 0$ }

       else {

              $S_{bark} = C_{bark}$

              $SF = preSF - RemC_{bark}$ }

else {

       $S_{bark} = C_{bark}$

       $SF = preSF$ }

At the stand scale, stemflow is obtained by summing stemflow fluxes over the tree species:

$$SF_{stand} = \sum_{sp} SF_{sp} \tag{51}$$

*Tree transpiration*

As for evaporation from foliage and bark, the Penman Monteith equation (see Eq. 28) is used to estimate hourly tree transpiration during the vegetation period. In this case, the radiation absorbed per unit of leaf area by each tree ($h\_aRAD_{tree\_leaf\_m^2}$, Watt per m² of leaf) is considered and is obtained by:

$$h\_aRAD_{tree\_leaf\_m^2} = \%aRAD_{tree\_leaf\_m^2} \cdot h\_RAD \tag{52}$$

The individual aerodynamic resistance is determined from Eq. (37) to Eq. (39) applied between the height of largest crown extension ($h_{lce}$, m) and the dominant height. The individual surface resistance ($r_{s\_foliage}$, s m$^{-1}$) is defined as the inverse of the foliage stomatal conductance ($g_{s\_foliage}$, m s$^{-1}$) which is estimated based on a potential x modifier approach considering soil and climate conditions as well as individual tree characteristics. This approach allows to account for the increase in stomatal

conductance with radiation and for the negative effect of increasing vapour pressure deficit and soil water potential (Granier and Breda, 1996; Tuzet et al., 2003; Buckley, 2017). For similar soil and climate conditions, the stomatal conductance is

acknowledged to be higher for trees with a larger sapwood to leaf area ratio and to decreases with crown height as stomata of top leaves close earlier to avoid cavitation when water stress occurs (Ryan and Yoder, 1997; Schäfer et al., 2000).

$$r_{s\_foliage} = \frac{1}{g_{s\_foliage}} \tag{53}$$

$$g_{s\_foliage} = g_{s0\_foliage} \cdot \frac{a_{sapwood}}{a_{leaf}} \cdot \frac{1}{h_{lce}} \cdot M_{radiation} \cdot M_{soil\,water} \cdot M_{vpd} \tag{54}$$

with

$g_{s0\_foliage}$: the reference stomatal conductance (m s$^{-1}$),

$\frac{a_{sapwood}}{a_{leaf}}$: the sapwood to leaf area ratio (m² m$^{-2}$) calculated at the tree level (see Jonard et al., accepted with major revisions, 2019 for details),

$M_{radiation}$: the radiation modifier $= \frac{h\_aRAD_{tree\_leaf\_m^2}}{h\_aRAD_{tree\_leaf\_m^2} + p_{radiation}}$, $\tag{55}$

where $p_{radiation}$ is a parameter characterizing stomatal response to radiation.

$M_{soil\,water}$: the soil water modifier $= e^{-p1_{SW}(pF-2.5)^{p2_{SW}}}$ when $pF > 2.5$, 1 otherwise $\tag{56}$

where $pF$ (cm) is the base-10 logarithm of the mean soil water potential ($\phi$) (mean value of the various horizons weighted based on root proportion, see below in the "root water uptake" section for calculation details of the soil water potential) and $p1_{SW}$ and $p2_{SW}$ are two parameters characterizing the stomatal response to soil water potential.

$M_{vpd}$, the VPD modifier $= 1.0 - p_{VPD} \cdot \ln VPD$. $\tag{57}$

where $p_{VPD}$ is a species-dependent parameter characterizing stomatal response to vapour pressure deficit.

The latent heat flux density (W m$^{-2}$) determined by applying this parametrization to Eq. (28) is then converted to tree transpiration ($TR_{tree}$, l h$^{-1}$) using the same approach as for foliage evaporation that was described in Eq. (41) and Eq. (42). Finally, $TR_{tree}$ is corrected by multiplying it by the proportion of green leaves (*greenProp*) and by the fraction of leaves not covered with water ($1 - \frac{S_{foliage}}{C_{foliage}}$), considering that transpiration occurs from photosynthetically active and dry leaves only.

*Ground vegetation transpiration and soil evaporation*

The Penman Monteith equation is also used to estimate ground vegetation transpiration and soil evaporation at the tree and stand scale. For this purpose, the radiation transmitted to the understory is subdivided for each time step into the radiation absorbed by per unit of leaf area of the ground vegetation ($h\_aRAD_{grd\_veg\_m^2}$, Watt per m² of leaf) and the radiation absorbed by the soil ($h\_aRAD_{soil\_m^2}$, W.m$^{-2}$) through application of the Beer-Lambert law:

$$h\_aRAD_{grd\_veg\_m^2} = \frac{\%transRAD \cdot rad \cdot (1 - \exp(-k \cdot LAI_{grd\_veg} \cdot greenProp_{stand}))}{LAI_{grd\_veg} \cdot greenProp_{stand}} \tag{58}$$

$$h\_aRAD_{soil\_m^2} = \%transRAD \cdot rad \cdot \exp\left(-k \cdot LAI_{grd_{veg}} \cdot greenProp_{stand}\right) \tag{59}$$

where $k$ is the extinction coefficient fixed to 0.5 (Teh, 2006), $LAI_{grd\_veg}$ is the leaf area index of the ground vegetation calculated as the difference between the ecosystem LAI and the tree or stand LAI, $greenProp_{stand}$ is the proportion of remaining green leaves at the stand level.

The energy effectively available for soil evaporation is obtained by subtracting the soil heat flux density ($G$, W m$^{-2}$) from $h\_aRAD_{soil\_m^2}$. $G$ is estimated based on the temperature gradient and the soil thermal conductivity ($K$, fixed to 0.25 W m$^{-1}$ K$^{-1}$) as follows:

$$G = K * \frac{T_{surf} - T_{int}}{th_{org}/100} \tag{60}$$

with

$T_{surf}$ (°C), the temperature at the soil surface, considered as equal to air temperature ($T$)

$T_{int}$ (°C), the temperature at the interface between the organic layers and the mineral soil (see Jonard et al., accepted with major revisions, 2019 for more information on the way $T_{int}$ is obtained),

$th_{org}$ (m), the thickness of the organic layer.

For ground vegetation transpiration and soil evaporation, the aerodynamic resistance is computed by applying Eq. (37) to (39) between the ground level and the dominant height.

The surface resistances of the ground vegetation ($r_{s\_grd\_veg}$) and of the soil ($r_{s\_soil}$) are the reciprocals of the ground vegetation and soil conductances, respectively. The ground vegetation conductance ($g_{s\_grd\_veg}$, m s$^{-1}$) is estimated based on the same approach as $g_{s\_foliage}$ for tree transpiration while the soil conductance ($g_{s\_soil}$, m s$^{-1}$) depends on the relative extractable water (see below for computation details) of the forest floor at the previous time step ($prevREW_{forest\_floor}$):

$$g_{s\_soil} = g_{s\_soil\_min} + (g_{s\_soil\_max} - g_{s\_soil\_min}) \cdot prevREW_{forest\_floor} \tag{61}$$

The latent heat flux density (W m$^{-2}$) is then converted to ground vegetation transpiration ($TR_{grd\_veg}$, l h$^{-1}$) and soil evaporation ($EV_{soil}$, l h$^{-1}$) using the same approach as for tree transpiration and foliage evaporation, Eq. (41) and Eq. (42).

*Soil hydraulic properties*

The modelling of water uptake distribution among soil horizons and of water transfer from a horizon to another requires estimates of the hydraulic properties for all soil horizons. The relationship between the soil water content ($\theta$, m$^3$ m$^{-3}$) and the absolute matric potential ($h$, cm) is described by the van Genuchten function

$$\theta = \theta_r + S \cdot (\theta_s - \theta_r) \tag{62}$$

that can be rearranged under the form

$$S = \frac{\theta - \theta_r}{\theta_s - \theta_r} \text{ and} \tag{63}$$

$$S = [1 + (\alpha|h|)^n]^{-\left(1 - \frac{1}{n}\right)} \tag{64}$$

with

$\theta_r$, the residual water content (m³ m⁻³),

$\theta_s$, the saturated water content (m³ m⁻³),

$S$, the relative water content

$\alpha$ and $n$, two parameters

The Mualem-van Genuchten function allows to estimate the soil hydraulic conductivity based on the relative water content and the saturated conductivity.

$$K = K_0 \left( S^\lambda \left\{ 1 - \left( 1 - S^{n/n-1} \right)^{1-\frac{1}{n}} \right\}^2 \right) \tag{65}$$

with

$K$, the hydraulic conductivity (cm day⁻¹),

$K_0$, the saturated conductivity (cm day⁻¹) and

$\lambda$, a parameter.

These two functions (Eqs 64 and 65) partly share the same parameters which are estimated based on soil horizon properties (i.e., organic carbon content, $C_{org}$, particle size distribution). For organic horizons, values from Dettmann et al. (2014) are used for $\alpha$, n and $\lambda$ ($\alpha$ = 0.251, n = 1.75, $\lambda$ = 0.5) and the equation of Päivänen (1973) for Sphagnum peat is considered for $K_0$.

$$K_0 = 10^{\left(-2.321 - 13.22 \cdot \rho_b \cdot \frac{1000}{1000000}\right) \cdot 24 \cdot 60 \cdot 60} \tag{66}$$

with

$\rho_b$ = bulk density (kg m⁻³)

For mineral horizons, pedotransfer equations elaborated by Weynants et al. (2009) are used:

$$\ln \alpha = -4.3003 - 0.0097 \cdot clay + 0.0138 \cdot sand - 0.0992 \cdot C_{org} \tag{67}$$

$$\ln(n-1) = -1.0846 - 0.0236 \cdot clay - 0.0085 \cdot sand + 0.0001 \cdot sand^2 \tag{68}$$

$$\ln K_0 = 1.9582 + 0.0308 \cdot sand - 0.6142 \cdot \rho_b - 0.1566 \cdot C_{org} \tag{69}$$

$$\lambda = -1.8642 - 0.1317 \cdot clay + 0.0067 \cdot sand \tag{70}$$

with

$clay$ and $sand$, the clay and sand content of the soil (10⁻² g g⁻¹) respectively

$C_{org}$, the organic carbon content of the soil (g kg⁻¹) and

$\rho_b$, the bulk density (g cm⁻³).

*Water uptake distribution among soil horizons*

Once tree and ground vegetation hourly transpiration has been calculated, the module sums transpiration on all trees for the

stand approach and add the ground vegetation transpiration to obtain the hourly stand transpiration, corresponding to the stand water uptake. Then, tree or stand water uptake is distributed among the horizons according to a method described in Couvreur

et al. (2012). This method assumes that water absorption occurs preferentially in horizons where the water potential (matric potential, $h$, plus a gravimetric component), $\phi$, is higher. Moreover, it considers that the amount of water uptake is proportional on the one hand to the difference between the horizon water potential and the averaged water potential weighted by the fine root proportion of the whole soil profile and on the other hand to the fine root proportion of the horizon. This can be transcribed

as:

$$UP_{root(hr)} = UP_{root}.f_{hr} + K_{comp}.3600.(\phi_{hr} - \sum_{hr=1}^{N}\phi_{hr}.f_{hr}).10.f_{hr}.(A_{pedon}\ or\ A_{stand}) \tag{71}$$

with

$UP_{root}$ and $UP_{root(hr)}$, the total water uptake and the water uptake of the $hr$ horizon respectively (l h$^{-1}$),

$f_{hr}$, the fine root proportion of the horizon $hr$,

$K_{comp}$, the compensatory conductivity set to $1.10^{-9}$ (s$^{-1}$),

$\phi_{hr}$, the horizon water potential (cm).

The right term of Eq. (71) is null when integrated on all the horizons. Then, it does not change the total amount of water uptake but it refines its distribution. Moreover, this method can generate water uplift that can occur when the top horizons are much drier than the deep ones.

*Water balance of the soil horizons*

At the tree and stand scale, the module performs an hourly water balance for each soil horizon $hr$ (numbered from the topsoil) and updates its water content ($\theta_{hr}$, m$^3$ m$^{-3}$) as follows:

$$\theta_{hr} = \theta_{hr\_prev} + \frac{(IN_{hr} - OUT_{hr})}{998 \cdot V_{hr}} \tag{72}$$

with

$\theta_{hr\_prev}$, the water content of the $hr$ horizon at the previous time step (m³ m$^{-3}$),

$V_{hr}$, the volume of the $hr$ horizon (m³),

$IN_{hr}$, the sum of the input water fluxes (l) and

$OUT_{hr}$, the sum of the output water fluxes (l).

The input fluxes are the drainage ($D$, l) and the water surplus ($S$, l) from the upper horizon ($hr-1$) and the capillary rise ($CR$, l) from the lower horizon ($hr+1$) described hereafter and represented in Fig. 1:

$$IN_{hr} = D_{hr-1} + S_{hr-1} + CR_{hr+1} \tag{73}$$

The output fluxes are the drainage, the soil evaporation ($E_{soil}$, l), the root water uptake ($UP_{root}$, l) and the capillary rise from the current horizon ($hr$) (Fig. 1):

$$OUT_{hr} = D_{hr} + EV_{soil(hr)} + UP_{root(hr)} + CR_{hr} \tag{74}$$

The water transfer ($WT$, l) between the horizon $hr$ and $hr+1$ (considered as drainage if directed downward or as capillary rise if directed upward) is estimated with the Darcy law and the average conductivity between the horizons is calculated according to the upwind scheme that takes into account the horizon water potential (e.g. An and Noh, 2014).

$$WT = \frac{K_{hr,hr+1}}{24} \cdot \left(\frac{\Delta h_m}{\Delta z} + 1\right) \cdot (A_{pedon} \; or \; A_{stand}) \cdot 100 \tag{75}$$

with

$$K_{hr,hr+1} = \begin{cases} K_{hr+1}, & \phi_{hr+1} > \phi_{hr} \\ K_{hr}, & \phi_{hr+1} \leq \phi_{hr} \end{cases} \text{(cm day}^{-1}) \tag{76}$$

$$\frac{\Delta h_m}{\Delta z} = \frac{|h_{hr+1}| - |h_{hr}|}{\frac{th_{hr} + th_{hr+1}}{2} \cdot 100} \tag{77}$$

where $th$ (m) is the horizon thickness.

To ensure the mass conservation, a variable time step ($\Delta t$, s) is considered based on a stability criterion derived from the Peclet number.

$$\Delta t = \frac{\theta_{hr_{prev}} \cdot th_{hr}}{10 \cdot \frac{K_{hr}}{100 \cdot 24 \cdot 3600}} \tag{78}$$

This criterion is calculated for each horizon and the minimum value is retained. Still, the mass conservation is tested for the whole soil profile at the end of each hour. If the water balance error exceeds 0.01 mm, the time step is divided by 10 (with 1000 as a maximum). The hourly water transfer is then obtained by cumulating the discretized values of water transfer.

For the top horizon, $D_{hr-1}$ is initialized at $TF + SF$ and $CR_{hr}$ is set to 0. For the current horizon, if $WT \geq 0$, $D_{hr} = WT$, else $D_{hr} = 0$ and $CR_{hr+1} = -WT$.

Soil evaporation occurs only in organic horizons. The amount of water evaporated from the horizon $hr$ ($EV_{soil(hr)}$, l) is obtained by taking the minimum value between the remaining water to evaporate ($remEV_{soil(hr)}$, l) and the volume of extractable water in the horizon ($VEW_{hr} = EW_{hr} \cdot (A_{pedon} \; or \; A_{stand})$, l). For the upper organic horizon, $remEV_{soil(hr)}$ is initialized to the total amount of water evaporated from the soil and is progressively decremented by subtracting $EV_{soil(hr)}$ for the deeper organic horizons:

$$remEV_{soil(hr)} = remEV_{soil(hr-1)} - EV_{soil(hr-1)} \tag{79}$$

In both mineral and organic horizons, if the water balance leads to a soil horizon water content higher than saturation, the soil horizon water content is set to the value of the saturated water content and a surplus is calculated. Part of this surplus is passed to the next horizon ($S_{hr-1}$) while the rest is considered as preferential flows and is added to the deep drainage ($DD$).

$$S_{hr-1} = IN_{hr} - \left(\theta_{s\_hr} - \theta_{hr\_prev}\right) \cdot V_{hr} \cdot 998 \cdot (1 - v_{hr}) - OUT \tag{80}$$

with

$v_{hr}$, the additional coarse fraction of the horizon (m³ m⁻³), not accounted for in the bulk density.

The deep drainage is calculated as the sum of $D_{hr}$ and $S_{hr-1}$ of the last horizon plus the preferential flows.

Before passing to the next horizon, $D_{hr-1}$ takes the value of $D_{hr}$ and $CR_{hr}$ the value of $CR_{hr+1}$.

*Absolute and relative extractable water*

The absolute extractable water ($EW$, mm) is defined as the amount of water stored in the soil that can be used by the plants:

$$EW = \sum_{hr=1}^{n}\left(\theta_{hr} - \theta_{\mathrm{wp\_}hr}\right) \cdot th_{hr} \cdot (1 - v_{hr}) \qquad (81)$$

where $\theta_{wp\_hr}$ is the water content of the soil horizon at the wilting point (m³ m⁻³).

The relative extractable water ($REW$, mm) corresponds to the ratio between this value of extractable water and the reference extractable water at the field capacity ($EW_{ref}$, mm):

$$REW = \frac{EW}{EW_{ref}} \qquad (82)$$

with

$$EW_{ref} = \sum_{hr=1}^{n}\left(\theta_{\mathrm{fc\_}hr} - \theta_{\mathrm{wp\_}hr}\right) \cdot th_{hr} \cdot (1 - v_{hr}) \qquad (83)$$

where $\theta_{fc\_hr}$ is the water content of the soil horizon at the field capacity (m³ m⁻³).

## 2.2 Parameter determination

Most of the model parameters were taken directly from the literature. In addition, an adjustment of some relationships was conducted using available data, which are described hereafter but no overall calibration of the model was performed. The model parameters for sessile oak and European beech are presented in Table 1. Regarding common hornbeam, less information is available. For this tree species, we used specific parameters for light interception, photosynthesis, respiration and carbon allocation but the same parameters as European beech for water balance and phenology given their similar morphology.

For the hydrological module, the parameters of the Eq. (54) determining the stomatal conductance were determined based on data from Jonard et al. (2011) using a non-linear fitting procedure.

For the soil hydraulic properties, the saturation $\theta_s$ was based on the 0.999 quantile of measured soil water contents (see Sect. 2.4 for more details). For horizon without soil water content sensor, $\theta_s$ was interpolated from the closest horizons. Then, the wilting point water content was determined using the obtained saturated water content and the Eq. (64) with a matric potential, $h$, of 15000 cm.

The parameters of the phenological module used to calculate the start of budburst were determined using observations from the Pan European Phenology dataset (PEP725) which provides data about phenological observations across different European countries, though not in Belgium. We selected 129 sites on the western border of Germany covering the latitudes of our 6 study plots (49.5-51.0°N), for which the budburst dates of a representative tree were available at least between 1951 and 2015. The daily minimum, maximum and mean temperatures required to achieve the calibration came from the meteorological stations of the DWD Climate Data Center (Deutscher Wetterdienst). Phenological data from each site were assigned to the nearest meteorological station (5 different stations were sufficient). The calibration was carried out with the Phenological Modeling Platform software (Chuine et al., 2013). This module enables the user to perform a Bayesian calibration procedure

using the algorithm of Metropolis et al. (1953). Some of the parameters can also be fixed. In our case, the chilling starting date of the uniChill and sequential models were fixed to the 1[st] of November of the previous year (e.g., Chiang and Brown, 2007; Roberts et al., 2015) in order to enhance the accuracy of the other parameter calibration. The length of the budburst period (necessary to determine the budburst shift), the leaf development, yellowing and falling rates were all adjusted from phenological observations conducted in our study sites on 20 trees.

## 2.3 Site description

Six sessile oak (*Quercus petraea* (Matt.) Liebl.) and European beech (*Fagus sylvatica* L.) stands located in Wallonia (Belgium) were used to evaluate the model. They all belong to long-term ecological research sites (Belgium LTER network). Three of them were located in Baileux and were monitored since 2001. The three other stands were part of the level II plot network of ICP Forests since 1998 and were located in Louvain-la-Neuve, Chimay and Virton. These sites were selected as their contrasted stand structure, species composition, soil and climate make them suitable for testing the ability of the model to account for structure complexity in various ecological conditions (at the regional scale).

### 2.3.1 Stand characteristics

The experimental site of Baileux was installed to study the impact of species mixture on forest ecosystem functioning (Jonard et al., 2006, 2007, 2008; André et al., 2008a, 2008b) and consisted of three plots. Two plots were located in stands dominated either by sessile oak or by European beech and the third one presents a mixture of both species. In these plots, sessile oak trees originated from a massive regeneration in 1880 and displayed the typical Gaussian distribution of even-aged stands, while European beech trees appeared progressively giving rise to an uneven-aged structure with all diameter classes represented. The stand in Chimay was an ancient coppice-with-standards, presently composed of mature sessile oak trees with an important hornbeam understorey. The stands in Louvain-la-Neuve and Virton were both more or less even-aged stands dominated by European beech but differed in their age, with much older trees in Louvain-la-Neuve than in Virton (130 *vs* 60 years old in 2009). All stand characteristics are provided in Table 2.

### 2.3.2 Soil properties

The Baileux, Chimay and Virton stands were all located on Cambisol but with some nuances, ranging from Dystric to the Calcaric variants in Chimay and Virton, respectively, while an Abruptic Luvisol was found in Louvain-la-Neuve (FAO soil taxonomy). All sites presented a moder humus, except Virton for which mull was observed. In Baileux, Chimay and Louvain-la-Neuve, the soil developed from the parent bedrock mixed with aeolien loess deposition that occurred at the interglacial period. In Virton, the soil originated only from the bedrock weathering. The parent materials were sandstone and shales, clayey sandstone and hard limestone bedrocks in Baileux, Chimay and Virton, respectively. In Louvain-la-Neuve, the soil was almost

exclusively built from the loess deposits. These differences in parent material generated contrasted physical and chemical soil properties (Table 3).

The soil textures also varied significantly among sites. Based on the USDA taxonomy, the soil texture was silty clayey loam and silty loam in Baileux and Louvain-la-Neuve, respectively. In Chimay and Virton, finer soil textures were observed with a clayey loam and a clay texture, respectively. In relation to the texture, drainage was good in Baileux and Louvain-la-Neuve, while the presence of inflating clay triggered the appearance of a shallow water table during the wet period and drought cracks during summer in Chimay. In Virton, despite the high clay content in the lower horizons, drainage was good due to the existence of faults in the bedrock (Table 3).

Finally, stoniness and drainage influenced the estimate of the maximum extractable water reserve. While the beech-dominated and mixed stands in Baileux and in Virton showed the lowest water reserve, the highest value was found in Louvain-la-Neuve, with intermediate values for the oak stand in Baileux and in Chimay (Table 3).

### 2.3.3 Climate

Even if the same type of climate occurred all over Belgium (temperate oceanic), the study sites were located in different bioclimatic zones (Van der Perre et al., 2015). Louvain-la-Neuve was in the *Hesbino-brabançon* zone with the highest average temperatures (11.0°C) between 2001 and 2016 and the driest conditions (818 mm). Despite their close locations, Baileux and Chimay were part of different zones. Baileux was in "*Basse et moyenne Ardenne*" while Chimay was in "*Fagne, Famenne et Calestienne*". Average temperatures are similar for both locations (i.e., 9.8°C in Baileux and 9.7°C in Chimay). Yet, a consistent difference in terms of precipitation is observed. Baileux being more elevated, it receives on average 1075 mm of precipitation each year while only 940 mm are measured in Chimay with respect to the rainfall-altitude relationship (Poncelet, 1956). Finally, Virton was part of the "*Basse Lorraine*" with elevated annual rainfall (1060 mm) and intermediate average temperature values (9.9°C) (Table 3).

For Chimay, Louvain-la-Neuve and Virton, we used data from the meteorological stations of the PAMESEB network. The records covered the 1999-2018 period. A tipping bucket located at 1 m height was used to monitor rainfall. Global radiation was registered with a pyranometer, air temperature with a resistance sensor thermistor, relative humidity with a psychrometer and wind speed with an anemometer. All these devices were placed at 1.5 m height. Data were collected at 12 min intervals and were then averaged hourly. For Baileux, an independent meteorological station managed by our laboratory was used to collect meteorological data since 2002. The devices were identical to those described before. Air temperature, relative humidity and rainfall were monitored at 1.5 m. Wind speed and global radiation were taken at 2.5 m above the ground.

### 2.4 Model evaluation

The various routines to calculate the budburst starting date were tested and the Sequential model was retained for the evaluation of the water balance module as this approach performed better (see Sect. 3.1.1).

*Phenology*

The phenological observations available on the level II sites of Chimay, Louvain-la-Neuve and Virton were used to evaluate the model predictions. These phenological observations were carried out on 20 dominant and co-dominant sessile oaks in Chimay (2012-2014) and 20 dominant and co-dominant European beeches in Louvain-la-Neuve and Virton (2012-2016) according to the ICP Forests manual (Beuker et al.*,* 2016). They consisted of weekly observations of the percentage of budburst, yellowing and leaf fall depending on the season. As the model predicted the budburst for an average tree, we evaluated it with the budburst observations of the median tree. In addition, we visually assessed the agreement between the predicted and observed increase in leaf biomass proportion (*leafProp*) during the leaf development period and between the predicted and observed decrease in green leaf proportion (*greenProp*) and in *leafProp* during leaf yellowing and leaf fall, respectively. We did not perform a statistical evaluation for these latter variables as the corresponding processes were not calibrated independently in the model. Finally, as there were no data available for trees of different social status, we could not directly evaluate the option '*Phenology at tree level*'. We evaluated however its impact on tree growth predictions for the three stands in Baileux.

*Water balance*

Regarding the water balance module, the evaluation was conducted using variables integrating most of the processes described in the model. The observed throughfall, extractable water dynamics, individual transpiration and deep drainage (considered in the next section) were compared to model predictions. For the evaluation of the throughfall, extractable water and drainage predictions, we used simulations carried out at the stand scale since the corresponding observations cannot be related to a particular tree. Regarding individual tree transpiration, the approaches at the two scales were compared (tree *vs* stand).

For the evaluation of throughfall predictions, only independent throughfall data collected in Chimay, Louvain-la-Neuve and Virton between 2000 and 2016 were used as the rainfall partitioning routine was calibrated based on data from the Baileux forest (André et al., 2008a, 2008b). The collecting devices consisted of three long gutters connected to plastic barrels. The throughfall volume was measured weekly based on the height of water in the barrels. A log transformation of both the observations and the predictions was necessary to remove the heteroscedasticity.

Individual tree transpiration predictions were evaluated against observations derived from sap flux measurements. These measurements were taken on 16 sessile oak and 16 European beech trees of different sizes in the three stands of Baileux between April and September 2003 (Jonard et al., 2011).

Extractable water was estimated based on Eq. (81) using soil water content measurements taken between 2005 and 2017 in Baileux and for the 2015-2018 period in the other sites. Soil water content was measured hourly using TDR inserted in some horizons (measurements at 3 to 5 different soil depths depending on the site). In order to decrease the influence of the soil disturbance due to the instrument installation, the first year of records was discarded. Indeed, Walker et al. (2004) showed that

inserting a moisture sensor in a soil disturbed its hydraulic properties and water content during at least 9 months. The electrical signal from the TDR was transformed in relative dielectric permittivity and then converted into soil volumetric water content ($m^3\ m^{-3}$) using the equation of Topp et al. (1980) for Baileux and resorting to our own calibration for the other sites (established based on gravimetric measurements of soil water content).

*Drainage*

Deep drainage can represent a large water output but is difficult to measure directly. Among the existing indirect approaches to estimate this component, we retained the mass-balance method using chloride ion ($Cl^-$) as tracer. This method has been widely used to estimate groundwater recharge (e.g. Bazuhair and Wood,1996; Ting et al., 1998; Scanlon et al., 2002) but can
be applied to assess deep percolation as well (Willis et al., 1997). It relies on the fact that $Cl^-$ is not subject to any chemical transformations in the soil and undergoes only temporary storage in soil (Öberg, 2003). The only $Cl^-$ input in our study plots comes from throughfall and stemflow and can be determined from $Cl^-$ deposition data obtained from monthly chemical analyses of throughfall and stemflow samples. For the deep drainage, which constitutes the only output, the $Cl^-$ concentration is also obtained from monthly chemical analyses of soil solution collected with zero-tension lysimeters at 1 m depth in the
three stands of Baileux between 2008 and 2016 and between 2013 and 2016 for the other sites. Deep drainage was estimated yearly by considering that the $Cl^-$ amount leaving the soil through drainage was equal to the $Cl^-$ input from throughfall and stemflow. As there is a clear annual pattern with a recharge and a discharge period in our study sites, the annual time step is therefore required to verify the hypothesis that chloride concentration in rainfall and in the soil are in a steady-state balance. Based on Eq. (84), the deep drainage flux was estimated and compared to our predictions.

$$Drainage = (Throughfall + Stemflow).\frac{[Cl]_{Throughfall-Stemflow}}{[Cl]_{Drainage}}$$ (84)

with

$[Cl]_{Throughfall-Stemflow}$, $Cl^-$ concentration in throughfall and stemflow and

$[Cl]_{Drainage}$, $Cl^-$ concentration in drainage water

*Statistical analyses*

To test the quality of the predictions, different statistical tests and indexes were used. The absolute bias, defined as the difference between the mean observation and prediction, and the relative bias, corresponding to the ratio between the absolute bias and the mean observation, were calculated to detect any over- or underestimation. To assess the precision of the predictions, the Root Mean Square Error (*RMSE*) was used and calculated as follows:

$$RMSE = \sqrt{\frac{\Sigma(obs-pred)^2}{n}}$$ (85)

with

$n$ the number of observations.

When the range of values differed considerably for one variable between the different sites, the RMSE was divided by the range, i.e. the difference between the maximum and the minimum values. This Normalised Root Mean Square Error (*NRMSE*) is much more adapted for comparisons in these situations.

The agreement between observations and predictions was also evaluated with the Pearson's correlation coefficient (*r*) and with a regression test conducted to analyse the linear relationship between observed and predicted values. As both predictions and observations are subject to uncertainties, we used Deming regression. Then, we tested whether the regression line confidence interval (95%) included the identity line. These tests were realized with the mcr package in R.

### 3 Results

### 3.1 Evaluation of model performance

### 3.1.1 Phenology

On average, the budburst was best predicted with the Sequential model (bias = 2.46 days compared with 8.23 and -5.88 days for Uniforc and Unichill, respectively). However, this option was less appropriate to capture the inter-annual variations (Pearson's $r = 0.537$) than Uniforc (Pearson's $r = 0.680$). The temporal variability was very poorly estimated with the Unichill model, which displayed an inverse trend for the ranking among years (Pearson's $r = -0.277$) (Fig. 2a). Moreover, as the Unichill model was not able to predict the end of the chilling period for some years in Louvain-la-Neuve (European beech), all results for this site were discarded. The predicted leaf development displayed a good agreement with observations (Fig. 3).

Simulated leaf yellowing and leaf fall were also evaluated by comparison with observations. While the leaf ageing threshold was taken from Dufrêne et al. (2005), the yellowing parameter determining the length of the yellowing period was adjusted with the five years of data from Chimay (sessile oak), Louvain-la-Neuve and Virton (European beech). Therefore, only the yellowing start was independently evaluated. The prediction of the start of the yellowing displayed a low absolute bias (2.7 days) and *RMSE* (7.0 days). However, a weak correlation (0.056) was found between predictions and observations (data not shown).

For the temporal dynamics of leaf yellowing and leaf fall, the agreement between model predictions and observations was just assessed visually since the parameter regulating these processes (yellowing, falling rate and falling frost amplifier) were adjusted with the same data. The overall agreement was good. The simulated decrease of green leaf proportion was similar for all sites as the photoperiod reduction is identical for each site and year (Fig. 4a, c and e). The only noticeable difference came from the yellowing starting date, which depended on air temperature. For Chimay (sessile oak), a close agreement was found between predictions and observations. For Louvain-la-Neuve (European beech), predictions were correctly centred but the predicted trend was more abrupt and the start of the decrease displayed some delay, except in 2012. For Virton (European beech), the decreasing trend was correctly displayed but the decrease start was less precise in 2016 (Fig. 4e).

Concerning the leaf fall, the temporal dynamics was correctly represented in Chimay (sessile oak). In Louvain-la-Neuve (European beech), the model predicted a slightly too slow decrease in leaf proportion in 2012 and 2015. For the other years, the observed and predicted leaf proportion matched well even if the predicted start of the fall appeared later than in the observations for some years. In Virton (European beech), the predictions were well centred with regards to the observations but the decrease in leaf proportion was a bit too fast in 2012 (Fig. 4b, d and f).

The option '*phenology at tree level*' was used to test if the agreement between predicted and observed basal area increment could be improved. With the default phenology option, HETEROFOR tended to overestimate the growth of dominant trees and underestimate that of suppressed trees (Jonard et al., accepted with major revisions, 2019). With the option '*phenology at tree level*', this bias was partially resorbed. The slope of the Deming regression went from 0.74 to 0.84 for sessile oak and

from 0.79 to 0.88 for European beech, being much closer to the identity line (Appendix 3). This was however at the price of slightly lower Pearson's correlation.

### 3.1.2 Water balance

For each site, the main water fluxes affecting the water balance were calculated daily, summed up and the annual values were averaged for the 2002-2016 period (Table 4). Depending on the site, 65 to 78% of the rainfall reached the floor as throughfall and 6 to 13% as stemflow. The remaining 16 to 22% was intercepted by the tree foliage and the bark and evaporated. Then, 31 to 45% of the water received as rainfall returned in the atmosphere through tree transpiration. The remaining 26 to 44% were lost from the ecosystem through drainage.

*Rainfall partitioning*

Rainfall partitioning was correctly reproduced by the HETEROFOR model. Across all considered sites (Virton, Chimay and Louvain-la-Neuve), the mean bias of throughfall predictions was very limited (-1.3%) and non-significant (*P* value of the paired *t*-test = 0.316). The confidence interval of the linear relationship between the logarithm of the observed and predicted throughfall contained the identity line corresponding to the perfect match (upper part of Fig. 5). The correlation between predictions and observations amounted to 0.86 and the *RMSE* to 16.62 mm which corresponded to 34.2% of the mean througfall (48.6 mm). The separate examination of the different sites revealed that throughfall in Virton (European beech) were very well predicted but that a slight underestimation of the throughfall predictions in Chimay (sessile oak) was compensating an overestimation of similar magnitude in Louvain-la-Neuve (European beech) (Appendix A).

*Transpiration*

The model well reproduced individual transpiration for sessile oak and European beech in the Baileux site (in 2003) with similar performances at the tree and stand scale (Pearson's r of 0.84 – 0.85 for sessile oak and 0.88 – 0.89 for European beech). For European beech, the tree approach corrected the slight bias observed with the stand approach due to an overestimation of high transpiration values. Regarding sessile oak, the small underestimation of transpiration remained whatever the scale considered (Fig. 6).

*Soil water content*

As the temporal variation of the extractable water was affected by all the water fluxes, it was used to check the performances of the water balance module (Fig. 7). A clear seasonal pattern appeared. At the beginning of the vegetation period, the extractable water values (*EW*) were highest. Then, tree and ground vegetation transpiration progressively depleted the water reserve which was partly recharged with rainfall events. Depending on their frequency, duration and intensity, the decline in *EW* was more or less pronounced and available water could reach levels close to zero. For all the sites, the Pearson's correlation

between observed and predicted relative extractable water ranged from 0.893 to 0.950. These high correlation values and the graph inspection show that the seasonal pattern was precisely reproduced by the HETEROFOR model. *NRMSE* values range from 10.54 to 13.96% while relative bias values were around -2 and -3% in Baileux-oak, Baileux-mixed and Chimay and close to -8 and -9 % in Baileux-beech, Louvain-la-Neuve and Virton. These higher negative bias in the latter stands originated

5    mainly from the model underestimation of the high values of *EW* (i.e. during wet periods). Despite these similar statistical results, the amount of extractable water in Virton displayed some peculiarities with regards to the other stands. Indeed, the observed *EW* levels fluctuated considerably more than in the other sites with frequent peaks both for high and low values that were not represented by the model. Finally, apart from Virton where some discrepancy between observations and predictions can be pointed out, the model quality did not decrease in Chimay or Virton during the 2018 summer that was categorized as

10   exceptionally dry by the Royal Meteorological Institute of Belgium. The comparison of the tree and stand approach in 2003 indicates that the extractable water calculated at the tree scale progressively deviated from that obtained at the stand scale during the course of the vegetation period and became slightly lower, especially in Baileux-oak and Baileux-mixed (Appendix B). On these graphs, one may notice the heterogeneity in extractable water within the various stands.

15   *Drainage*

The predicted deep drainage was compared with estimates calculated on a yearly basis using Cl as a tracer. The *RMSE* (100.6 mm) and the bias (-19.9%) were quite large but a surprisingly good correlation was found between the predicted and estimated drainage (Pearson's *r* = 0.963). Due to the systematic bias, the identity line was not within the confidence interval of the Deming regression despite a regression slope of 0.97 (lower part of Fig. 5).

## 4 Discussion

In order to predict the impact of global changes on forests, it is crucial to integrate and structure the existing knowledge in process-based models. However, this first step is not sufficient. A detailed documentation of the models as well as an evaluation of their performances are also needed in order to use them knowing exactly their strengths and limits. While most models were described in scientific articles or reports, their evaluation was often limited to one or two sites used to illustrate the model functioning and was generally based on integrative response variables such as radial tree growth (Vanclay and Skovsgaard, 1997; Schmidt et al., 2006). Yet, to provide robust predictions of tree growth under changing conditions, the model must be able to accurately reproduce not only the observed tree growth but also the intermediate processes describing resource availability (light, water and nutrient) (Soares et al., 1995). In the following section, we discuss the quality of the predictions for two main drivers of tree growth (phenology and water balance) in relation with the concepts used to describe them.

### 4.1 Phenology

The Sequential model that calculates both chilling and forcing periods was the least biased variant for predicting budburst. However, Uniforc model including only the forcing period better captured the inter-annual variability. While the bias is likely to originate from the model calibration (data used for calibration were observations from western Germany) and could be corrected, the ability of the model to predict temporal variability is more representative of its structural quality. It is common that models accounting only for the forcing period better represent the budburst temporal variability (Leinonen and Kramer, 2002; Yuan et al., 2007; Fu et al., 2014). Indeed, in areas where the chilling requirements are always met, as in Western Europe, the inclusion of chilling in models generally has a negative impact on model predictions. Consequently, we considered the Uniforc model as the most adapted to simulate budburst variability in long-term simulations. Still, given the expected rise in winter temperatures, accounting for chilling could become essential to make goods predictions (Clark et al., 2014) but would require more data for calibration. This highlights once again the importance of having several options to describe budburst. Most process-based models listed in Table 5 had however only one phenological variant except 4C. Moreover, apart from 4C that considers the opposite actions of inhibitory and promotory agent concentrations driven by the temperature and photoperiod, all the models used a classical one or two-phase approach based on air temperature sum (e.g., Sequential) or sigmoid function (e.g., Uniforc and Unichill), with an additional photoperiod effect for MAESPA and PSIM-DNDC (Table 5).

Depending on the phenological variant, HETEROFOR explained between 29 and 46% of the budburst variability and the *RMSE* amounted to 2.46 and 8.23 days for Sequential and Uniforc, respectively. Given the limited number of observations, these model performances are only indicative. By comparison, the phenological model of BALANCE explained 54 and 55% of the budburst variability and displayed a mean absolute error of 4.9 and 4.7 days for beech and oak respectively (Rötzer et al., 2004). In Fu et al. (2014), the $R^2$ obtained for budburst prediction ranged from 0.36 to 0.82 and the mean absolute error between 4.8 and 7.5 days for the sequential model.

A possible improvement of the phenological models accounting for chilling would be to integrate the photoperiod effect on budburst. Indeed, some recent studies have shown evidences that photoperiod can compensate for a lack of chilling temperature that would prevent the buds to open and for an early frost episode that would trigger budburst before winter (Vitasse and Basler, 2013; Pletsers et al., 2015). This mechanism is particularly present for late-successional species like beech and oak

trees and is regularly cited as a key element to simulate the phenology under climate change (Basler and Körner, 2012). Some models tried to account for the photoperiod effect simply by replacing chilling by photoperiod (Kramer, 1994; Schaber and Badeck, 2003) but, in this way, failed to represent the combined effect of these variables. Recently, a few models integrating the compensatory effect of photoperiod on chilling have appeared. However, these models include more phenological parameters for similar predictive ability (Gauzere et al., 2017). It remains indeed difficult to disentangle the co-varying effect

of chilling and day length with *in situ* measurements (Flynn and Wolkovich, 2018) since photoperiod variations only occur for sites with different latitudes where other confusing factors play a role as well (Primack et al., 2009). Therefore, many data is necessary to calibrate these models. Then, we decided to privilege the accuracy of our phenological model to a more process-based approach but we are looking forward for improvements in these kinds of models and a more consensual body of literature.

The better growth predictions obtained for the small trees when the phenology was calculated at the individual scale highlights the importance of the "phenological avoidance strategy" displayed by understory trees. This had already been mentioned by Lopez et al. (2008) who observed that early-leafing species received between 45 and 80% of their photon flux during the budburst period. Moreover, a simulation study showed that a one (two) week lengthening of the understory vegetation period with regards to the overstory in both spring and autumn generated a productivity increase of 32% (55%) on such a short period

(Jolly et al., 2004).

### 4.2 Water balance

In a first step, the annual water fluxes predicted by HETEROFOR were compared to measurements and predictions of other studies (Table 4). Then, some water fluxes were individually evaluated when data was available. Finally, some potential improvement of the water balance module were discussed.

Various studies were taken from the literature to compare our water module predictions with observations. They cover a range of annual rainfall comprised between 425 and 1476 mm (Table 4), which is comparable to what can be found in Belgium. The proportions of rainfall converted to stemflow obtained with HETEROFOR (6.1 to 13.1%) are within the range reported in the literature (0.6 to 20.4%). This large observation spectrum comes from the important seasonal (higher stemflow proportion in winter than during the vegetation period) and species differences (stemflow importance is higher for beech than oak trees),

which features are accounted for in HETEROFOR. However, the mean value from the literature (7.3% of rainfall) is close to the average value for the six study sites (10.3%). The proportions of intercepted rainfall (15.9 to 22.0%) and throughfall (64.8 to 78.0%) are also consistent with the ranges reported in other studies (1.9 to 31.0% and 59.8 to 83.1%). Moreover, we observed

a good matching between the average values (respectively 19.5 and 73.8% from literature and 19.4% and 70.2% for our study sites). For transpiration, the range found in the literature is large (14.8 to 52.3% for an average value of 31.9%), which is not surprising since inter-annual and inter-site variabilities are high for this variable (Schipka et al., 2005; Vincke et al., 2005). The predicted transpiration proportions are less variable (31.2 to 44.9%) and their average value of 36.0% is slightly superior

to the mean observed transpiration (31.9%). Regarding drainage, no direct measurements can be made; all the estimates from the literature come from indirect methods or modelling also subject to uncertainties. The range of drainage values reported in the literature (13 to 70%) is very large and contains that obtained with HETEROFOR (26.3 to 44.2%). The mean predicted drainage (39.7%) is close to the mean value of the literature (37.5%). By this comparison with the water fluxes reported in the literature, we show that HETEROFOR provides plausible estimates of the various components of the water cycle.

Comparing predicted and observed throughfall is interesting to evaluate the water balance module since throughfall is an integrative variable depending on the water storage capacity of foliage, on evaporation, and on the proportion of stemflow. The good agreement between observations and predictions indicates that the partitioning of rainfall when passing through the canopy and the evaporation of the water intercepted by foliage and bark are well described. Among the different models of the Table 5, no one accounts separately for stemflow and throughfall but other models not included in the list consider the two

fluxes separetly (e.g. Gotilwa+ and Castanea). Yet, separating throughfall and stemflow is important, especially for structurally-complex stands. In these stands, rainfall interception cannot be simulated based on a mean foliage storage capacity and a mean partitioning between throughfall and stemflow since these parameters vary with stand composition and structure. Our tree-level approach estimating foliage storage capacity and stemflow proportion based on individual tree characteristics allows to overcome this difficulty. Moreover, if one wants to accurately describe the nutrient cycle, partitioning rainfall is

essential as nutrient concentrations in stemflow and throughfall can be 10 to 100 times higher than in rainfall due to dry and wet deposition and canopy exchange (Levia and Herwitz, 2000; André et al., 2008c; van Stan and Gordon, 2018). Even if the rainfall partitioning can still be improved from a theoretical perspective (e.g., including canopy drainage after rain events or the impact of wind on the foliage storage capacity like in Hörmann et al. (1996) or Muzylo et al. (2009)), we chose to limit the level of complexity in order to avoid calibration difficulties.

HETEROFOR satisfyingly reproduced individual tree transpiration with similar prediction quality for the tree and stand approach regarding the water balance calculation. For European beech, the water balance calculation at the tree scale allowed even correcting the small bias which appeared with the stand approach (Fig. 6). The year selected for the simulation (2003) was particularly dry and hot during summer, which allowed to cover a large range of meteorological and soil water conditions. It is indeed interesting to test the tree approach under dry conditions since horizontal water redistribution is much less efficient

in this case.

Twenty to thirty percent of the transpiration variability remained unexplained by the model, which can be partly ascribed to model inaccuracies but also to the large uncertainty associated with the sap flux measurements. Among others, the

measurements made by Jonard et al. (2011) did not take the azimuthal variation of the sap flux into account since only one sensor per tree was installed.

This first evaluation of tree transpiration predictions indicates that no loss of precision occurred with the tree scale approach while this detailed spatial representation could have increased the variability of transpiration predictions since it generated some heterogeneity in soil water availability (Appendix B). These good results show that the water balance calculation at the tree scale provides a promising tool to better understand the individual variability and local environment effects on tree water use and sensitivity to drought. This must be considered in a dynamics of continuous improvement of the model and will require more transpiration measurements and in-depth comparisons of predictions and observations.

The amount of extractable water ($EW$), directly influenced by tree transpiration and soil evaporation, is also a key element of the water cycle, driving, among others, the drought resistance of a stand. The temporal dynamics of $EW$ was well captured by HETEROFOR as evidenced by the high correlations (Pearson's coefficient comprised between 0.893 and 0.950) between observed and predicted $EW$ for the various study sites (Fig. 7). These correlations are within the high end of the range reported for similar models. With the BALANCE model, Gröte and Pretzsch (2002) obtained a Pearson's correlation of 0.85 between the observed and predicted soil water content of the upper soil (0-20 cm horizon) in a beech forest in Germany (Freising). Applying BALANCE on three broadleaved stands of oak or beech in Germany, Rötzer et al. (2005) were also able to correctly reproduce soil water content dynamics but they mentioned a significant decrease in the quality of predictions during the 2003 drought due to an overestimation of the soil drying, which was not observed with HETEROFOR in 2018. Comparing the observed soil water content at various soil depths with that predicted by the 4C model in mixed oak and pine forest (Brandeburg, Germany), Gutsch et al. (2015) obtained Pearson's correlations ranging from 0.59 to 0.74. In an oak stand in Tennessee (USA), Hanson et al. (2004) compared the ability of nine process-based forest models to reproduce soil water dynamics in the 0-35 cm horizon of the soil and obtained correlations ranging from 0.81 to 0.96.

In the study of Hanson et al. (2004), relative bias was evaluated as well for soil water content and ranged between -1.3 and 4.0%. These values are comparable to those found in this study yet a bit lower. Furthermore, discrepancies between predicted and observed $EW$ occurred during limited periods. Several reasons can be advanced to explain them. Errors in the prediction of the budburst date can result in a too early or too late restarting of tree transpiration and induce an inaccurate depletion of the soil extractable water during the vegetation period. In order to distinguish this error source from the others, one could force the model with the observed budburst date. This option is however not yet implemented in the model. The lack of agreement between observed and predicted $EW$ could also be ascribed to the strong heterogeneity of soil properties in forest ecosystems. Similarly, local rainfall events recharging soil extractable water during summer (often associated with thunderstorms) are sometimes not correctly taken into account when missing meteorological data (due to failed sensors or other technical problems) are replaced by rainfall data of a meteorological station further away.

Simplifications and errors in the model conception may also generate divergence between observations and predictions. However, this structural uncertainty can be limited by selecting the most appropriate concepts. HETEROFOR predicts water

transfer between soil horizons using the Darcy law. We tried to implement an approach of intermediate complexity between simple bucket models and the Richards equations. From a theoretical point of view, the Richards approach is the most state-of-the-art but requires very long calculation times (Fatichi et al., 2016) and is usually implemented in models specifically dedicated to water flow simulations (in Table 5, only one of the models, MAESPA, use them). Forest ecosystem models

generally use simpler approaches such as the bucket model declined in a large variety of forms (Table 5). These models consider one or several buckets with a specified water storage capacity that is filled with rainfall and is emptied by evapotranspiration. If the soil water content is at field capacity, water is transferred to the underlying layer and finally lost by drainage. Improved versions can account for transfer between buckets in unsaturated conditions using the Darcy law (leaky bucket model).

Our water transfer routine discretises the soil in horizons whose thickness varies from a few centimetres (upper horizons) to half a meter (deeper horizons). Compared to the numerical resolution of Richards equation which requires thin soil layers (1 to 2 cm), our vertical discretisation of the soil profile is quite coarse and inaccurately predict the advance of the wetting front. As the tree transpiration and photosynthesis depend on the soil water conditions of the whole soil profile, this inaccuracy has very limited implications on the simulated tree growth. In our approach, water transfer during a time step is calculated based

on the horizon water potentials estimated at the end of the previous time step. As such, the model makes the hypothesis that the water content does not change significantly during the time step, which is certainly not the case close to the wetting front and cannot ensure mass conservation. In order to limit this problem, the model uses an adaptive time step estimated based on the Peclet number described in Eq. (78). This allows to ensure mass conservation.

Finally, another reason that could explain the discrepancy between predictions and observations is the presence of macropores

that cause preferential flows. These water fluxes defined as water movements in the soil along preferred pathways that bypass the soil matrix (Hardie et al., 2011) can be generated by soil shrinkage, root growth, chemical weathering, cycles of freezing and thawing or bioturbation (Aubertin, 1971). These macropores are more frequent in forest soils than in agricultural soils as the latter are often ploughed and homogenized. They are however difficult to characterize given their strong spatial heterogeneity in both vertical and horizontal directions (Aubertin, 1971). Adaptations of the Richards equations can be used

to account for the preferential flows (dual porosity and dual permeability) but require a good characterisation of soil macropores (not possible to achieve routinely in forest soils given their heterogeneity) and are still more complicated to solve than the classical Richards equations. We implemented in the model the transfer of the soil water surplus (when water saturation is reached) to the underlying horizon and the possibility to redirect part of this surplus as deep drainage to account empirically for preferential flows. Indeed, preferential flows in macropores become significant only when rainfall exceeds the

water infiltration rate in the soil matrix and accumulates in the soil surface. The fraction of the water surplus considered as preferential flows is an empirical parameter reflecting the macroporosity of the site.

The performances of the soil water transfer routine can also be checked based on the deep drainage flux. In this study, we compared the deep drainage estimated with HETEROFOR and with the chloride mass balance approach. The mean drainage

predicted with HETEROFOR was 379 mm per year while the average drainage obtained with the chloride approach amounted to 472 mm per year, which corresponds to a bias of -19.9%. The correlation between the two types of estimate amounted to a Pearson's coefficient of 0.963, with a *RMSE* value of 100.6 mm. These values depict a constant negative bias in the predictions that can easily be seen on the lower part of Fig. 5. It is hard to tell whether the gap originates from the model or the method

used to estimate drainage from the chloride approach. It is more likely that the bias must be ascribed to both. Indeed, on the one hand, even if the use of chemical tracers to estimate drainage or groundwater recharge is commonly used (Scanlon et al., 2002), its application remains subject to uncertainties. First, the chloride method supposes that the main chloride source is rainfall and that the other sources can be neglected (Murphy et al., 1996). This hypothesis is not always fulfilled due to anthropogenic chloride introduction (road salting, wastewater) or when chloride is present in the bedrock (Ping et al., 2014).

Then, preferential flows have been regularly highlighted as an error source since the associated water fluxes are not well sampled by zero-tension lysimeters (Tyler and Walker, 1994; Nkotagu, 1996). Finally, this method displays better results when rainfall and soil water is richer in chloride (e.g., sites close to the sea with high marine deposits or with low drainage flux) because the chemical analyses are more accurate for higher concentrations (Sammis et al., 1982; Grismer et al., 2000). On the other hand, modelling errors could explain the bias presence. One of them could be the overestimation of the transpired

water amount. However, deep drainage tends to produce during winter while transpiration only takes place during the vegetation period (spring and summer). Therefore, if transpiration was overestimated we should observe an underestimation of the EW during spring and summer (low values), which is not the case (Fig. 7).

Hanson et al. (2004) measured deep drainage at the watershed level by accounting for rainfall and stream flow outputs and compared their measurements with the predictions of several models. Their multi-model comparison displayed similar *RMSE*

(65.5 to 225.6 mm) and relative bias (-27.6 to 20.5 %) values but the Pearson's coefficient displayed by HETEROFOR is definitely located in the high tail of the study range (0.61 to 0.95). However, the performances of their models are not strictly comparable to ours since the reference method for estimating drainage differs (Sammis et al., 1982; Grismer et al., 2000; Obiefuna and Orazulike, 2011).

### 4.3 Simulating phenology and water balance in heterogeneous stands

Increasing the functional trait diversity and promoting uneven-aged stands are among the management strategies that foresters can use to make their forests more resistant to stressful conditions and more resilient after a disturbance (Pedro et al., 2015; Jactel et al., 2017; Anderegg et al., 2018). With the growing interest for mixed and uneven-aged stands, various attempts have been made to better account for stand structure in process-based forest models. Some of these models present very detailed 3D representations of individual tree structure but describe generally only specific physiological processes (e.g. LIGNUM,

EMILION, MAESPA). Such models are very useful tools for analysing outcomes of eco-physiological experiments and obtain a better understanding of specific eco-physiological processes (e.g. drought sensitivity) in structurally complex stands. Since they are generally computationally expensive, applied to one or a limited number of individuals and do not account for tree

dimensional growth, they can however not be used for simulating long-term forest dynamics according to various climate and forest management scenarios. Other individual-based models can be applied on all the trees of a stand in long-term simulations but at the cost of a coarse representation of physiological processes (e.g. SORTIE/BC). These models are interesting to analyse tree growth dynamics in heterogeneous forests but are less suitable for taking into account the changing environment. Since

they simplify stand structure representation, cohort-based models can afford a detailed process-based description of the main processes involved in tree growth (e.g. 4C, ANAFORE, PSIM-DNDC, 3D-CMCC, see Table 5 for model characteristics). Here, the compromise is made on the spatial representation which accounts for the vertical gradient in growing conditions but not for the horizontal heterogeneity. Such models can indeed not distinguish between stands composed of the same trees but with various degrees of spatial aggregation (e.g. intimate *vs* patch-wise mixture). Similarly, some individual-based models

choose to sacrifice the horizontal heterogeneity of some processes (e.g. iLand and Hybrid that calculate most of the water balance at stand scale, see Table 5 for model characteristics).

To simulate the impact of management in heterogeneous forests under changing conditions, we developed a spatially explicit individual-based approach designed to account for individual variability, local environment and adaptive behaviour of trees (Berger et al., 2008). The compromise was not achieved by strongly reducing the complexity of a particular aspect (spatial

representation, process description or spatial or temporal coverage) but instead we tried to develop a balanced approach in which each aspect is described with the same level of complexity.

Among the existing individual-based models, BALANCE and NOTG-3D are close to HETEROFOR since they were designed according to the same philosophy. They present however some substantial conceptual differences (Table 5). Except BALANCE for leaf yellowing, HETEROFOR is the only model determining budburst, leaf yellowing and fall at the tree level.

While rainfall partitioning is only calculated in HETEROFOR, the spatial representation of local climate conditions in the canopy is finer in BALANCE and NOTG-3D that consider different canopy layers or voxels. Regarding transpiration, HETEROFOR and BALANCE implement the widely used Penman-Monteith equation while it is determined as part of detailed energy budget in NOTG-3D. Finally, they all describe soil water dynamics at the individual scale but HETEROFOR displays a more mechanistic approach for describing soil water transfer among horizons (bucket *vs* Darcy model).

In HETEOFOR, some processes were described at two spatial scales (tree or stand level) in order to have the opportunity to compare the two approaches and choose the most appropriate one depending on the pursued objective. The phenological timing is species dependent and can optionally vary with tree size. This option (phenology at tree level) is very interesting since it accounts for both the ontogeny effect and the vertical gradient in climate conditions. With this option, a longer vegetation period is assigned to the understory compared to the overstory, which allows improving radial growth predictions by correcting

the growth underestimation in small trees and the overestimation in bigger ones (Appendix C). This first attempt to describe phenology at tree scale is quite empirical and could be adapted in the future as knowledge on inter-individual phenology differences improves. Individual phenology observations for trees of all social status will be necessary to better calibrate and evaluate this module in an iterative cycle of model improvement.

For the water balance, HETEROFOR accounts for a direct tree size effect on stomatal conductance (stomatal conductance is inversely proportional to the height of largest crow extension) and for an indirect effect on the sapwood to leaf area ratio whose both components depends on tree size (Jonard et al., accepted with major revisions, 2019). In addition, individual transpiration is a function of the radiation intercepted by the tree, the local wind speed and of the soil water availability. Finally, the tree

adaptive behaviour to the local environment is described by an adaptation of the foliage biomass to local competition conditions and by specific leaf area varying with crown position within the canopy (Jonard et al., accepted with major revisions, 2019). Whatever the considered scale (tree or stand), HETEROFOR was able to correctly reproduce individual tree transpiration. Additional sap flux measurements as well as a characterization of the horizontal soil water content heterogeneity (using GPR technique for example) would be very useful to further evaluate the model performances and still enhance its ability to describe

the complex hydrological functioning of heterogeneous forest. Among the possible improvements, mortality representation could be enhanced by considering hydraulic failure during severe droughts (Martin-StPaul et al., 2017). Another model improvement would be to take the interaction between the water cycle and the phenology into account by integrating a drought effect on budburst, leaf yellowing and fall as reported in some observation studies (Sanz-Perez and Castro-Diez, 2010; Xie et al., 2018).

**5 Conclusion**

In this paper, two key modules of HETEROFOR are described in details and evaluated in 4 sites / 6 stands. The phenological module correctly predicts the leafed period, which is essential to simulate light interception by trees, evapotranspiration, photosynthesis and respiration. With the hydrological module, HETEROFOR properly estimates rainfall interception, individual transpiration, soil water and deep drainage. Reproducing correctly the soil water dynamics is necessary to adequately

predict photosynthesis since stomatal conductance closely depends on it. In addition, the description of the nutrient cycling requires accurate estimates of the water fluxes since water is the main vehicle for nutrient transport.

Our spatially explicit individual-based approach allows describing phenology and water balance in structurally-complex stands by partly accounting for the tree size effect on phenology and on tree transpiration, for the local environment modification (radiation and water availability) and for the adaptive behaviour of trees to local conditions (e.g. tree leaf area). Given the

complexity of the functioning of heterogeneous forests, there are still a lot of ways to explore to improve the model, which will be done progressively as part of an iterative approach based on the prediction comparison with observations. Our model will also be used to compare various spatial representation scales (tree, cohort, stand) and determines the most appropriate one depending on the considered process and the pursued objective.

Simulating properly resource availability is necessary to produce robust predictions of tree growth under changing climate

conditions. The next steps will be to extend the model validation to other European sites to cover a larger range of ecological conditions and to use HETEROFOR to simulate stands dynamics under various management options and climate scenarios.

**Table 1: Description of the different module parameters for sessile oak and European beech and origin of their value**

| Symbol | Description | Units | Value Sessile oak | European beech | Origin |
|---|---|---|---|---|---|
| **Storage capacity** | | | | | |
| $c_{foliage\_sp}$ | foliage storage capacity | l per m² of leaf | 0.272 | 0.174 | André et al. (2008b) |
| $c_{bark\_sp\_ll}$ | bark storage capacity c parameter (leafless) | l mm$^{-1}$ | -9.08 | -9.53 | André et al. (2008b) |
| $d_{bark\_sp\_ll}$ | bark storage capacity d parameter (leafless) | l cm$^{-1}$ mm$^{-1}$ | 0.16 | 0.18 | André et al. (2008b) |
| $R_{min\_sp\_ll}$ | stemflow rainfall threshold (leafless) | mm | 6 | 1.5 | André et al. (2008b) |
| $c_{bark\_sp\_ld}$ | bark storage capacity c parameter (leaved) | l mm$^{-1}$ | -4.21 | -4.15 | André et al. (2008b) |
| $d_{bark\_sp\_ll}$ | bark storage capacity d parameter (leaved) | l cm$^{-1}$ mm$^{-1}$ | 0.08 | 0.09 | André et al. (2008b) |
| $R_{min\_sp\_ld}$ | stemflow rainfall threshold (leaved) | mm | 10.9 | 3.4 | André et al. (2008b) |
| **Evaporation of water on foliage and trunk** | | | | | |
| $l_{sp}$ | mean leaf width of the species sp | m | 0.08 | 0.07 | measured |
| $g_{s\_bark\_min}$ | bark minimum vapour conductance | m.s$^{-1}$ | 0.0077519 | | soil values x 100 |
| $g_{s\_bark\_max}$ | bark maximum vapour conductance | m.s$^{-1}$ | 0.125 | | soil values x 100 |
| **Tree transpiration** | | | | | |
| $g_{s0\_foliage}$ | reference stomatal conductance | m s$^{-1}$ | 308.4 | 281.9 | calibrated based on Jonard et al. (2011) |
| $p_{radiation}$ | parameter of the stomatal response to radiation | W m$^{-2}$ | 37.2 | | calibrated based on Jonard et al. (2011) |
| $p1_{sw}$ | parameter 1 of the stomatal response to soil water potential | adimensional | 0.127 | 0.527 | calibrated based on Jonard et al. (2011) |
| $p2_{sw}$ | parameter 2 of the stomatal response to soil water potential | adimensional | 5 | 3 | calibrated based on Jonard et al. (2011) |
| $p_{rew\_sensitivity}$ | parameter of the stomatal response to vapour pressure deficit | adimensional | -11.1 | -2.15 | calibrated based on Jonard et al. (2011) |
| **Soil evaporation** | | | | | |
| $k$ | extinction coefficient | adimensional | 0.5 | | Teh (2006) |
| $g_{s\_soil\_min}$ | soil minimum vapour conductance | m s$^{-1}$ | 7.75E-05 | | Dufrêne (2005) |
| $g_{s\_soil\_max}$ | soil maximum vapour conductance | m s$^{-1}$ | 0.00125 | | Dufrêne (2005) |
| **Phenology** | | | | | |
| *Unichill* | | | | | |
| $t_0$ | chilling starting date | day of year | 305 (1st of November) | | Chuine (2000) |
| $C_a, C_b, C_c$ | chilling parameters | adimensional | 0.37, -6.48, -7.91 | 1.17, -29.21, -13.51 | calibrated |
| $C^*$ | chilling threshold | °C | 132.82 | 153.80 | calibrated |
| $F_b, F_c$ | forcing parameters | adimensional | 0.23, 13.17 | 0.19, 15.58 | calibrated |
| $F^*$ | forcing threshold | °C | 9.72 | 4.77 | calibrated |
| *Uniforc* | | | | | |
| $t_1$ | forcing starting date | day of year | 57 (26th of Feb) | 44 (13th th of Feb) | calibrated |
| $F_b, F_c$ | forcing parameters | adimensional | -0.12, 18.28 | -0.08, 11.77 | calibrated |
| $F^*$ | forcing threshold | °C | 12.88 | 28.12 | calibrated |
| *Sequential* | | | | | |
| $t_0$ | chilling starting date | day of year | 305 (1st of November) | | Chuine (2000) |
| $T_{min}, T_{max}, T_{opt}$ | minimal, maximal and optimal chilling temperatures | °C | -35.08, 41.61, 0.26 | -9.89, 42.87, 28.5 | calibrated |
| $C^*$ | chilling threshold | °C | 50.25 | 3.40 | calibrated |
| $F_a, F_b, F_c$ | forcing parameters | adimensional | 1.0, 0.07, 11.23 | 1.0, 0.05, -1.43 | calibrated |
| $F^*$ | forcing threshold | °C | 46.72 | 94.18 | calibrated |
| $t2a\_shift$ | budburst shift | Days | 12.0 | 15.0 | calibrated |
| *Other phases* | | | | | |
| $LD^*$ | leaf development threshold | °C | 260.0 | 312.0 | calibrated |
| $t_3$ | ageing starting date | day of year | 213 (1st of August) | | Dufrêne et al. (2005) |
| $T_{b\_age}$ | base temperature for ageing | °C | 20.0 | | Dufrêne et al. (2005) |
| $A^*$ | ageing threshold | °C | 230.0 | | Dufrêne et al. (2005) |
| $y$ | leaf yellowing parameter | adimensional | 0.07557 | 0.1384 | calibrated |
| $Y^*$ | yellowing threshold | °C | 0.01 | | fixed |
| $R_{fall}$ | falling rate | s m$^{-1}$ d$^{-1}$ | 0.010 | 0.007 | calibrated |
| $F_{ampl}$ | frost amplifier coefficient | adimensional | 3.0 | 2.0 | calibrated |

**Table 2: Initial stand characteristics for the main tree species and for the whole stands**

| Stand Inventory year | Species | Tree density (N/ha) | Basal Area (m²/ha) | $C_{130}$ (cm) | Dominant Height (m) | LAI (m²/m²) |
|---|---|---|---|---|---|---|
| Baileux (oak) 2001 | Sessile oak | 187 | 16.2 | 100.6 (26.5) | 21.9 | |
| | European beech | 118 | 4.0 | 46.4 (35.6) | 15.5 | |
| | Common hornbeam | 152 | 1.3 | 31.4 (11.4) | 11.6 | |
| | Total | 468 | 21.6 | 63.7 (40.4) | 22.2 | 4.17 |
| Baileux (beech) 2001 | Sessile oak | 72 | 6.4 | 103.3 (18.1) | 23.0 | |
| | European beech | 217 | 16.5 | 87.5 (41.5) | 25.0 | |
| | Total | 297 | 23.1 | 90.3 (38.5) | 24.8 | 4.86 |
| Baileux (mixed) 2001 | Sessile oak | 118 | 12.9 | 115.5 (21.0) | 24.5 | |
| | European beech | 352 | 17.0 | 91.2 (39.3) | 25.7 | |
| | Common hornbeam | 9 | 0.1 | 22.6 (17.3) | 9.4 | |
| | Total | 484 | 30.0 | 101.2 (42.0) | 25.9 | 5.99 |
| Chimay 1999 | Sessile oak | 63 | 13.1 | 158.7 (35.0) | 20.4 | |
| | Common hornbeam | 634 | 5.3 | 30.5 (10.8) | 15.8 | |
| | Total | 697 | 18.4 | 42.4 (40.1) | 19.2 | 3.96 |
| Louvain-la-Neuve 1999 | Sessile oak | 21 | 4.7 | 165.9 (23.0) | 30.9 | |
| | European beech | 87 | 24.6 | 179.1 (53.6) | 32.1 | |
| | Total | 108 | 29.4 | 176.6 (49.6) | 32.9 | 6.34 |
| Virton 1999 | Sessile oak | 5 | 1.3 | 190.0 (10.0) | 24.1 | |
| | European beech | 340 | 16.8 | 70.9 (31.7) | 24.0 | |
| | Common hornbeam | 22 | 0.4 | 48.4 (15.4) | 14.5 | |
| | Total | 425 | 23.3 | 73.6 (36.0) | 24.0 | 6.93 |

**Table 3: Soil and meteorological characteristics of the different study sites (2001-2016 period)**

| Stand | Location | Altitude (m) | Soil type | Soil texture (USDA) | Max extractable water (mm) | Annual rainfall (mm) | Mean air temperature (°C) |
|---|---|---|---|---|---|---|---|
| Baileux (beech/mixed/oak) | 50°01'N, 4°24'E | 305-312 | Cambisol | Silt (clay) loam | 178/154/239 | 1075 | 9.8 |
| Chimay | 50°06'N, 4°16'E | 260 | Dystric Cambisol | Clay loam | 205 | 940 | 9.7 |
| Louvain-la-Neuve | 50°41'N, 4°36'E | 130 | Abruptic Luvisol | Silt loam | 450 | 818 | 11.0 |
| Virton | 49°31'N, 5°34'E | 370 | Calcaric Cambisol | Clay | 167 | 1060 | 9.9 |

**Table 4: Predicted annual water fluxes and the corresponding percentage of rainfall in brackets for the different study sites during the period 2002-2016. The minimum, maximum and mean values from literature are indicated with the number of studies (n) they are based on. The studies taken into account were restricted to sites dominated by beech or by oak in temperate regions with similar meteorological conditions. Data from the same site were averaged so that long monitoring studies do not influence too much the average value.**

| Site/Study | Rainfall (mm) | Stemflow (mm) (%R) | Throughfall (mm) (%R) | Interception (mm) (%R) | Transpiration (mm) (%R) | Drainage (mm) (%R) |
|---|---|---|---|---|---|---|
| Baileux-beech | 1059 | 124 (11.7) | 728 (68.7) | 207 (19.5) | 366 (34.5) | 428 (40.4) |
| Baileux-mixed | 1059 | 139 (13.1) | 686 (64.8) | 233 (22.0) | 331 (31.2) | 432 (40.8) |
| Baileux-oak | 1059 | 94 (8.9) | 763 (72.0) | 202 (19.1) | 343 (32.4) | 465 (43.9) |
| Chimay | 897 | 55 (6.1) | 700 (78.0) | 143 (15.9) | 351 (38.7) | 384 (42.3) |
| Louvain-la-Neuve | 800 | 81 (10.1) | 545 (68.1) | 174 (21.8) | 353 (44.9) | 206 (26.3) |
| Virton | 1014 | 123 (12.1) | 705 (69.5) | 186 (18.3) | 361 (34.4) | 464 (44.2) |
| Van der Salm et al. (2004) - oak | 725 | - | - | 177 (24.4) | 338 (46.6) | 123 (17.0) |
| Van der Salm et al. (2004) - beech | 891 | - | - | 241 (27.0) | 356 (40.0) | 138 (15.5) |
| Min literature value | 425 | 5.0 (0.6) | 209.9 (59.8) | 19.0 (1.9) | 117.5 (14.8) | 82.0 (13.0) |
| Max literature value | 1476 | 162.0 (20.4) | 864.0 (83.1) | 241.0 (31.0) | 397.0 (52.3) | 626.0 (70.0) |
| Mean literature value | 805.2 | 44.3 (7.3) | 514.6 (73.8) | 109.2 (19.5) | 263.5 (31.9) | 312.1 (37.5) |
| n | | 9 (20) | 13 (23) | 12 (23) | 24 (22) | 11 (13) |

5   Papers included in the literature review: Cepel, 1967. Aussenac, 1968. Aussenac, 1970. Lemée, 1974. Nagy, 1974. Szabo, 1975. Aussenac and Boulangeat, 1980. Matzner and Ulrich, 1981. Rowe, 1983. Bücking and Krebs, 1986. Gerke, 1987. Giacomin and Trucchi, 1992. Neal et al., 1993. Leuschner, 1994. Ulrich et al., 1995. Heil, 1996. Tarazona et al., 1996. Bellot and Escarre, 1998. Didon-Lescot, 1998. Herbst et al., 1998. Nizinski and Saugier, 1998. Forgeard et al., 1980. Granier et al., 2000. Bent, 2001. Michopoulos et al., 2001. Knoche et al., 2002. Mosello et al., 2002. Dripps, 2003. Bastrup-Birk and Gundersen, 2004. Hanson et al., 2004. Ladekarl et al., 2005. Schipka et al., 2005. Vincke et al., 2005. Carlyle-Moses and Price, 2006. Christiansen et al., 2006. Roberts and Rosier, 2006.Schmidt, 2007. Herbst et al., 2008. Staelens et al., 2008. Ahmadi et al., 2009. Müller and Bolte, 2009. Risser et al., 2009. Gebauer et al., 2012.

**Table 5: Comparison of the spatial scale (S=stand, C=cohort, I=individual, I\*=individual target tree) and concepts used for describing phenological and hydrological processes in HETEROFOR and in other individual and cohort-based models. Backslash is used to distinguish the various model options. Abbreviations used in for describing transpiration (P-M= Penman-Monteith, SPAC = Soil-Plant-Atmosphere Continuum)**

| Model | Spatial resolution | Phenology | | Water balance | | | |
| --- | --- | --- | --- | --- | --- | --- | --- |
| | | Budburst model | Individual variability | Rainfall partitioning | Canopy micro-climate variations | Transpiration | Soil water dynamics |
| HETEROFOR | Individual | one-phase/ two-phase (C) | Y (C) | Y (C) | wind, light (I) | P-M with modifiers (S) | Darcy model + mass conservation (S) |
| HETEROFOR - tree-scale phenology - fine resolution water | Individual | one-phase/ two-phase (C) | Y (I) | Y (I) | wind, light (I) | P-M with modifiers (I) | Darcy model + mass conservation (I) |
| BALANCE[a,b] | Individual | one-phase (C) | Y (I) (yellowing) | N | air T°, wind, light (I) | P-M with modifiers (I) | multi-layer bucket (I) |
| HYBRID[c] | Individual | parallel chilling forcing (C) | N | N | light (I) | plot conductance and energy balance (S) | single-layer bucket (S) |
| iLand[d] | Individual | two-phase (C) | N | N | light (I) | P-M with modifiers (S) | single-layer bucket (S) |
| MAESPA[e,f] | Individual | one-phase + photoperiod (C) | N | N | wind, light (I) | P-M with SPAC resistance (I\*) | Richards equation (S) |
| NOTG-3D[g] | Individual | one-phase (C) | N | N | air T°, wind, light (I) | energy balance with modifiers (I) | multi-layer bucket (I) |
| 4C[h,i] | Cohort | promot.-inhibit. and others (C) | N | N | light (C) | P-M and others with modifiers (C) | multi-layer bucket (C) |
| ANAFORE[j] | Cohort | two-phase (C) | N | N | wind, light (C) | P-M with SPAC resistance (C) | spilling multi-layer bucket (C) |
| PSIM-DNDC[k] | Cohort | one-phase + photoperiod (C) | N | N | air T°, light (C) | Carbon demand driven with modifiers (C) | Darcy model (S) |
| 3D-CMCC[l,m] | Cohort | one-phase (C) | N | N | light (C) | P-M lookalike function with modifiers (C) | single-layer bucket (S) |

**a**. Grote and Pretzsch, 2002 **b**. Rötzer et al., 2010 **c**. Friend et al., 1997 **d**. Seidl et al., 2012 **e**. Duursma and Medlyn, 2012 **f**. Duursma, 2008 **g**. Simioni et al., 2016

**h**. Gutsch et al., 2015 **i**. Model description on 4C website **j**. Deckmyn et al., 2008 **k**.Grote et al., 2011 **l**. Collalti et al., 2014 **m**. Collalti et al., 2016

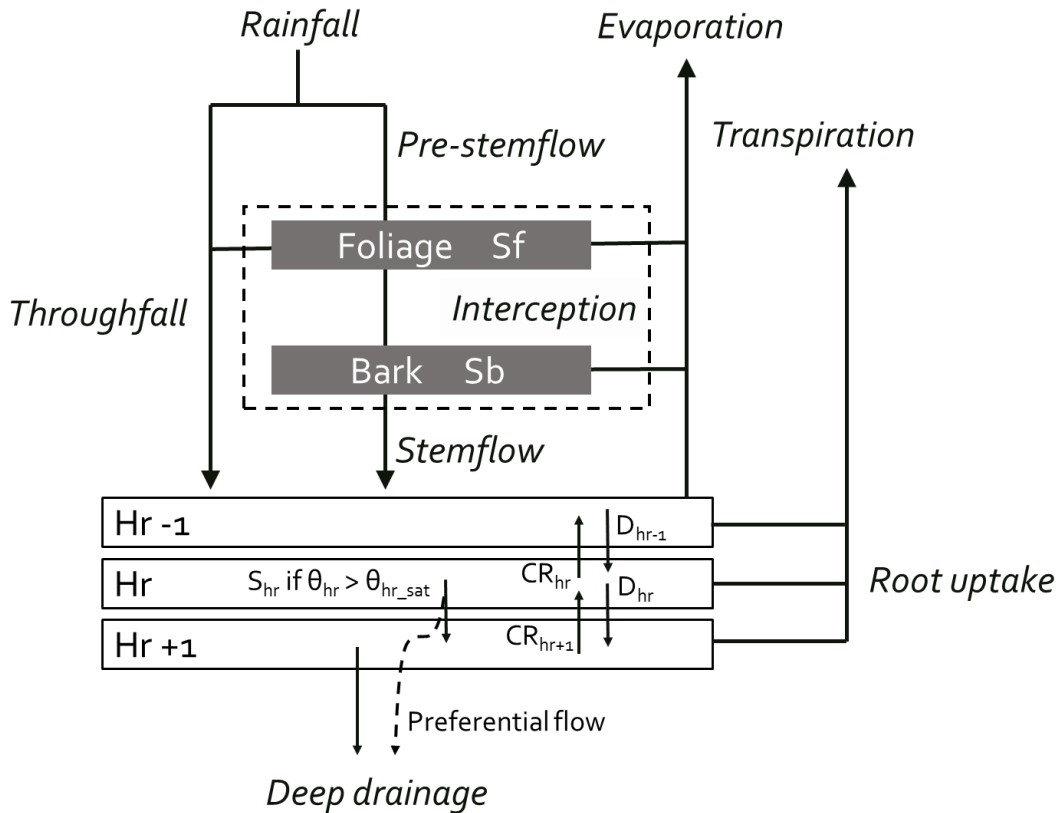

**Figure 1: Schematic representation of the water fluxes and pools in the water balance module. Rainfall is divided into throughfall reaching directly the forest floor and a pre-stemflow component intercepted by the foliage and the bark. Once the foliage and bark are saturated, the water surplus increases the throughfall flux and flows along the branches and the trunk to generate stemflow. The throughfall and stemflow fluxes enter in the upper part of the soil and then, move from one horizon to the other according to the Darcy's law. For a soil horizon *hr*, the water input fluxes can be the drainage from the upper horizon ($D_{hr-1}$) and the capillary rise from the lower horizon ($CR_{hr+1}$) that depend on the water potential gradient between the concerned horizons and on their hydraulic conductivity. The output fluxes are the drainage ($D_{hr}$) and the capillary rise ($CR_{hr}$), the root water uptake ($UP_{root(hr)}$) and the surplus ($S_{hr}$) that appears when the horizon water content exceeds the saturated water content. One part of this latter flux can directly leaves the system as deep drainage ($DD$) when preferential flow is considered, in addition to the water drainage of the last horizon. In parallel, water evaporates from foliage, bark and soil and is taken up by roots to enable tree transpiration. The evapo-transpiration fluxes are all calculated with the Penman-Monteith equation.**

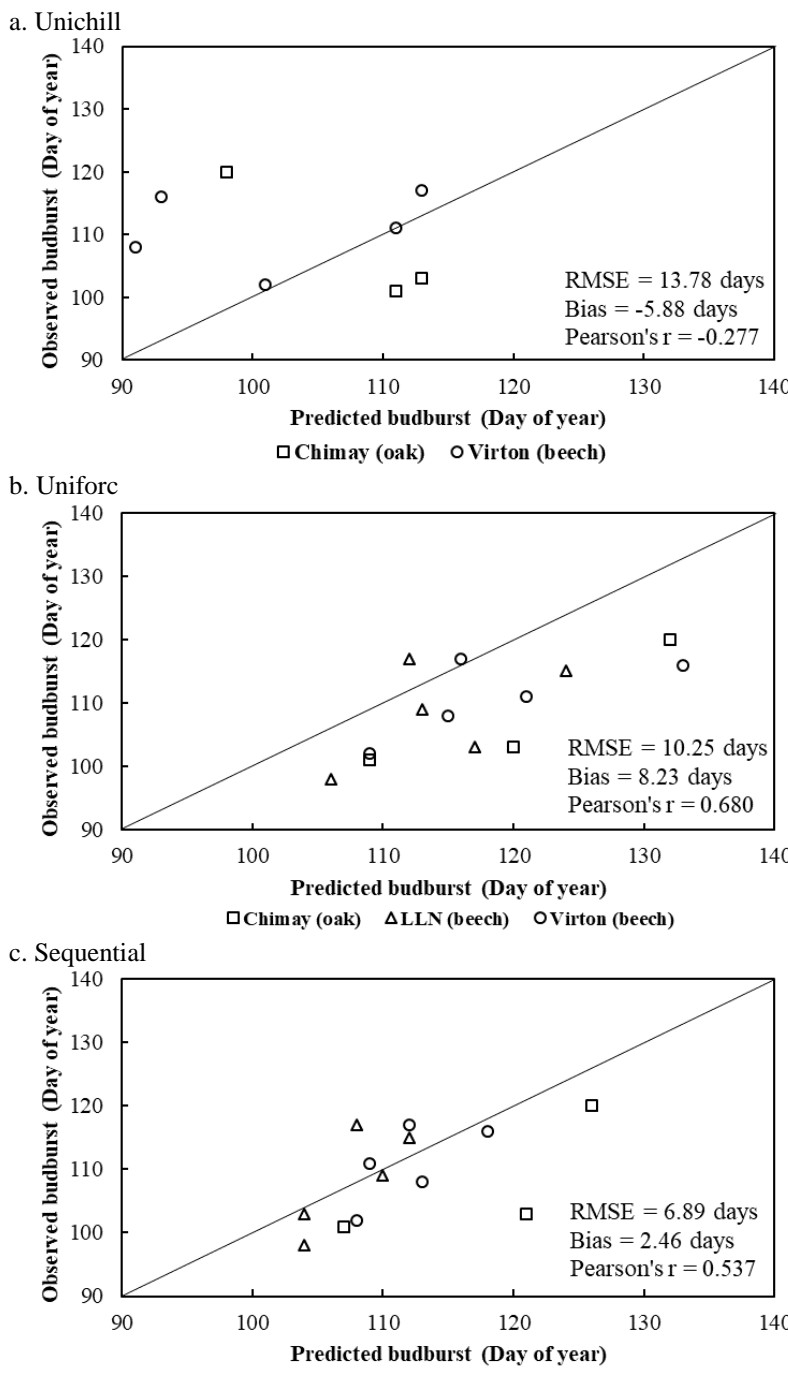

**Figure 2: Comparison of the observed and predicted budburst of the median tree in Chimay, Virton and Louvain-la-Neuve for the three phenological variants implemented: Unichill, Uniforc and Sequential. The quality of predictions is indicated by the *RMSE*, the absolute bias and the Pearson's correlation coefficient (*r*).**

a. Chimay (oak)

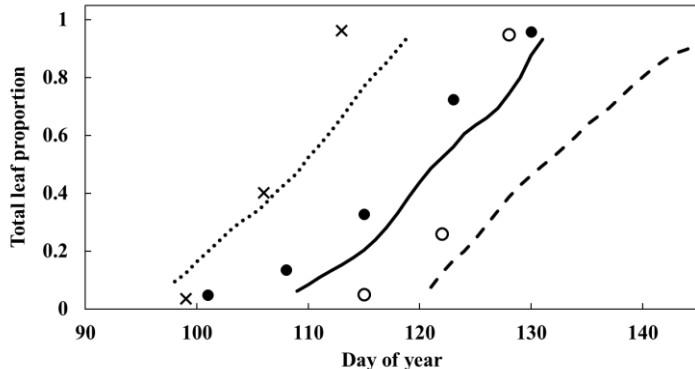

b. Louvain-la-Neuve (beech)

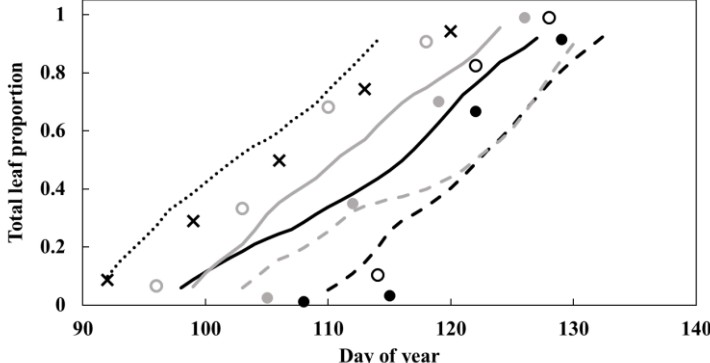

c. Virton (beech)

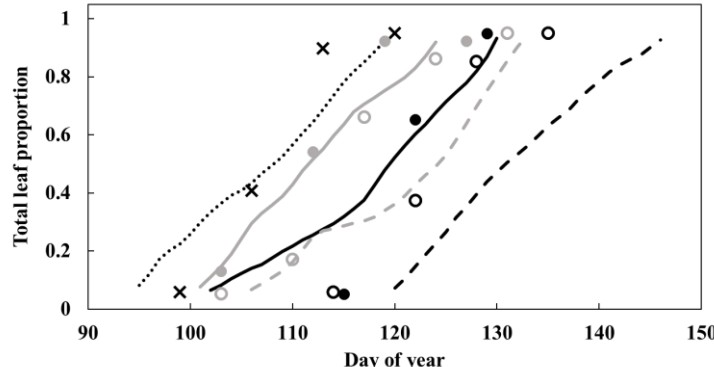

**Figure 3: Observed and predicted increase in leaf proportion in Chimay, Louvain-la-Neuve and Virton during the budburst and leaf development phase (data from 2012-2016). Observations are missing in Chimay for 2013, in Louvain-la-Neuve for 2012 and 2013 and in Virton for 2013.**

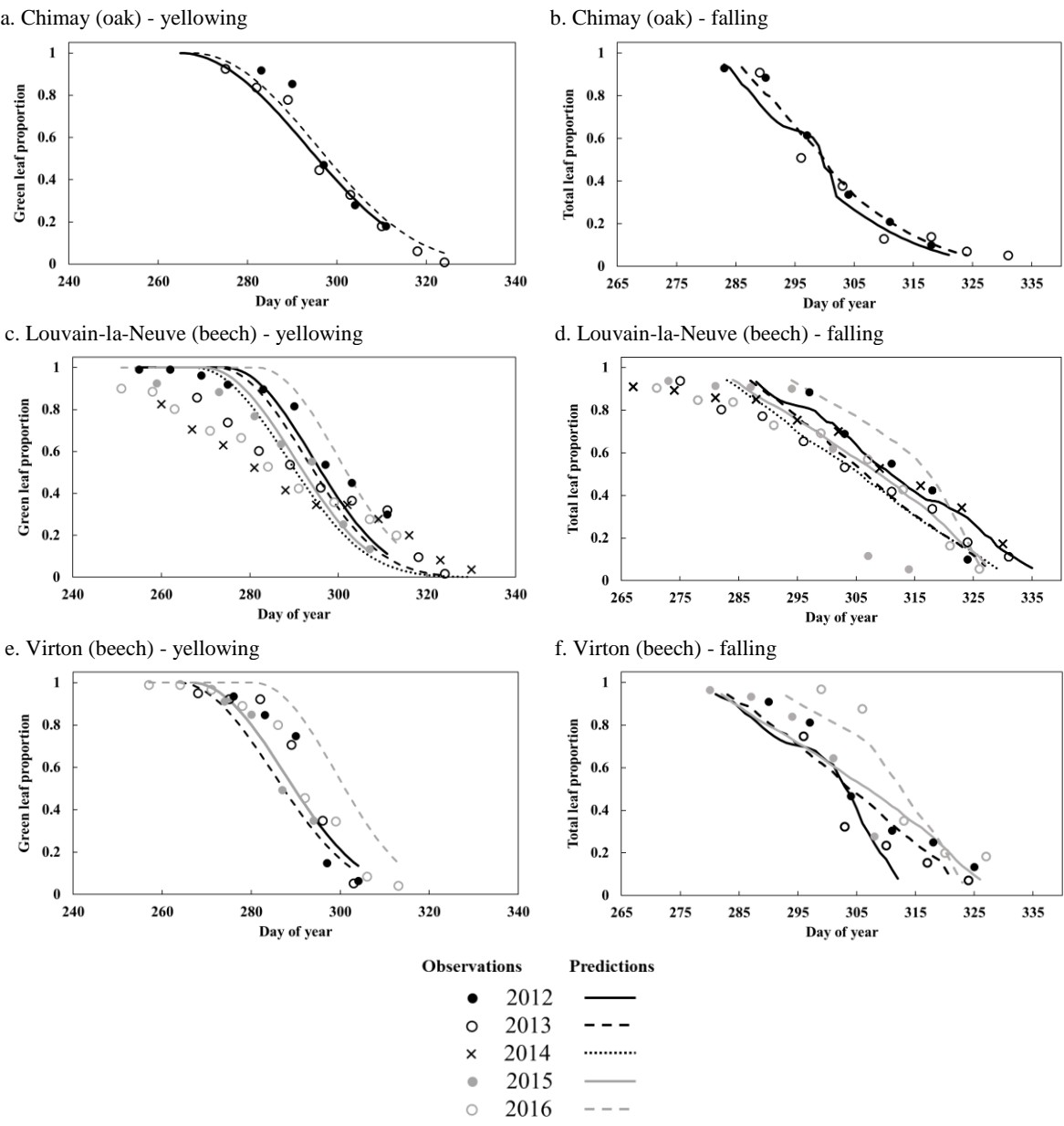

**Figure 4: Observed and predicted temporal dynamics in leaf yellowing and in leaf fall in Chimay, Louvain-la-Neuve and Virton (data from 2012-2016). Yellowing is represented by the decrease in green leaf proportion (left) and leaf fall by the decrease in total leaf proportion (right).**

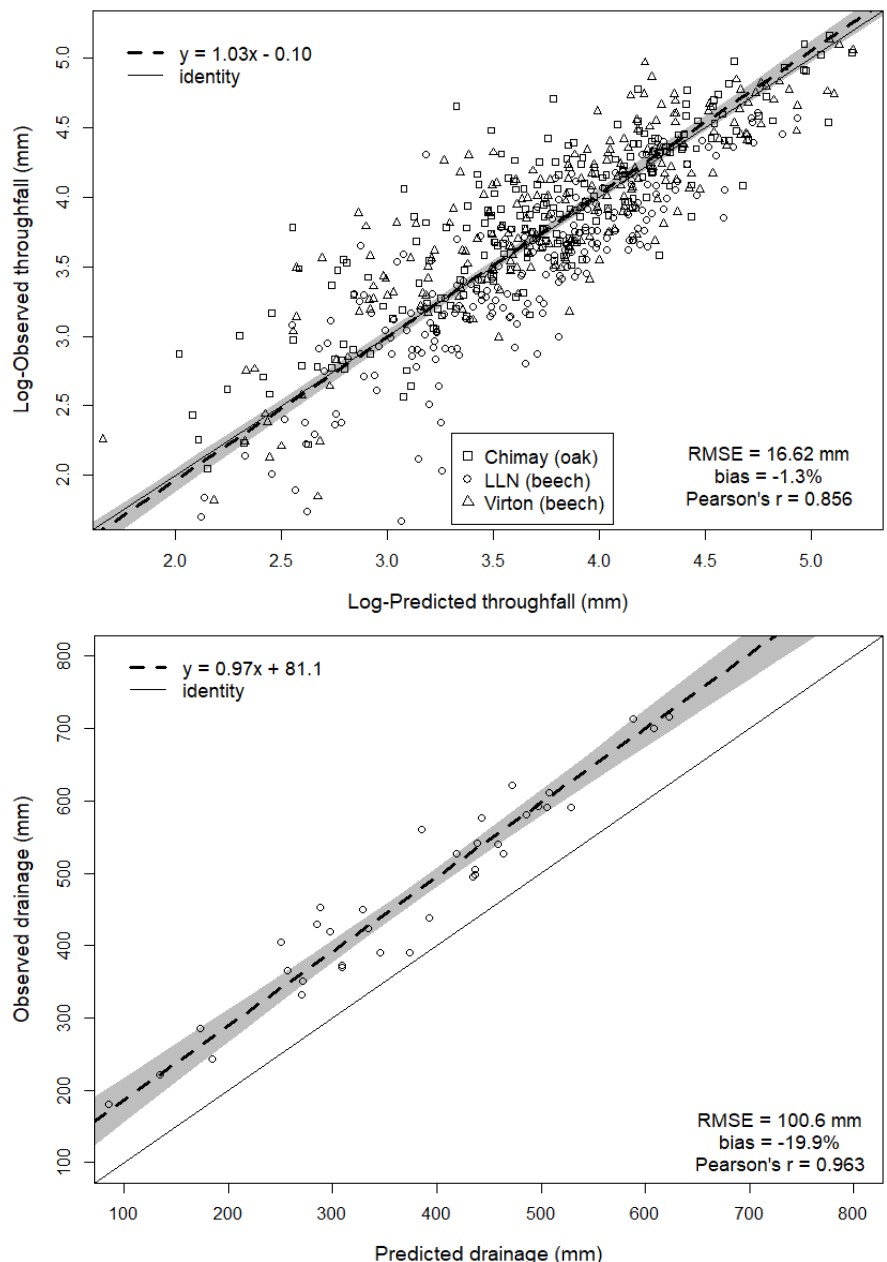

**Figure 5: Comparison of the log-transformed observed and predicted monthly throughfall in Chimay (oak), Louvain-la-Neuve (beech) and Virton (beech) between 2000 and 2016 (upper part) and comparison of observed and predicted annual drainage in all study stands between 2008 and 2016 (lower part). The quality of the non-transformed predictions is indicated by the *RMSE*, the relative bias and the Pearson's correlation coefficient (*r*). The shaded area represents the confidence interval of the Deming regression (95%) of observations on predictions and the solid line corresponds to the identity line.**

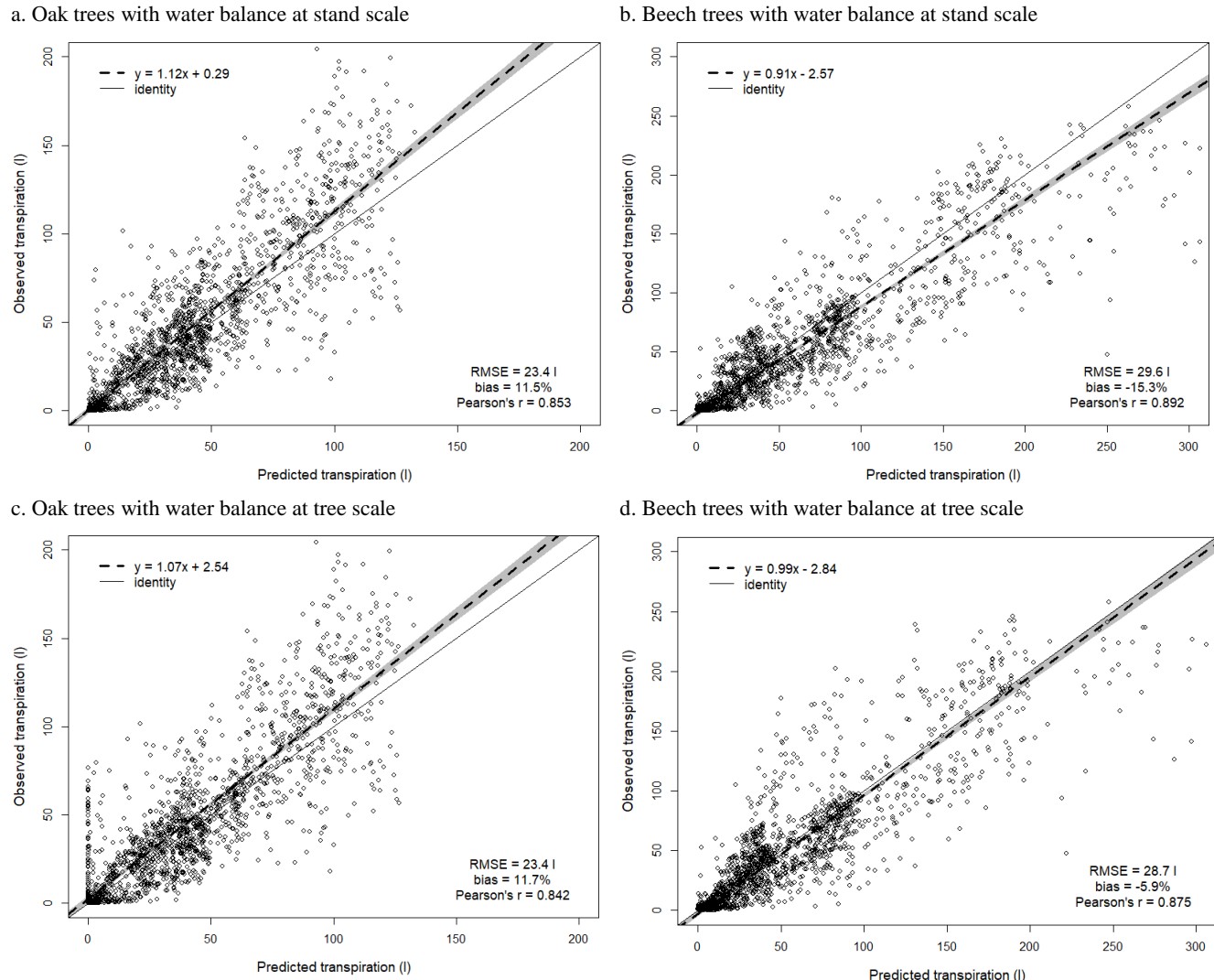

**Figure 6: Comparison of the observed and predicted daily transpiration of sessile oak and European beech in 2003 considering the tree and the stand scale for the water balance calculation. The quality of predictions is indicated by the *RMSE*, the relative bias and the Pearson's correlation coefficient (*r*). The shaded area represents the confidence interval of the Deming regression (95%) of observations on predictions and the solid line corresponds to the identity line.**

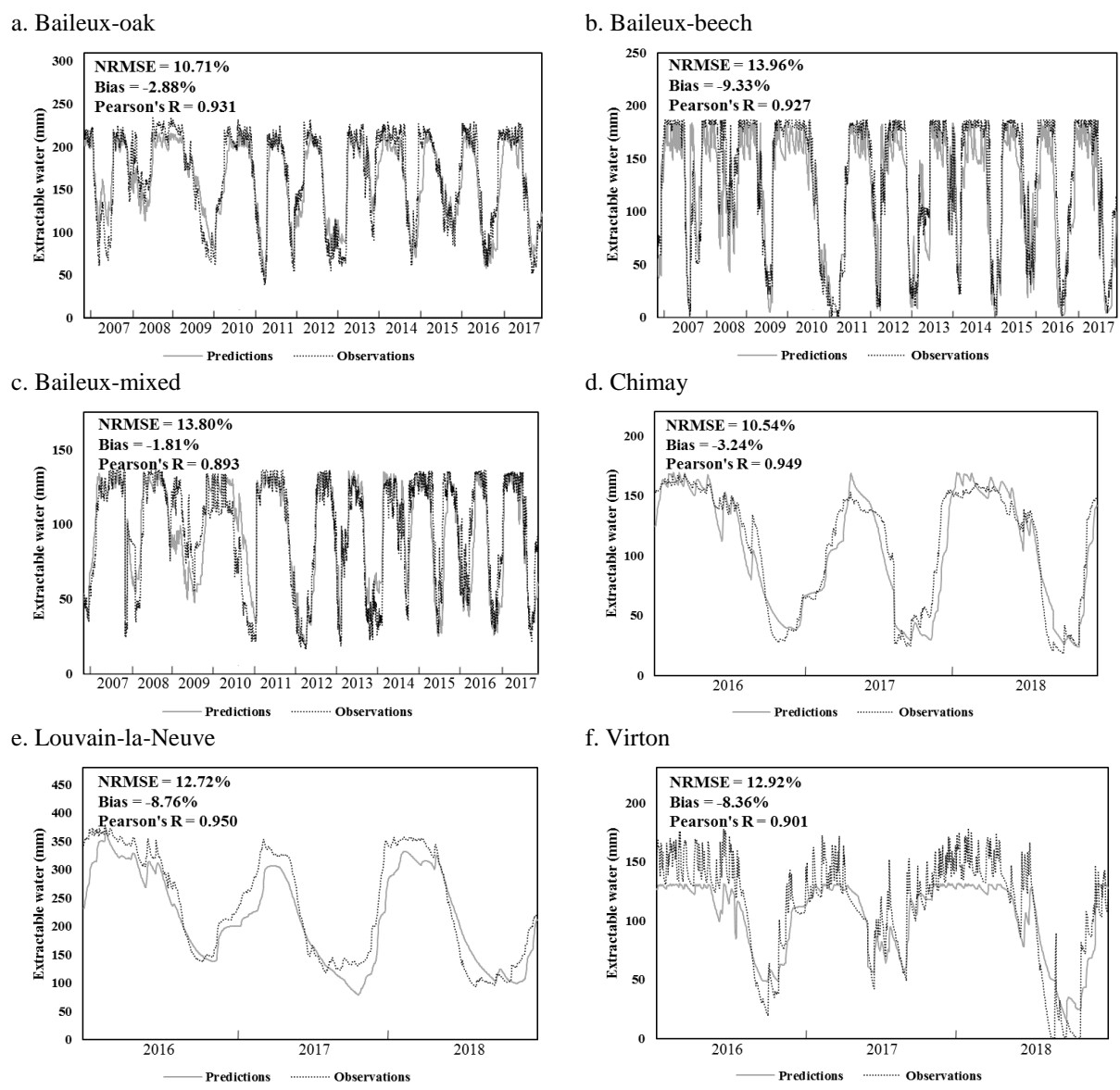

**Figure 7: Temporal dynamics of observed and predicted extractable water amount (mm) in the various stands. The prediction quality is indicated by the *NRMSE*, the relative bias and the Pearson's correlation coefficient (*r*).**

## 6 Code availability

The source code of CAPSIS and HETEROFOR is accessible to all the members of the CAPSIS co-development community. Those who want to join this community are welcome but must contact François de Coligny (coligny@cirad.fr) or Nicolas Beudez (nicolas.beudez@inra.fr) and sign the CAPSIS charter (http://capsis.cirad.fr/capsis/charter). This charter grants access

on all the models to the modellers of the CAPSIS community but only to them. The modellers may distribute the CAPSIS platform with their own model but not with the models of the others without their agreement. CAPSIS4 is a free software (LGPL licence) which includes the kernel, the generic pilots, the extensions and the libraries. For HETEROFOR, we also choose an LGPL license and decided to freely distribute it through an installer containing the CAPSIS4 kernel and the latest version (or any previous one) of HETEROFOR upon request from Mathieu Jonard (mathieu.jonard@uclouvain.be). The

source code for the modules published in Geoscientific Model Development (Jonard et al., accepted with major revisions, 2019; de Wergifosse et al., submitted) can be downloaded from the CAPSIS website (http://amap-dev.cirad.fr/projects/capsis/files) or obtained by contacting directly Mathieu Jonard.

The end-users who do not need access to the source code can install CAPSIS from an installer containing only the HETEROFOR model while the modellers who signed the CAPSIS charter can have access the complete version of CAPSIS

15 with all the models. Depending on your status (end-user vs modeller or developer), the instructions to install CAPSIS are given on the CAPSIS website (http://capsis.cirad.fr/capsis/documentation). The source code for the modules published in Geoscientific Model Development (Jonard et al., accepted with major revisions, 2019; de Wergifosse et al., submitted) can be downloaded from https://github.com/jonard76/HETEROFOR-1.0_LGPL_REVISED (DOI: 10.5281/zenodo.3591348).

## 7 Data availability

The data used in this paper are available through the input files for HETEROFOR which are embedded in the installer (see Sect. 6).

# 8 Appendices

## 8.1 Appendix A: Comparison of the log-transformed observed and predicted monthly throughfall in Chimay (sessile oak), Louvain-la-Neuve (European beech) and Virton (European beech) between 2000 and 2016. The shaded area represents the confidence interval of the Deming regression (95%) of observations on predictions and the solid line corresponds to the identity line.

a. Chimay (oak)

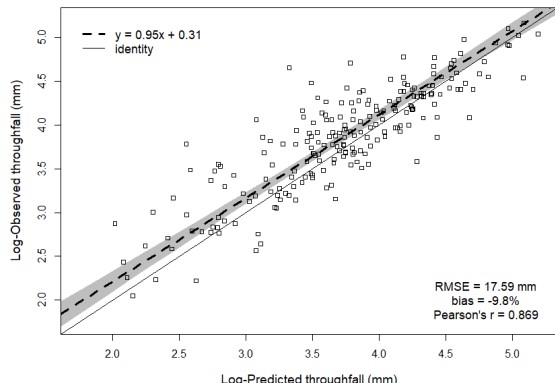

b. Louvain-la-Neuve (beech)

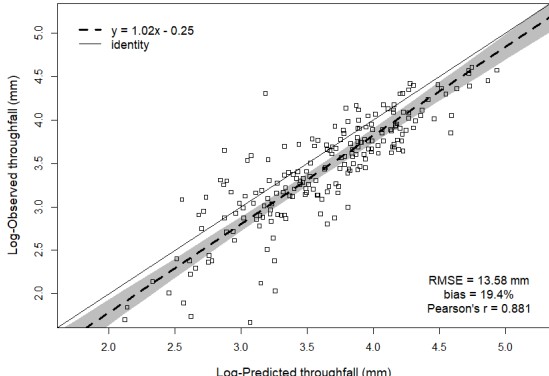

c. Virton (beech)

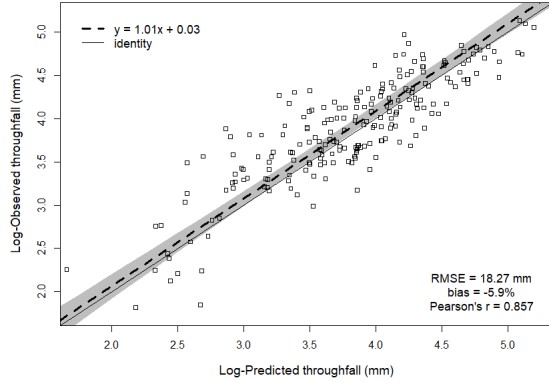

**8.2 Appendix B: Temporal dynamics of soil extractable water simulated with the tree approach in the three stands of Baileux for 2003. The shaded area represents the 80% confidence interval of the values obtained for the various pedons. For comparison, the mean extractable water calculated with the stand approach is represented with a dashed line.**

a. Baileux-beech

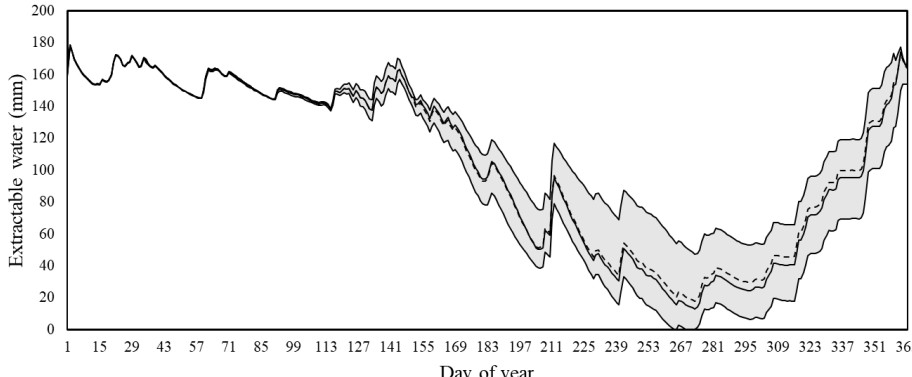

b. Baileux-mixed

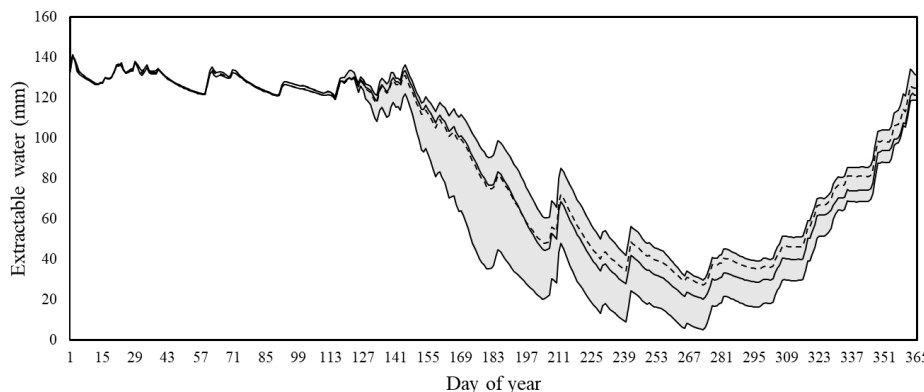

c. Baileux-oak

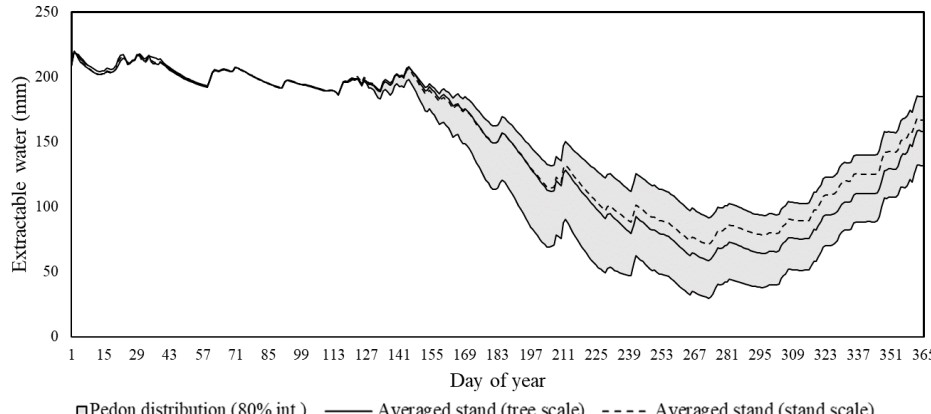

□ Pedon distribution (80% int.)  ⎯ Averaged stand (tree scale)  - - - - Averaged stand (stand scale)

**8.3 Appendix C: Comparison of observed and predicted basal area increments for sessile oak and European beech considering the two phenology modalities (tree vs stand scale). The quality of predictions is indicated by the *RMSE*, the relative bias and the Pearson's correlation coefficient (*r*). The shaded area represents the confidence interval of the Deming regression (95%) of observations on predictions and the solid line corresponds to the identity line.**

a. Oak trees with phenology at stand scale
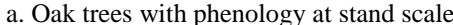

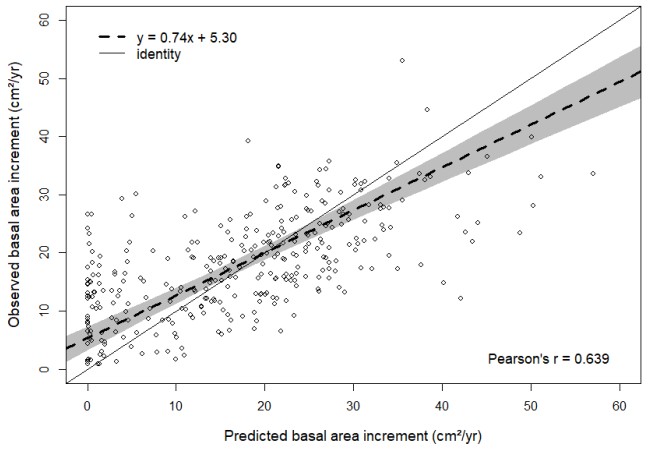

b. Beech trees with phenology at stand scale
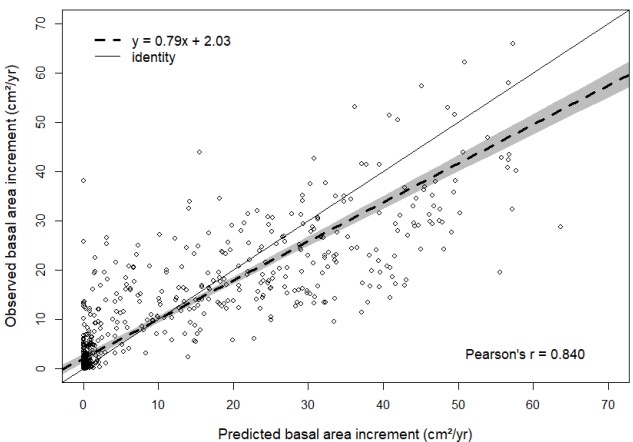

c. Oak trees with phenology at tree scale
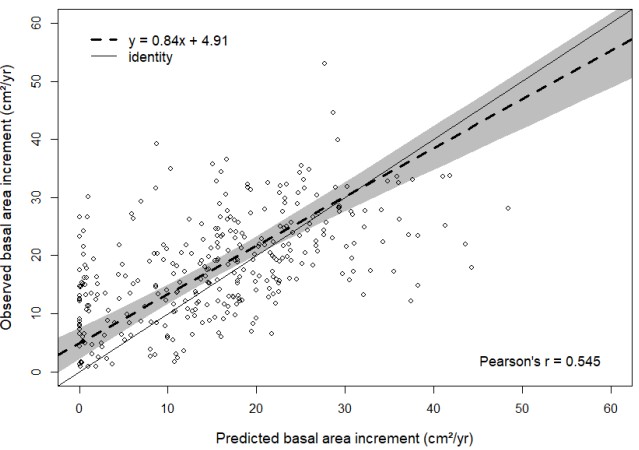

d. Beech trees with phenology at tree scale
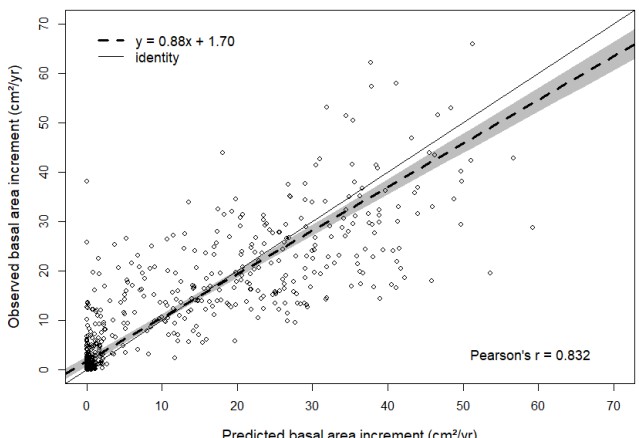

**9 Author contribution**

LdW, MJ, FA, NB and FdC developed the model code. LdW performed the simulation and analysed the model outputs. LdW and MJ prepared the manuscript with contributions from all co-authors.

5 **10 Competing interests**

The authors declare that they have no conflict of interest.

**11 Acknowledgements**

This work was supported by the FRIA grant n°1.E005.18, the *Service Public de Wallonie* (SPW/DGO 3/DNF) through the
10 *Accord-Cadre de Recherche et Vulgarisation Forestières 2014–2019* and by the *Fonds de la Recherche Scientifique – FNRS* under the PDR-WISD Grant n°09 (project SustainFor). We are also grateful to the two anonymous reviewers whose suggestions and comments help us to significantly improve the quality of this paper. We also would like to thank Mathieu Javaux for his sound advice on modeling water flows in the soil.

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
