# Peer review of "HETEROFOR 1.0: a spatially explicit model for exploring the response of structurally complex forests to uncertain future conditions. II. Phenology and water cycle."

_Geoscientific Model Development, 2019_

## Referee Comment (RC1) · Anonymous Referee #1 · 10 Sep 2019

I was curious to learn about new routines for phenology and water balances that can be used with a single-tree physiologically based model. However, I am quite disappointed because what is presented is not very well connected to the specific issues that should be addressed with the new spatially explicit model. For example, it should be crucial for a model which is particularly designed to represent the competition between individual trees under changing environmental conditions, to capture the differences in budburst or drought stress between understorey and dominant trees. I feel that this hasn't been addressed adequately, neither theoretical (neglecting several issues that are relevant

for individual modelling, not differentiating current model approaches appropriately, and lacking many relevant references) nor practically (i.e. without evaluation of individual transpiration rates, simulation of soil water development seems not very indicative).

The paper is quite long but not always stringent and to the point. If I understood it correctly, phenology is calculated species-specifically but with uniform drivers, while water balance is considering individual flows but not the spatial heterogeneity of water availability in the soil. So where is the benefit in comparison to cohort-based approaches? The evaluation of the water balance is done with averaged results from the literature where the influence of a structural component is not visible or by integrated values from the same sites that are (partially) also used for parametrization. So not even the impact of the differently structured forests that are described here has been evaluated. Furthermore, a sensitivity analysis, e.g. to explore the effect of other potential stand structures, has not been carried out. Therefore, I cannot say that the overall benefit of the new model has been sufficiently demonstrated.

More specific comments

Abstract:

It is a bit surprising to me that no reference to the specific demands on phenology and water balance regarding the spatial differentiation is made in the abstract. Also, the good results that the evaluation seems to provide cannot be judged if it is not indicated what kind of forests have been investigated. In particular, I would expect that the structure of the evaluation sites and not only the number is highlighted. Also, throughfall and average soil water development might not be the most important processes to be judged in a model that is designed to represent heterogeneous forest conditions. What about individual transpiration as could be determined by sapflux measurements?

Introduction:

It is of course an important motivation for the development of new models to better

judge the impacts of climate change. However, although the list of references about potential changes that are going to happen to forests in Europe in general is rather long, a reference about the impact of extreme weather events (which would be smaller in mixed forests) seems to be missing (e.g. see Kornhuber et al. 2019). You also might consider that the references about phenology as well as tree mortality are a bit outdated and should be replaced or complemented by newer ones (e.g. Piao et al. 2019, Waldau et al. 2018 and Klein et al. 2019, Etzold et al. 2019, Greenwood et al. 2017, respectively). Also, possible negative developments related to climate changes need to be mentioned (e.g. Liu et al. 2018) and the need to develop more mixed and structured forests should be much more emphasized (e.g. Rasche et al. 2013).

Overall, the introduction is rather lengthy with regard to general issues while quite short when it comes to reasoning about the importance of how individual phenology and water conditions drive the competition within a mixed and structured forest (see e.g. Jolly et al. 2004, Grote et al. 2016, Schaefer et al. 2018). In addition, please check English in places (e.g. page 4, line 3 or page 5 lines 11ff) and if the given references actually support what you want to say (i.e. is Fontes et al. 2010 really requesting a spatially differentiated modelling approach?).

Model description:

The phenological models include chilling and warming influences (any other as indicated by 'mainly' in line 31, page 7??) and comparisons are always interesting. With this respect, I wonder if the respective temperature is the air temperature as read from the input files or if the temperature is somehow processed before used here (considering e.g. a partial canopy cover that changes the temperature for the smaller trees). This is important because using the same temperature for understory and overstory trees is bound to lead to mismatches in phenology in structured forests. I was also a bit surprised about the reasoning that three models are necessary to be applied for specific sites or regions. Since the model has been designed to investigate climate change impacts – how can you ever know which model will work best on these new

conditions? In other words: If you are asked to select a phenological model for future conditions in northern regions, would you apply one that now works best in the North or one that now works best under the expected climate conditions – which are now in more southern regions? There certainly should be more deterministic criteria for model selection – otherwise a choice of models only increases the uncertainty connected to future model investigations. Since only one of these models has been used in the evaluation, I would advice to frame it so that this is the standard module for the model but that others (such as...) can be added (and shorten the description). Alternatively, all three models should be applied and results shown (at least in a supplement).

The water balance model looks quite complicated with many equations that do not necessarily be indicated as an equation. Therefore, I would like to see a much better scheme of flows than indicated in Fig. 1. The model seems to calculate interception and evaporation from the bark in unusual detail – although I guess that the estimation of bark surface and the interception of water at this surface is rather uncertain. I wonder if this can be justified by the size of the flux and the sensitivity of water balance fluxes to this compartment's properties? Furthermore, I think the ground/ soil vegetation evaporation needs a better description. As far as I understand, ground vegetation LAI dynamics are not part of this model (which is a pity) and are derived from 'ecosystem LAI' which needs to be given by the user (correct?). Doesn't this lead to quite some large errors due to ground vegetation being much more abundant during the early spring (e.g. Schulze et al. 2009)? Finally, water content in different soil layers seems to be calculated on a stand basis only, which assumes that water is sufficiently fast transported from places where transpiration is low and/or stemflow is high to sites with high water uptake and less input (this assumption should explicitly be stated). What I couldn't find is that the water uptake of smaller trees with less or no roots in deeper soils should be restricted to the upper soil layers. Did I miss this somehow?

Further Questions are: What is the difference between 'leaf biomass' and 'green leaves' (page 6, line 14)? How is it decided if maintenance respiration is calculated

as a fraction of gpp or as a separate process (page 6, line 25ff) and what are 'structural components' (stem, branches, coarse roots, bark? What about reserves and fruits? (page 6, line 31) – which are questions that are also not answered in Jonard et al. in review (at least not in the present form of the manuscript). I also wonder, if 'runoff is not included' means that percolation rates are bound to be unreasonable large at times or if there is an unreasonable large surface water pool if the rainfall is high. Why does the bark water storage depend on the state of the leaves? In the calculation of throughfall as difference between rainfall and stemflow (eq. 21) the canopy interception seems to be missing. Is the radiation used in eq. 29 the same as calculated in eq. 23 and 24? Please explicitly refer to eq. 29 and that ra and rs are estimated as 1/ga (gs), similarly to rs_foliage in eq. 53. How can the leaf width be fixed to be 4cm for all species reaching e.g. from Juglans to Salix? How can pre-stemflow be independent from tree species (page 16, line 29) if it is calculated for species- and individual-specific stemflow; and why is it calculated at all (being a part of the bark-intercepted water pool that is available for evaporation independent of its particular location)? Are there any references for the derivation of the different modifiers for gs_foliage (if not, a separate derivation based on data needs to be presented)?

In addition, I would recommend that names of parameters used and listed in table 1 are exactly the same (with/without sp indicator, y small or capital). Also, there shouldn't be parameters with the same name even in different equations (e.g. a, b, c, d in eqs. 4 and 12, 13, 16, . . .). If parameters are derived from literature data rather than directly taken from literature, the derivation should be shown (if only in a supplement). It is also strange, that the evaluation is carried out with stands that consist of three species, while parameters in table 1 indicate only two species. Where is the third? The derivation of some other variables is not explained either (e.g. BAI). Also check, if the range of the respective equation is valid, e.g. in eq. 5 it looks that Rld can get larger than 1 which would result in leafProp values larger than 1 which is not possible based on what has been said before. Again, English needs to be checked in places (e.g. page 7/line 4, page 10/line 16, page 8/line 11, . . .) and sentences have to be evaluated for their logic

(e.g. how can the calculated leaf biomass 'proportion' (of what?) allow to 'predict' the seasonal foliage development of individuals (a development predicted by a single mass??); a general model does not calculate a term only at a 'this moment'; . . .). In fact, I couldn't at all understand what is meant by the paragraph at page 8 lines 27ff.

Results:

None of these results couldn't have been produced with a plantation model. Why isn't there at least a differentiation by species for the phenological evaluation? The throughfall data and water content might be used for a minimum sensitivity test if measurements would have been compared using for example a) full individual data, b) no species differentiation, c) or no dimensional differentiation in order to demonstrate that some benefit arrives from the proposed model.

Discussion:

As the authors say, there are not many models that calculate carbon- and water fluxes and pool sizes with physiologically-based principles on the individual scale. A gradient might be indicated that starts with highly structured individual models (e.g. LIGNUM, Perttunen et al. 1988) which are too computationally expensive and would need too much spatially distributed boundary conditions to be used at the stand scale. Other models can be applied on stands but at the expense of a rough representation of physiological processes (e.g. SORTIE, Coates et al. 2003) or a neglect of some part of the full balances, i.e. the soil carbon processes (e.g. MAESPA but also FORMIND, Köhler and Huth 1998). Cohort-based models represent another compromise because horizontal differentiation is simplified (apart from 4C and ANAFORE there is also LandscapeDNDC-PSIM, Grote et al. 2011). The usefulness of such approaches has been discussed in reviews (e.g. Pacala et al. 1995, Berger et al. 2008, Bravo et al. 2019) and the role of a differentiated phenology (e.g. Gressler et al. 2015) as well as water balance (Roetzer et al. 2017) for individual competition and stand development has been demonstrated. I think that these approaches that all try to better account

for stand-structural issues should be separated from individual empirical approaches (e.g. FOREST) on the one hand and homogeneous physiological models (GOTILWA, CASTANEA) on the other. This differentiation, pros and cons are not nearly reflected in Table 5. In the current discussion, the model results are only judged by the closeness to the averaged measurements. There is no differentiation even between different species and no discussion about the effect of different sizes or particular positions of trees (see e.g. Wesolowski et al. 2006, Simioni et al....). This all doesn't seem related to the particular emphasize of the model. Also, potential improvements that could well be related to individual stress conditions (e.g. locally different water availability due to stemflow pattern or rooting depth) – eventually combining phenology and water balance issues (e.g. Sanz-Perez and Castro-Diez 2010, Xie et al. 2018) - are not mentioned.

Mentioned references not in the manuscript

Berger, U., Piou, C., Schiffers, K. and Grimm, V. (2008). Competition among plants: Concepts, individual-based modelling approaches, and a proposal for a future research strategy. Perspect. Plant Ecol. Evol. Syst. 9, 121-135.

Bravo, F., Fabrika, M., Ammer, C., Barreiro, S., Bielak, K., Coll, L., et al. (2019). Modelling approaches for mixed forests dynamics prognosis. Research gaps and opportunities. For. Syst. 28, 1-17.

Deckmyn, G., Verbeeck, H., Op de Beeck, M., Vansteenkiste, D., Steppe, K. and Ceulemans, R. (2008). ANAFORE: A stand-scale process-based forest model that includes wood tissue development and labile carbon storage in trees. Ecol. Modelling 215, 345-368.

Etzold, S., Ziemińska, K., Rohner, B., Bottero, A., Bose, A. K., Ruehr, N. K., et al. (2019). One Century of Forest Monitoring Data in Switzerland Reveals Species- and Site-Specific Trends of Climate-Induced Tree Mortality. Frontiers in Plant Science 10. doi: 10.3389/fpls.2019.00307

Greenwood, S., Ruiz-Benito, P., Martínez-Vilalta, J., Lloret, F., Kitzberger, T., Allen, C. D., et al. (2017). Tree mortality across biomes is promoted by drought intensity, lower wood density and higher specific leaf area. Ecol. Lett. 20, 539–553. doi: 10.1111/ele.12748

Gressler, E., Jochner, S., Capdevielle-Vargas, R. M., Morellato, L. P. C. and Menzel, A. (2015). Vertical variation in autumn leaf phenology of Fagus sylvatica L. in southern Germany. Agric. Forest Meteorol. 201, 176-186.

Grote, R., Korhonen, J. and Mammarella, I. (2011). Challenges for evaluating process-based models of gas exchange at forest sites with fetches of various species. For. Syst. 20, 389-406.

Grote, R., Gessler, A., Hommel, R., Poschenrieder, W. and Priesack, E. (2016). Importance of tree height and social position for drought-related stress and mortality. Trees-Struct. Funct. 30, 1467-1482.

Jolly, W. M., Nemani, R. and Running, S. W. (2004). Enhancement of understory productivity by asynchronous phenology with overstory competitors in a temperate deciduous forest. Tree Physiol. 24, 1069-1071.

Klein, T., Cahanovitc, R., Sprintsin, M., Herr, N. and Schiller, G. (2019). A nation-wide analysis of tree mortality under climate change: Forest loss and its causes in Israel 1948–2017. Forest Ecol. Manage. 432, 840-849. doi: 10.1016/j.foreco.2018.10.020

Köhler, P. and Huth, A. (1998). The effects of tree species grouping in tropical rainforest modelling: Simulations with the individual-based model FORMIND. Ecol. Modelling 109, 301-321.

Kornhuber, K., Osprey, S., Coumou, D., Petri, S., Petoukhov, V., Rahmstorf, S., et al. (2019). Extreme weather events in early summer 2018 connected by a recurrent hemispheric wave-7 pattern. Environ. Res. Lett. 14, 054002. doi: 10.1088/1748-9326/ab13bf

Liu, Q., Piao, S., Janssens, I. A., Fu, Y., Peng, S., Lian, X., et al. (2018). Extension of the growing season increases vegetation exposure to frost. Nature Commun. 9, 426. doi: 10.1038/s41467-017-02690-y

Pacala, S. W. and Deutschman, D. H. (1995). Details that matter: The spatial distribution of individual trees maintains forest ecosystem function. Oikos 74, 357-365.

Perttunen, J., Sievänen, R. and Nikinmaa, E. (1998). LIGNUM: a model combining the structure and the functioning of trees. Ecol. Modelling 108, 189-198.

Piao, S., Liu, Q., Chen, A., Janssens, I. A., Fu, Y., Dai, J., et al. (2019). Plant phenology and global climate change: current progresses and challenges. Glob. Change Biol. 25, 1922-1940. doi: 10.1111/gcb.14619

Rasche, L., Fahse, L. and Bugmann, H. (2013). Key factors affecting the future provision of tree-based forest ecosystem goods and services. Clim. Change 118, 579-593. doi: 10.1007/s10584-012-0664-5

Rötzer, T., Häberle, K. H., Kallenbach, C., Matyssek, R., Schütze, G. and Pretzsch, H. (2017). Tree species and size drive water consumption of beech/spruce forests - a simulation study highlighting growth under water limitation. Plant Soil 418, 337-356.

Sanz-Perez, V. and Castro-Diez, P. (2010). Summer water stress and shade alter bud size and budburst date in three mediterranean Quercus species. Trees-Struct. Funct. 24, 89-97

Schäfer, C., Thurm, E. A., Rötzer, T., Kallenbach, C. and Pretzsch, H. (2018). Daily stem water deficit of Norway spruce and European beech in intra- and interspecific neighborhood under heavy drought. Scand. J. Forest. Res. 33, 568-582.

Schulze, I.-M., Bolte, A., Schmidt, W. and Eichhorn, J. (2009). "Phytomass, Litter and Net Primary Production of Herbaceous Layer," In: Functioning and Management of European Beech, eds Brumme, R. and Khanna, P. K. (Heidelberg: Springer-Verlag), 155-181.

Wesolowski, T. and Rowinski, P. (2006). Timing of bud burst and tree-leaf development in a multispecies temperate forest. Forest Ecol. Manage. 237, 387-393.

Waldau, T. and Chmielewski, F.-M. (2018). Spatial and temporal changes of spring temperature, thermal growing season and spring phenology in Germany 1951–2015. Meteorol. Zeitschrift 27, 335 - 342. doi: 10.1127/metz/2018/0923

Xie, Y., Wang, X., Wilson, A. M. and Silander, J. A. (2018). Predicting autumn phenology: How deciduous tree species respond to weather stressors. Agric. Forest Meteorol. 250-251, 127-137.
* * *

---

## Short Comment (SC1) · 20 Sep 2019

**Response to reviewer comment on "HETEROFOR 1.0: a spatially explicit model for exploring the response of structurally complex forests to uncertain future conditions. II. Phenology and water cycle" by Louis de Wergifosse et al.**

**Anonymous Referee #1**

I was curious to learn about new routines for phenology and water balances that can be used with a single-tree physiologically based model. However, I am quite disappointed because what is presented is not very well connected to the specific issues that should be addressed with the new spatially explicit model. For example, it should be crucial for a model which is particularly designed to represent the competition between individual trees under changing environmental conditions, to capture the differences in budburst or drought stress between understorey and dominant trees. I feel that this hasn't been addressed adequately, neither theoretical (neglecting several issues that are relevant for individual modelling, not differentiating current model approaches appropriately, and lacking many relevant references) nor practically (i.e. without evaluation of individual transpiration rates, simulation of soil water development seems not very indicative).

Author response:

The difference in budburst between understorey and dominant trees is not considered in HETEROFOR since the processes responsible for this trend are still poorly known. While the vertical profile in air temperature was first advanced to explain this difference in budburst, it seems that the buffering effect of canopy on air temperature only plays a secondary role and that ontogeny has the major effect (Vitasse 2013). Yet, in uneven-aged stands, the tree age is generally not known. In addition, long-term monitoring program such as ICP Forests focus on dominant and co-dominant trees for the phenological observations. Therefore, even if we were able to conceptually elaborate a phenological module that accounts for the tree social status, we would not be able to calibrate it. We have however designed our model in order to facilitate the implementation of new budburst routines as knowledge in phenology improves.

For the drought stress, HETEROFOR captures at least part of the differences between understorey and dominant trees since it accounts for the impact of tree vertical position within the canopy on the intercepted radiation (ray tracing approach for describing the light competition), on aerodynamic resistance (Eq. 36 to 38) and on stomatal conductance (Eq. 54). Through the calculation of these variables as a function of tree position in the canopy, HETEROFOR generates differences in the transpiration rate between understorey and overstorey trees.

Physiologically, the drought stress can be measured based on the stomatal conductance which is, among others, dependent on the tree height in the model (Eq. 54). Consequently, for a same soil water potential, the stomatal conductance of a taller tree is lower than that of a smaller one. In addition, if the smaller tree is in the understorey, it receives less radiation and the associated aerodynamic resistance is lower, resulting in a lower transpiration rate.

The paper is quite long but not always stringent and to the point. If I understood it correctly, phenology is calculated species-specifically but with uniform drivers, while water balance is considering individual flows but not the spatial heterogeneity of water availability in the soil. So where is the benefit in comparison to cohort-based approaches?

R:

The advantage of the cohort-based models over the stand models is to account for the various tree species and size classes of the trees of a stand. When considering a high number of cohorts, the cohort-based approach is close to the tree-level one, except that it is not able to account for the spatial arrangement of the trees within the stand. In a cohort-based model, a stand composed of several cohorts with different tree species and size classes is represented as an intimate mixture of all these cohorts. In this perspective, all the trees of a same cohort are undergoing the same growth conditions. Actually, the trees of a same cohort can experience very different competition conditions depending on the composition of their neighbourhood. Using a spatially-explicit and individual-based approach, HETEROFOR calculates light interception, photosynthesis, respiration, carbon allocation, dimensional growth and transpiration at the individual level. With HETEROFOR, all possible spatial arrangements of the trees within a stand can be considered and the impact of this spatial arrangement on stand productivity and water balance is accounted for since many processes are modelled at the individual level. It is however not relevant or possible to model all the processes at the tree level. Regarding phenology, the current knowledge on the processes explaining the difference in budburst between understorey and dominant trees is not sufficient to elaborate a robust model (see our detailed response below) and the calibration and validation of such a model would be problematic given the lack of phenological observations on understorey trees at the regional and continental levels.

For the water dynamics within the soil, taking into account the spatial heterogeneity (in the horizontal dimension) would significantly increase the model complexity (as well as the computing time) without necessarily improving the quality of the predictions. Indeed, the rooting area of each tree would be determined only very roughly since the belowground processes (especially the root competition among tree species) are still poorly understood. Therefore, we decided not considering the soil spatial heterogeneity in the horizontal dimension since the increase in complexity would generate a lot of uncertainties (Seidl et al., 2012). The same choice was done by the developers of two well-known process- and individual-based models (MAESPA and iLand). Simioni *et al.* (2016) chose to discretize the aboveground and belowground space in 3D voxels and evaluated the benefit of such an approach. Their 3D representation provided better results for light interception, tree growth and tree water stress but it performed similarly to the 1D representation regarding soil water content and evapotranspiration. An individual-based model without soil spatial discretisation in the horizontal dimension is however still interesting since estimating transpiration at the tree level provides more accurate predictions at the stand level and this has a direct impact on the evaluation of the soil water availability and on its effect on tree growth.

The evaluation of the water balance is done with averaged results from the literature where the influence of a structural component is not visible or by integrated values from the same sites that are (partially) also used for parametrization. So not even the impact of the differently structured forests that are described here has been evaluated. Furthermore, a sensitivity analysis, e.g. to explore the effect of other potential stand structures, has not been carried out. Therefore, I cannot say that the overall benefit of the new model has been sufficiently demonstrated.

R:

We recognize that the overall benefit of our model could have been better demonstrated and we thank the reviewer for the very relevant suggestion to carry out a sensitivity analysis to explore the effect of stand structure and spatial heterogeneity. For different stands composed of the same trees, we will evaluate the impact of grouping the trees per patch of the same size and/or tree species compared to an intimate mixture. In addition, we will evaluate the loss of accuracy when replacing our trees by average trees in order to reproduce the functioning of a cohort-based model.

**More specific comments**

**Abstract:**

It is a bit surprising to me that no reference to the specific demands on phenology and water balance regarding the spatial differentiation is made in the abstract. Also, the good results that the evaluation seems to provide cannot be judged if it is not indicated what kind of forests have been investigated. In particular, I would expect that the structure of the evaluation sites and not only the number is highlighted. Also, throughfall and average soil water development might not be the most important processes to be judged in a model that is designed to represent heterogeneous forest conditions. What about individual transpiration as could be determined by sapflux measurements?

R:

The abstract will be rewritten to better highlight the diversity of stand structure and composition used to evaluate the model. In addition, we will compare the individual transpiration with sapflux measurements in order to better demonstrate the benefit of using an individual-based model.

**Introduction:**

It is of course an important motivation for the development of new models to better judge the impacts of climate change. However, although the list of references about potential changes that are going to happen to forests in Europe in general is rather long, a reference about the impact of extreme weather events (which would be smaller in mixed forests) seems to be missing (e.g. see Kornhuber et al. 2019). You also might consider that the references about phenology as well as tree mortality are a bit outdated and should be replaced or complemented by newer ones (e.g. Piao et al. 2019, Waldau et al. 2018 and Klein et al. 2019, Etzold et al. 2019, Greenwood et al. 2017, respectively). Also, possible negative developments related to climate changes need to be mentioned (e.g. Liu et al. 2018) and the need to develop more mixed and structured forests should be much more emphasized (e.g. Rasche et al. 2013).

R:

Thanks a lot for suggesting all these additional references! We will use them to improve the introduction.

Overall, the introduction is rather lengthy with regard to general issues while quite short when it comes to reasoning about the importance of how individual phenology and water conditions drive the competition within a mixed and structured forest (see e.g. Jolly et al. 2004, Grote et al. 2016, Schaefer et al. 2018). In addition, please check English in places (e.g. page 4, line 3 or page 5 lines 11ff) and if the given references actually support what you want to say (i.e. is Fontes et al. 2010 really requesting a spatially differentiated modelling approach?).

R:

We will develop more the benefit of using an individual-based approach for modelling water balance and its impact on tree growth. We will also check English throughout the manuscript as well as the relevance of the cited references.

Model description:

The phenological models include chilling and warming influences (any other as indicated by 'mainly' in line 31, page 7??) and comparisons are always interesting. With this respect, I wonder if the respective temperature is the air temperature as read from the input files or if the temperature is somehow processed before used here (considering e.g. a partial canopy cover that changes the temperature for the smaller trees). This is important because using the same temperature for understory and overstory trees is bound to lead to mismatches in phenology in structured forests.

Compared to dominant trees, the budburst of understorey trees generally occur earlier according to most of the studies (Seiwa, 1999a and b; Augspurger and Bartlett, 2003; Gill et al., 1998, Schieber, 2006; Vitasse, 2013) even if a few authors reported the opposite trend for some tree species (Augspurger and Bartlett, 2003; Richardson and O'Keefe, 2009). Warmer temperatures in the understorey is one of the hypotheses advanced to explain this difference in budburst between under- and over-storey (Augspurger and Bartlett, 2003; Schieber, 2006). Using a construction crane, Vitasse (2013) tested this hypothesis by transplanting seedlings of 5 tree species at 30 and 35 m height in the canopy. He observed that the budburst of the seedling growing at these heights was much earlier than that of the dominant trees. He concluded that the main factor to explain the difference in budburst is driven by ontogeny (tree age and height) as stated by Seiwa (1999a and b) and that the vertical profile in temperature within the canopy only plays a secondary role.

To capture the differences in budburst between understorey and dominant trees, ontogeny must be taken into account in priority. However, tree age is generally not known in uneven-aged stands. In addition, most of the phenological observations are currently achieved on dominant and codominant trees (see ICP Forest manual on phenology) which would make the calibration of such a detailed modelling approach difficult or valid only for specific sites. Modelling the vertical profile of temperature within the canopy is not trivial. This research field is experiencing renewed interest and most studies are currently observation studies. They highlight that understorey temperature compared to temperature outside of the canopy is highly dependent of the stand characteristics i.e. the canopy cover and its species composition (von Arx et al., 2013; Kovacs et al., 2017; Zellweger et al., 2019; Leuzinger and Körner, 2007) as well of the plot slope, exposure and topographic position (Daly et al., 2010; Zellweger et al., 2019).

In conclusion, we consider that the current knowledge on phenology as well as the availability of data at large scale is not sufficient to implement a robust model that would be able to capture the differences in budburst between understorey and dominant trees. We have however designed our model in order to facilitate the implementation of new budburst routines as knowledge in phenology improves.

Augspurger, C. K., & Bartlett, E. A. (2003). Differences in leaf phenology between juvenile and adult trees in a temperate deciduous forest. Tree Physiology, 23(8), 517-525.

Daly, C., Conklin, D. R., & Unsworth, M. H. (2010). Local atmospheric decoupling in complex topography alters climate change impacts. International Journal of Climatology, 30(12), 1857-1864.

Gill, D. S., Amthor, J. S., & Bormann, F. H. (1998). Leaf phenology, photosynthesis, and the persistence of saplings and shrubs in a mature northern hardwood forest. Tree Physiology, 18(5), 281-289.

Kovács, B., Tinya, F., & Ódor, P. (2017). Stand structural drivers of microclimate in mature temperate mixed forests. Agricultural and Forest Meteorology, 234, 11-21.

Leuzinger, S., & Körner, C. (2007). Tree species diversity affects canopy leaf temperatures in a mature temperate forest. Agricultural and forest meteorology, 146(1-2), 29-37.

Richardson, A. D., & O'Keefe, J. (2009). Phenological differences between understory and overstory. In Phenology of ecosystem processes (pp. 87-117). Springer, New York, NY.

Schieber, B. (2006). Spring phenology of European beech (Fagus sylvatica L.) in a submountain beech stand with different stocking in 1995–2004. Journal of Forest Science, 52(5), 208-216.

Seiwa, K. (1999a). Changes in leaf phenology are dependent on tree height in Acer mono, a deciduous broad-leaved tree. Annals of Botany, 83(4), 355-361.

Seiwa, K. (1999b). Ontogenetic changes in leaf phenology of Ulmus davidiana var. japonica, a deciduous broad-leaved tree. Tree physiology, 19(12), 793-797.

Vitasse, Y. (2013). Ontogenic changes rather than difference in temperature cause understory trees to leaf out earlier. New Phytologist, 198(1), 149-155.

Von Arx, G., Graf Pannatier, E., Thimonier, A., & Rebetez, M. (2013). Microclimate in forests with varying leaf area index and soil moisture: potential implications for seedling establishment in a changing climate. Journal of Ecology, 101(5), 1201-1213.

Zellweger, F., Coomes, D., Lenoir, J., Depauw, L., Maes, S. L., Wulf, M., ... & Schmidt, W. (2019). Seasonal drivers of understorey temperature buffering in temperate deciduous forests across Europe. Global Ecology and Biogeography.

I was also a bit surprised about the reasoning that three models are necessary to be applied for specific sites or regions. Since the model has been designed to investigate climate change impacts – how can you ever know which model will work best on these new conditions? In other words: If you are asked to select a phenological model for future conditions in northern regions, would you apply one that now works best in the North or one that now works best under the expected climate conditions – which are now in more southern regions? There certainly should be more deterministic criteria for model selection – otherwise a choice of models only increases the uncertainty connected to future model investigations. Since only one of these models has been used in the evaluation, I would advice to frame it so that this is the standard module for the model but that others (such as. . .) can be added (and shorten the description). Alternatively, all three models should be applied and results shown (at least in a supplement).

R :

We implemented three approaches to allow the model user to compare them and to choose the most appropriate. Using the three approaches could also be interesting to characterize the conceptual uncertainty. In the future, new approaches could be added as the knowledge on budburst improves. In our case, we retained the budburst model that best reproduce the current observations.

The water balance model looks quite complicated with many equations that do not necessarily be indicated as an equation. Therefore, I would like to see a much better scheme of flows than indicated in Fig. 1.

R:

We will revise Fig. 1 in order to have a more detailed representation of the water balance routine. However, part of the information is in Fig. 2. We will evaluate in which extent the two figures could be merged.

The model seems to calculate interception and evaporation from the bark in unusual detail – although I guess that the estimation of bark surface and the interception of water at this surface is rather uncertain. I wonder if this can be justified by the size of the flux and the sensitivity of water balance fluxes to this compartment's properties?

The rainfall interception by bark varies from 22 to 54 mm per year depending on the stand type and represents on average 20% of the rainfall interception by canopy (foliage + bark). This intercepted amount is therefore not negligible and deserves to be taken into account since it varies with stand structure and composition.

Furthermore, I think the ground/ soil vegetation evaporation needs a better description. As far as I understand, ground vegetation LAI dynamics are not part of this model (which is a pity) and are derived from 'ecosystem LAI' which needs to be given by the user (correct?). Doesn't this lead to quite some large errors due to ground vegetation being much more abundant during the early spring (e.g. Schulze et al. 2009)?

R:

Our approach for modelling the LAI of ground vegetation is just a temporary one until the development of a more complete approach is finished. For the master thesis of Brieuc Reylandt, we have developed a regeneration routine based on the regeneration library of CAPSIS. This routine predicts seedling and ground vegetation growth based on light interception. For seedlings and saplings, a cohort-based approach is used. When the saplings reach a given height (e.g. 12 m), they are individualized during the recruitment phase. This regeneration routine has already been tested but still needs further evaluation before being coupled with the water balance module and presented in a paper.

Finally, water content in different soil layers seems to be calculated on a stand basis only, which assumes that water is sufficiently fast transported from places where transpiration is low and/or stemflow is high to sites with high water uptake and less input (this assumption should explicitly be stated).

R:

We will state in the revised manuscript that HETEROFOR assumes a fast horizontal redistribution of water from places of low transpiration to sites of higher one. In addition, we will better explain why we decided not discretised the soil volume in a first approach (see above).

What I couldn't find is that the water uptake of smaller trees with less or no roots in deeper soils should be restricted to the upper soil layers. Did I miss this somehow?

R:

In our sites, the soil depth varies from 0.5 to 1.5 m. If we consider a vertical root development of 25 cm per year (Collet et al., 2006), the bottom of the soil profile would be reached after 2 to 6 years by seedling roots. In addition, most of the fine roots are located within the upper soil horizon (Claus and George, 2005). According to Bakker et al. (2008), 90 to 95% of beech fine roots are located in the first 60 cm of the soil. Even if oak roots are distributed a little bit more deeply, they are also in majority in the upper horizons. Jonard et al. (2011) reported that 39% of the fine roots are in the 0-30 cm layer. For these reasons, some authors concluded that the vertical fine root profile is only slightly affected by tree age (Bakker et al., 2008; Claus and George, 2005).

In HETEROFOR, trees are recruited at a height of 12 m which correspond to 25 -30 years old trees. So, when trees are recruited (individualized in HETEROFOR), they already have a vertical fine root development similar to that of the adult trees. However, we agree with the reviewer that, for very small trees (seedlings less than 10 years old), the water uptake should be restricted to upper soil layers and we will take this into account in our regeneration routine.

Bakker, M. R., Turpault, M. P., Huet, S., & Nys, C. (2008). Root distribution of Fagus sylvatica in a chronosequence in western France. Journal of Forest Research, 13(3), 176-184.

Claus, A., & George, E. (2005). Effect of stand age on fine-root biomass and biomass distribution in three European forest chronosequences. Canadian Journal of Forest Research, 35(7), 1617-1625.

Collet, C., Löf, M., & Pagès, L. (2006). Root system development of oak seedlings analysed using an architectural model. Effects of competition with grass. Plant and Soil, 279(1-2), 367-383.

Jonard, F., André, F., Ponette, Q., Vincke, C., & Jonard, M. (2011). Sap flux density and stomatal conductance of European beech and common oak trees in pure and mixed stands during the summer drought of 2003. Journal of Hydrology, 409(1-2), 371-381.

Further Questions are: What is the difference between 'leaf biomass' and 'green leaves' (page 6, line 14)?

R:

The total leaf biomass includes the biomass of the green leaves as well as that of the discoloured leaves (during the yellowing phase). The green leaf proportion is the ratio between the green leaf biomass and the total leaf biomass at the full leaf development. This will be clarified in the text.

How is it decided if maintenance respiration is calculated as a fraction of gpp or as a separate process (page 6, line 25ff) and what are 'structural components' (stem, branches, coarse roots, bark? What about reserves and fruits? (page 6, line 31) – which are questions that are also not answered in Jonard et al. in review (at least not in the present form of the manuscript).

R:

The option used to calculate maintenance respiration (gpp fraction or temperature dependent routine) is selected by the user during the initialisation of the model (see user manual). The structural components are the stem and branches including bark and the coarse roots. This will be mentioned more clearly in the revised versions.

I also wonder, if 'runoff is not included' means that percolation rates are bound to be unreasonable large at times or if there is an unreasonable large surface water pool if the rainfall is high.

R:

When water saturation is reached in a soil layer, the water surplus is transferred to the next horizon.

Why does the bark water storage depend on the state of the leaves?

R:

Contrary to what is mentioned in the manuscript, the bark water storage capacity is not influenced by the season (leafed or leafless period). This bad wording results from the fact that the bark storage capacity is derived from the relationship predicting individual stemflow volume based on trunk circumference and rainfall depth, which depends on the season (André et al., 2008). As presented in details by these authors, this dependence of the model parameters with the season reflects the corresponding variations of the stemflow rate (i.e., volume of stemflow generated per unit of rainfall depth) and of the rainfall threshold for stemflow (i.e., minimum rainfall depth required for stemflow to occur at the base of the trunk). Stemflow rates are larger during the leafless period compared with the leaved season and inversely for the rainfall threshold. With the stemflow relationship, the bark storage capacity can be derived from the stemflow rate and the rainfall threshold (Eq. 13). As they vary in an opposite way with the season, the corresponding variation of the bark storage capacity is not significant as shown by André et al. (2008). The corresponding section will be reworded accordingly in the revised manuscript.

André F., Jonard M., Ponette Q., 2008. Influence of species and rain event characteristics on stemflow volume in a temperate mixed oak-beech stand. Hydrological Processes 22, 4455-4466.

In the calculation of throughfall as difference between rainfall and stemflow (eq. 21) the canopy interception seems to be missing. Is the radiation used in eq. 29 the same as calculated in eq. 23 and 24? Please explicitly refer to eq. 29 and that ra and rs are estimated as 1/ga (gs), similarly to rs_foliage in eq. 53.

R:

Eq. 21 estimates throughfall proportion as 1 – stemflow proportion, and this stemflow proportion is then applied to non-intercepted rainfall to calculate throughfall. Throughfall is therefore the difference between non-intercepted rainfall (which accounts for canopy interception) and stemflow.

Regarding the radiation used in eq. 29, the explanation is provided at page 14 lines 5-9. At line 11, it is specified that the aerodynamic resistance is defined as the inverse of the aerodynamic conductance, but we will add it as an equation in the revised version.

How can the leaf width be fixed to be 4cm for all species reaching e.g. from Juglans to Salix?

R:

In the revised version of the model, we will consider the leaf width as a species-specific parameter.

How can pre-stemflow be independent from tree species (page 16, line 29) if it is calculated for species- and individual-specific stemflow; and why is it calculated at all (being a part of the bark-intercepted water pool that is available for evaporation independent of its particular location)?

R:

The pre-stemflow is indeed not independent from tree species. We used the term "independently" to say 'separately' for each species. We will revise the wording to avoid confusion.

Pre-stemflow is calculated as a first step to obtain stemflow (see page 17 lines 5-19).

Are there any references for the derivation of the different modifiers for gs_foliage (if not, a separate derivation based on data needs to be presented)?

R:

The mathematical form of the relationships used to describe the effect of the different modifiers is largely inspired from the literature (Granier and Bréda, 1996; Tuzet et al., 2003 and Schaëfer et al., 2000) while the parameters were adjusted based on canopy conductance data from Jonard et al. (2011). In the revised paper, Tuzet et al. (2003) will be added in the reference list.

Granier, A., & Bréda, N. (1996). Modelling canopy conductance and stand transpiration of an oak forest from sap flow measurements. In Annales des Sciences Forestieres (Vol. 53, No. 2-3, pp. 537-546). EDP Sciences.

Jonard, F., André, F., Ponette, Q., Vincke, C., & Jonard, M. (2011). Sap flux density and stomatal conductance of European beech and common oak trees in pure and mixed stands during the summer drought of 2003. Journal of Hydrology, 409(1-2), 371-381.

Tuzet, A., Perrier, A., & Leuning, R. (2003). A coupled model of stomatal conductance, photosynthesis and transpiration. Plant, Cell & Environment, 26(7), 1097-1116.

Schäfer, K. V. R., Oren, R., & Tenhunen, J. D. (2000). The effect of tree height on crown level stomatal conductance. Plant, Cell & Environment, 23(4), 365-375.

In addition, I would recommend that names of parameters used and listed in table 1 are exactly the same (with/without sp indicator, y small or capital). Also, there shouldn't be parameters with the same name even in different equations (e.g. a, b, c, d in eqs. 4 and 12, 13, 16, . . .). If parameters are derived from literature data rather than directly taken from literature, the derivation should be shown (if only in a supplement).

R:

These suggestions will be taken into account in the revised version of the paper.

It is also strange, that the evaluation is carried out with stands that consist of three species, while parameters in table 1 indicate only two species. Where is the third?

R:

The third species is a species from the understorey (hornbeam) for which less information is available. For this species, we used specific parameters for light interception, photosynthesis, respiration and carbon allocation but the same parameters as beech for water balance and phenology given their similarity. This will be clarified.

The derivation of some other variables is not explained either (e.g. BAI).

In the revised version of the paper, we will provide more information on these variables.

Also check, if the range of the respective equation is valid, e.g. in eq. 5 it looks that Rld can get larger than 1 which would result in leafProp values larger than 1 which is not possible based on what has been said before. Again, English needs to be checked in places (e.g. page 7/line 4, page 10/line 16,

page 8/line 11, . . .) and sentences have to be evaluated for their logic (e.g. how can the calculated leaf biomass 'proportion' (of what?) allow to 'predict' the seasonal foliage development of individuals (a development predicted by a single mass??); a general model does not calculate a term only at a 'this moment'; . . .). In fact, I couldn't at all understand what is meant by the paragraph at page 8 lines 27ff.

R:

When all the reviews will be available, we will double check again the different equations and revise English and logic, especially in the mentioned paragraphs. Regarding Eq.5 and 6, the sum of Rld, that can indeed be superior to 1, is divided by the leaf development threshold (LD*) and this ratio increase progressively with time reaching 1 at the end of the leaf development phase. Anyway, the code does not allow the leafProp to be greater than 1.

The paragraph at page 8 explains how we integrated the within-population variability in the budburst process. As the module was calibrated based on observations carried out on trees representative of the stand, the predicted budburst starting date is expected to be that of an average tree. Since, at this date, the leaf expansion of some trees has already started, the model shifts it to obtain the budburst starting date of the earliest trees. This shift is equal to half the period between the budburst of the first and the last tree.  The leaf development starts at the budburst of the earliest trees and is calibrated to represent the average evolution of all the trees of a same species.

**Results:**

None of these results couldn't have been produced with a plantation model. Why isn't there at least a differentiation by species for the phenological evaluation? The throughfall data and water content might be used for a minimum sensitivity test if measurements would have been compared using for example a) full individual data, b) no species differentiation, c) or no dimensional differentiation in order to demonstrate that some benefit arrives from the proposed model.

R:

For the phenological evaluation, the differentiation was made by site but this also corresponds to a species differentiation since only the dominant species was observed in each site. This will be clarified in the revised version.

To better highlight the benefit of our approach, we will carry out a sensitivity analysis to explore the effect of stand structure and spatial heterogeneity. For different stands composed of the same trees, we will evaluate the impact of grouping the trees per patch of the same size and/or tree species compared to an intimate mixture. In addition, we will evaluate the loss of accuracy when replacing our trees by averaged trees in order to reproduce the functioning of a cohort-based model as suggested by the reviewer. A similar approach has been developed by Collalti et al. (2016) who compared a cohort- and stand-based approach.

In addition, we will also compare the individual transpiration with sapflux measurements in order to better demonstrate the benefit of using an individual-based model.

Collalti, A., Marconi, S., Ibrom, A., Trotta, C., Anav, A., D'andrea, E., ... & Grünwald, T. (2016). Validation of 3D-CMCC Forest Ecosystem Model (v. 5.1) against eddy covariance data for 10 European forest sites. Geosci. Model Dev., 9, 479–504.

**Discussion:**

As the authors say, there are not many models that calculate carbon- and water fluxes and pool sizes with physiologically-based principles on the individual scale. A gradient might be indicated that starts with highly structured individual models (e.g. LIGNUM, Perttunen et al. 1988) which are too computationally expensive and would need too much spatially distributed boundary conditions to be used at the stand scale. Other models can be applied on stands but at the expense of a rough representation of physiological processes (e.g. SORTIE, Coates et al. 2003) or a neglect of some part of the full balances, i.e. the soil carbon processes (e.g. MAESPA but also FORMIND, Köhler and Huth 1998). Cohort-based models represent another compromise because horizontal differentiation is simplified (apart from 4C and ANAFORE there is also LandscapeDNDC-PSIM, Grote et al. 2011). The usefulness of such approaches has been discussed in reviews (e.g. Pacala et al. 1995, Berger et al. 2008, Bravo et al. 2019) and the role of a differentiated phenology (e.g. Gressler et al. 2015) as well as water balance (Roetzer et al. 2017) for individual competition and stand development has been demonstrated. I think that these approaches that all try to better account for stand-structural issues should be separated from individual empirical approaches (e.g. FOREST) on the one hand and homogeneous physiological models (GOTILWA, CASTANEA) on the other. This differentiation, pros and cons are not nearly reflected in Table 5.

R:

Thanks a lot for all these suggestions that will help us to improve the discussion on the impact of stand structure on carbon and water fluxes and to better position our model compared to the other existing approaches. Table 5 will be revised to include information on the spatial resolution of the model as well as of the process considered (tree, cohort or stand level).

In the current discussion, the model results are only judged by the closeness to the averaged measurements. There is no differentiation even between different species and no discussion about the effect of different sizes or particular positions of trees (see e.g. Wesolowski et al. 2006, Simioni et al. . ..). This all doesn't seem related to the particular emphasize of the model. Also, potential improvements that could well be related to individual stress conditions (e.g. locally different water availability due to stemflow pattern or rooting depth) – eventually combining phenology and water balance issues (e.g. Sanz-Perez and Castro-Diez 2010, Xie et al. 2018) - are not mentioned.

R:

The species are implicitly differentiated in the discussion as, for phenology and throughfall, observations are only made for a few representative individuals of the dominant species (according to ICP protocol). We will change our figure captions to make it more explicit.

In the revised version, we will discuss new results (sensitivity analysis and evaluation of individual transpiration based on the sap flux measurements) that will highlight the benefit of using an individual-based approach. All the references suggested by the reviewer will be considered to improve this discussion.

**Mentioned references not in the manuscript**

Berger, U., Piou, C., Schiffers, K. and Grimm, V. (2008). Competition among plants: Concepts, individual-based modelling approaches, and a proposal for a future research strategy. Perspect. Plant Ecol. Evol. Syst. 9, 121-135.

Bravo, F., Fabrika, M., Ammer, C., Barreiro, S., Bielak, K., Coll, L., et al. (2019). Modelling approaches for mixed forests dynamics prognosis. Research gaps and opportunities. For. Syst. 28, 1-17.

Deckmyn, G., Verbeeck, H., Op de Beeck, M., Vansteenkiste, D., Steppe, K. and Ceulemans, R. (2008). ANAFORE: A stand-scale process-based forest model that includes wood tissue development and labile carbon storage in trees. Ecol. Modelling 215, 345-368.

Etzold, S., Zieminska, K., Rohner, B., Bottero, A., Bose, A. K., Ruehr, N. K., et al. ´ (2019). One Century of Forest Monitoring Data in Switzerland Reveals Species- and Site-Specific Trends of Climate-Induced Tree Mortality. Frontiers in Plant Science 10. doi: 10.3389/fpls.2019.00307

Greenwood, S., Ruiz-Benito, P., Martínez-Vilalta, J., Lloret, F., Kitzberger, T., Allen, C. D., et al. (2017). Tree mortality across biomes is promoted by drought intensity, lower wood density and higher specific leaf area. Ecol. Lett. 20, 539–553. doi: 10.1111/ele.12748

Gressler, E., Jochner, S., Capdevielle-Vargas, R. M., Morellato, L. P. C. and Menzel, A. (2015). Vertical variation in autumn leaf phenology of Fagus sylvatica L. in southern Germany. Agric. Forest Meteorol. 201, 176-186.

Grote, R., Korhonen, J. and Mammarella, I. (2011). Challenges for evaluating processbased models of gas exchange at forest sites with fetches of various species. For. Syst. 20, 389-406.

Grote, R., Gessler, A., Hommel, R., Poschenrieder, W. and Priesack, E. (2016). Importance of tree height and social position for drought-related stress and mortality. Trees-Struct. Funct. 30, 1467-1482.

Jolly, W. M., Nemani, R. and Running, S. W. (2004). Enhancement of understory productivity by asynchronous phenology with overstory competitors in a temperate deciduous forest. Tree Physiol. 24, 1069-1071.

Klein, T., Cahanovitc, R., Sprintsin, M., Herr, N. and Schiller, G. (2019). A nation-wide analysis of tree mortality under climate change: Forest loss and its causes in Israel 1948–2017. Forest Ecol. Manage. 432, 840-849. doi: 10.1016/j.foreco.2018.10.020

Köhler, P. and Huth, A. (1998). The effects of tree species grouping in tropical rainforest modelling: Simulations with the individual-based model FORMIND. Ecol. Modelling 109, 301-321.

Kornhuber, K., Osprey, S., Coumou, D., Petri, S., Petoukhov, V., Rahmstorf, S., et al. (2019). Extreme weather events in early summer 2018 connected by a recurrent hemispheric wave-7 pattern. Environ. Res. Lett. 14, 054002. doi: 10.1088/1748- 9326/ab13bf

Liu, Q., Piao, S., Janssens, I. A., Fu, Y., Peng, S., Lian, X., et al. (2018). Extension of the growing season increases vegetation exposure to frost. Nature Commun. 9, 426. doi: 10.1038/s41467-017-02690-y

Pacala, S. W. and Deutschman, D. H. (1995). Details that matter: The spatial distribution of individual trees maintains forest ecosystem function. Oikos 74, 357-365.

Perttunen, J., Sievänen, R. and Nikinmaa, E. (1998). LIGNUM: a model combining the structure and the functioning of trees. Ecol. Modelling 108, 189-198.

Piao, S., Liu, Q., Chen, A., Janssens, I. A., Fu, Y., Dai, J., et al. (2019). Plant phenology and global climate change: current progresses and challenges. Glob. Change Biol. 25, 1922-1940. doi: 10.1111/gcb.14619

Rasche, L., Fahse, L. and Bugmann, H. (2013). Key factors affecting the future provision of tree-based forest ecosystem goods and services. Clim. Change 118, 579-593. doi: 10.1007/s10584-012-0664-5

Rötzer, T., Häberle, K. H., Kallenbach, C., Matyssek, R., Schütze, G. and Pretzsch, H. (2017). Tree species and size drive water consumption of beech/spruce forests - a simulation study highlighting growth under water limitation. Plant Soil 418, 337-356.

Sanz-Perez, V. and Castro-Diez, P. (2010). Summer water stress and shade alter bud size and budburst date in three mediterranean Quercus species. Trees-Struct. Funct. 24, 89-97

Schäfer, C., Thurm, E. A., Rötzer, T., Kallenbach, C. and Pretzsch, H. (2018). Daily stem water deficit of Norway spruce and European beech in intra- and interspecific neighborhood under heavy drought. Scand. J. Forest. Res. 33, 568-582.

Schulze, I.-M., Bolte, A., Schmidt, W. and Eichhorn, J. (2009). "Phytomass, Litter and Net Primary Production of Herbaceous Layer," In: Functioning and Management of European Beech, eds Brumme, R. and Khanna, P. K. (Heidelberg: Springer-Verlag), 155-181.

Wesolowski, T. and Rowinski, P. (2006). Timing of bud burst and tree-leaf development in a multispecies temperate forest. Forest Ecol. Manage. 237, 387-393.

Waldau, T. and Chmielewski, F.-M. (2018). Spatial and temporal changes of spring temperature, thermal growing season and spring phenology in Germany 1951–2015. Meteorol. Zeitschrift 27, 335 - 342. doi: 10.1127/metz/2018/0923

Xie, Y., Wang, X., Wilson, A. M. and Silander, J. A. (2018). Predicting autumn phenology: How deciduous tree species respond to weather stressors. Agric. Forest Meteorol. 250-251, 127-137. Interactive comment on Geosci. Model Dev. Discuss., https://doi.org/10.5194/gmd-2019-201, 2019.

---

## Referee Comment (RC2) · Anonymous Referee #2 · 14 Oct 2019

1. The authors present two modules (phenology and water budget) of a new forest growth model HETEROFOR, which is a process-based model considering spatial distribution of individual trees of mixed-species forests. This paper is the second one of the series papers (seems will be more than 2). After reading the first paper (Jonard et al., under review process in GMD) and the comprehensive comments by Referee #1 to this one, my additional comments will be given here.

2. The connection between the phenology module and the processes of photosynthesis and the allocation of NPP was not described in this paper and the first paper. The

leaved period from budburst to the start point of yellowing is the period of leaf development. In the phenology module, the progress of leaf development is solely controlled by the temperature. However, the photosynthesis process modelled in HETEROFOR (as described in the first paper) has complicated controlling mechanism including other factors like PAR. The allocation of NPP to leaves and fine roots is further controlled by the nutrient status of the plant. Both the photosynthetic rate and the allocation to leaves will determine the leaf development and the further photosynthesis. Please clearly specify the relationship of photosynthesis, allocation, and leaf development.

3. For the soil water simulation, what's the domain of each individual tree? How does HETEROFOR deal with the spatial heterogeneity of soil water budget? Although the partition of rainwater to interception, throughfall, and stemflow for each tree is one-dimensional in the module, the spatial distribution of the individual trees will make the soil water input different under each individual tree. As the soil water availability will have control on foliage conductance (equation 54) and thus on photosynthesis, it is necessary for a clearer description of the 3-dimensional soil water budget.

4. P3L15: I don't see any focus on climate change in this paper

5. P3L16: a temperature increase in which year?

6. P7L10: what's the endpoint of the gradual loss of photosynthesis and transpiration of the yellowing leaves

7. P8L1: the average budburst date, average of what?

8. P8L16: equals -> reaches

9. P11L4: epsilon in equation 12 is not defined

10. P11L10: how can the threshold for stemflow appearance is tree size-independent?

11. P1123: why equation 16 does not use equation 12 as the stemflow rate?

12. P27L9: for the calculation of annual drainage by the chloride mass balance, a

monthly calculation won't produce more reliable result?

---

## Referee Comment (RC3) · Anonymous Referee #1 · 14 Oct 2019

I understand that it is difficult to parameterize and evaluated a phenological model that differentiates for leaves in different layers in a structural forest. However, if the model is not considering these structural effects, it may be accurate but is not new and is not particular related to the specific approach of HETEROFOR. This would change if at least the option would be available (e.g. as a yet uniform parameter) to make budburst (I see you found some examples already) and leaf fall (Gressler et al. 2015) dependent on tree size or specific environmental conditions. In particular, since I assume that the model should be applied to more tree species than those presented here, which may

respond differently.

Regarding the question on "the benefit in comparison to cohort-based approaches" I see that you would like to differentiate between processes that you can more or less accurately determine on the individual trees (such as light availability) and those where the uncertainty is larger and that are thus treated in the same way as in cohort or even stand scale models (such as water availability). Not be able to precisely determine root competition indeed increases uncertainty. On the other hand, neglecting it makes the model inconsistent and biased towards aboveground competition process (that by the way are relying on crown form assumptions and leaf area distribution also based on rough assumptions). Actually, this is the reasoning to apply stand-scale and cohort models which are therefore more consistent. I am even aware of an approach that recalculates individual growth from cohort-based biomass gain (Poschenrieder et al. 2013). The basic criteria, however, may be if your half-individual approach is actually performing better than a cohort-based approach. I am therefore excited to see your analysis in this behalf.

P.S. Please try to understand my remark about 'runoff is not included'. It refers to the water at the surface when the water capacity of the soil is reached. Is it nevertheless forced to percolate into or through the soil or does it pile up at the surface?

Gressler, E., Jochner, S., Capdevielle-Vargas, R. M., Morellato, L. P. C. and Menzel, A. (2015). Vertical variation in autumn leaf phenology of Fagus sylvatica L. in southern Germany. Agric. Forest Meteorol. 201, 176-186. doi: 10.1016/j.agrformet.2014.10.013

Poschenrieder, W., Grote, R. and Pretzsch, H. (2013). Extending a physiological forest model by an observation-based tree competition module improves spatial representation of diameter growth. Eur. J. Forest Res. 132, 943-958. doi: 10.1007/s10342-013-0730-1

---

## Author Comment (AC1) · 21 Nov 2019

**Response to reviewer comment on "HETEROFOR 1.0: a spatially explicit model for exploring the response of structurally complex forests to uncertain future conditions. II. Phenology and water cycle" by Louis de Wergifosse et al.**

**Anonymous Referee #1**

Author response:

In addition to the previous response posted on the 20th of September, the following comments are addressed hereafter.

I understand that it is difficult to parameterize and evaluated a phenological model that differentiates for leaves in different layers in a structural forest. However, if the model is not considering these structural effects, it may be accurate but is not new and is not particular related to the specific approach of HETEROFOR. This would change if at least the option would be available (e.g. as a yet uniform parameter) to make budburst (I see you found some examples already) and leaf fall (Gressler et al. 2015) dependent on tree size or specific environmental conditions. In particular, since I assume that the model should be applied to more tree species than those presented here, which may respond differently.
R:

As suggested, we have tested the implementation of an option to make the phenology size-dependent. In the initialisation phase of the model, if the user selects the option "phenology at the tree scale", the leaf development is first triggered in the smallest trees of each tree species and then progressively in the taller ones according to their height. Inversely, during leaf fall, the tallest trees lose their leaves before the smaller ones. At the stand scale, the option "phenology at the tree scale" provide exactly the same leaf development/fall than the default option but the difference appear at the tree scale. The default option assumes that all trees initiate budburst/leaf fall at the same time and display the same progressive leaf development/fall. The alternative option "phenology at the tree scale" supposes trees break down or lose their leaves one after the other depending on their size. These two options give the opportunity to compare two contrasted hypotheses regarding individual tree phenology and to evaluate to which extent it has an impact on tree growth. This alternative option will be described in the revised manuscript and discussed.

Regarding the question on "the benefit in comparison to cohort-based approaches" I see that you would like to differentiate between processes that you can more or less accurately determine on the individual trees (such as light availability) and those where the uncertainty is larger and that are thus treated in the same way as in cohort or even stand scale models (such as water availability). Not be able to precisely determine root competition indeed increases uncertainty. On the other hand, neglecting it makes the model inconsistent and biased towards aboveground competition process (that by the way are relying on crown form assumptions and leaf area distribution also based on rough assumptions).

R:

The default option for the water balance module in HETEROFOR considers that all trees take up water in the same soil horizons assuming that soil water is redistributed homogeneously between two hourly time steps.

Following the discussions initiated on this subject with the reviewer 1 (see previous comments and responses), we have developed a new option to perform water balance on an individual scale. With this alternative option called "Detailed spatial resolution", all the water fluxes (throughfall, stemflow, foliage, bark and soil evaporation, transpiration, water uptake, soil water movements and drainage) are calculated at the individual level. For this option, the model distributes the total soil volume in individual soil volumes (called pedon). The pedon area ($a_{pedon}$) is determined proportionally to the leaf area of the associated tree (but is limited to two times its crown projection):

$$a_{pedon} = \frac{a_{leaf}}{A_{leaf}}.A_{stand}$$

with    $a_{leaf}$, the tree leaf area (m²)

$A_{leaf}$, the total stand leaf area (m²)

$A_{stand}$, the total stand area (m²)

In sparse stands, all the stand area is not allocated to the trees and the remaining area is considered as a pedon without any associated tree. With the detailed water balance option, the model performs a water balance for each tree pedons and also for the remaining pedon (without tree). Contrary to the default option assuming a homogeneous horizontal water redistribution, the alternative option supposes no water redistribution among pedons. These two options allow the user to test two contrasted hypotheses regarding soil water redistribution in the horizontal dimension.

This new option will be presented in the revised manuscript and evaluated against individual transpiration measurements.

Actually, this is the reasoning to apply stand-scale and cohort models which are therefore more consistent. I am even aware of an approach that recalculates individual growth from cohort-based biomass gain (Poschenrieder et al. 2013). The basic criteria, however, may be if your half-individual approach is actually performing better than a cohort-based approach. I am therefore excited to see your analysis in this behalf.

R:

As promised in our response to the comments of reviewer 1, we have realised a first sensitivity analysis of our model to evaluate how changing tree spatial distribution and restricting our individual approach into a cohort approach affect our results.

To do so, we created different stands from an existing stand. The stand we used was Baileux-mixed that is constituted from oak and beech trees in similar proportions and that was used for the evaluation of the model (cfr. manuscript). For testing the effect of the tree spatial distribution, we created two stands composed of exactly the same trees but with a contrasted spatial distribution: a patch-wise mixture in which the trees of a similar size and of the same species have a higher probability to be grouped and an intimate mixture where trees of different tree species and size are intimately mixed. In addition, to test the effect of restricting our approach to a cohort model, we distributed the trees of the intimate mixture in seven different cohorts (four beech and three oak cohorts) and replaced the dimensions of the trees by those of the corresponding cohort. The cohorts were defined according to the girth-class distribution (0-60 cm, 61-105 cm, 106-140 cm and > 140 cm) and the tree dimensions in each cohort were defined based on those of the average tree of the cohort.

In the end, we obtained one stand where all the trees are represented with their own characteristics grouped in patches according to their size and species called patch-wise mixture (tree level approach), another where all the trees are represented according to their real dimensions but where trees of

different size and species are mixed in an intimate way called intimate tree by tree mixture (tree level approach) and a last one with an intimate tree by tree mixture but where the trees of the same cohort have the same averaged dimensions. They are all represented in Fig. 1.

The evolution of the three different stands was simulated between 2001 and 2011 and they were compared based on the LAI and on the annual transpiration (Figs. 2 and 3). The differences in LAI and in annual transpiration among stands were analysed for each cohorts and for all trees together. In addition, mixed linear models were fitted to highlight the effects of the tree spatial distribution and of the clustering in cohorts taking the tree and the year into account as random factors. The cohort clustering effect was tested by comparing the intimate mixtures at the tree and cohort level while the tree spatial distribution effect was assessed by comparing the patch-wise and intimate mixture.

The tree spatial distribution effect (patch-wise vs intimate mixture) generated differences in LAI ranging from -35% to +23% depending on the year and on the cohort considered (Fig. 2). According to the linear mixed models, this effect was always significant, except for the beech > 140 cm cohort (Table 1). Regarding annual transpiration, the relative differences between the patch-wise and the intimate mixture range between -25% to +9% and the corresponding effect was significant, except for the beech 106-140 cm cohort (Table 2).

Compared to the individual approach, the clustering in cohorts induced differences from -9% to +30% for the LAI (Fig. 2) and from -10% to +6% for the annual transpiration (Fig. 3). The corresponding effect was significant, except for the oak 61-105 cm cohort concerning LAI and for the oak and beech 106-140 cm cohorts regarding annual transpiration (Tables 1 and 2).

The effects of tree spatial distribution and of clustering in cohorts are less pronounced at the stand than at the cohort level since they vary from one cohort to the other and partly offset each other at the stand level.

These first results indicate that HETEROFOR is sensitive to the tree spatial distribution and to the clustering in cohorts. To fully address this question, the analysis should be repeated on more sites with various stand types, focused on more model outputs and also tested with longer simulations. However, this is beyond the scope of this paper, which is already quite long with 5 tables and 8 figures and will be even longer with the presentation of the new options and the evaluation of tree transpiration predictions against measurements. The objectives of this paper are to describe the phenology and water balance modules of a new individual-based model that allows to account for the stand spatial and structural complexity and to evaluate its performances. We propose not to introduce this first sensitivity analysis in the article to avoid overloading it and to treat this question in another paper.

P.S. Please try to understand my remark about 'runoff is not included'. It refers to the water at the surface when the water capacity of the soil is reached. Is it nevertheless forced to percolate into or through the soil or does it pile up at the surface?

R:

The water is forced to percolate into the soil. We will insist more on that in the revised manuscript.

[Figure]

Patch-wise mixture (tree level approach)

Intimate tree by tree mixture (cohort level approach)

Intimate tree by tree mixture (tree level approach)

Fig. 1: Visual representation of the three stands created for analysing the model sensitivity to the tree spatial distribution and the restriction of the individual approach into a cohort approach.

[Figure]

Fig.2: Temporal change in LAI (m²/m²) of the mixed stand in Baileux for all trees together and per cohort (four beech and three oak cohorts) according to three different stand configurations: a patch-wise mixture with a tree level approach and two intimate mixtures with a tree or a cohort level approach.

[Figure]

Fig. 3: Temporal change in the annual transpiration (mm) of the mixed stand in Baileux for all trees together and per cohort (four beech and three oak cohorts) according to three different stand configurations: a patch-wise mixture with a tree level approach and two intimate mixtures with a tree or a cohort level approach.

Table 1: Mixed linear model results for the different tree cohorts and for all trees to highlight the impact of tree spatial distribution (patch-wise vs intimate tree by tree mixture) and of cohort clustering (tree vs cohort approach) on LAI.

| Effect | Cohort | Estimate (std dev) | p-value |
|---|---|---|---|
| | Beech (circ. 0-60 cm) | -0.00032 (8.9 $E^{-6}$) | <.0001*** |
| | Beech (circ. 61-105 cm) | 5.8 $E^{-5}$ (2.7 $E^{-5}$) | 0.028* |
| | Oak (circ. 61-105 cm) | 0.00016 (2.7 $E^{-5}$) | <.0001*** |
| Intimate *vs* patch | Beech (circ. 106-140 cm) | 0.0018 (3.9 $E^{-5}$) | <.0001*** |
| mixture | Oak (circ. 106-140 cm) | 0.00071 (3.6 $E^{-5}$) | <.0001*** |
| | Beech (circ. 140+ cm) | 7.6 $E^{-5}$ (0.00013) | 0.56 |
| | Oak (circ. 140+ cm) | 0.00019 (8.1 $E^{-5}$) | 0.018* |
| | All trees | 0.00021 (1.3 $E^{-5}$) | <.0001*** |
| | Beech (circ. 0-60 cm) | 0.00027 (1.0 $E^{-5}$) | <.0001*** |
| | Beech (circ. 61-105 cm) | 0.00017 (2.8 $E^{-5}$) | <.0001*** |
| | Oak (circ. 61-105 cm) | -4.9 $E^{-5}$ (3.3 $E^{-5}$) | 0.14 |
| Tree *vs* cohort | Beech (circ. 106-140 cm) | 9.7 $E^{-5}$ (4.0 $E^{-5}$) | 0.016* |
| approach | Oak (circ. 106-140 cm) | -0.00015 (3.4 $E^{-5}$) | <.0001*** |
| | Beech (circ. 140+ cm) | -0.00027 (9.9 $E^{-5}$) | 0.0070** |
| | Oak (circ. 140+ cm) | -0.00033 (8.0 $E^{-5}$) | <.0001*** |
| | All trees | 0.00010 (1.2 $E^{-5}$) | <.0001*** |

Table 2: Mixed linear model results for the different tree cohorts and for all trees to highlight the impact of tree spatial distribution (patch-wise vs intimate tree by tree mixture) and of cohort clustering (tree vs cohort approach) on annual transpiration.

| Effect | Cohort | Estimate (std dev) | p-value |
|---|---|---|---|
| | Beech (circ. 0-60 cm) | -0.012 (0.00043) | <.0001*** |
| | Beech (circ. 61-105 cm) | 0.0028 (0.00094) | 0.0028*** |
| | Oak (circ. 61-105 cm) | 0.0070 (0.00087) | <.0001*** |
| Intimate *vs* patch | Beech (circ. 106-140 cm) | 0.0034 (0.0014) | 0.016 |
| mixture | Oak (circ. 106-140 cm) | 0.0036 (0.00089) | <.0001*** |
| | Beech (circ. 140+ cm) | -0.011 (0.0040) | 0.0049*** |
| | Oak (circ. 140+ cm) | -0.019 (0.0018) | <.0001*** |
| | All trees | -0.0037 (0.00043) | <.0001*** |
| | Beech (circ. 0-60 cm) | 0.0014 (0.00051) | 0.0082** |
| | Beech (circ. 61-105 cm) | -0.019 (0.0014) | <.0001*** |
| | Oak (circ. 61-105 cm) | -0.0025 (0.0011) | 0.026* |
| Tree *vs* cohort | Beech (circ. 106-140 cm) | -0.0036 (0.0022) | 0.11 |
| approach | Oak (circ. 106-140 cm) | 0.0014 (0.00092) | 0.14 |
| | Beech (circ. 140+ cm) | 0.025 (0.0060) | <.0001*** |
| | Oak (circ. 140+ cm) | 0.0042 (0.0018) | 0.019* |
| | All trees | -0.0022 (0.00057) | 0.0001*** |

**Anonymous Referee #2**

1. The authors present two modules (phenology and water budget) of a new forest growth model HETEROFOR, which is a process-based model considering spatial distribution of individual trees of mixed-species forests. This paper is the second one of the series papers (seems will be more than 2). After reading the first paper (Jonard et al., under review process in GMD) and the comprehensive comments by Referee #1 to this one, my additional comments will be given here.

2. The connection between the phenology module and the processes of photosynthesis and the allocation of NPP was not described in this paper and the first paper. The leaved period from budburst to the start point of yellowing is the period of leaf development. In the phenology module, the progress of leaf development is solely controlled by the temperature. However, the photosynthesis process modelled in HETEROFOR (as described in the first paper) has complicated controlling mechanism including other factors like PAR. The allocation of NPP to leaves and fine roots is further controlled by the nutrient status of the plant. Both the photosynthetic rate and the allocation to leaves will determine the leaf development and the further photosynthesis. Please clearly specify the relationship of photosynthesis, allocation, and leaf development.

Author response:

During the vegetation period, photosynthesis and allocation are independent since they are calculated with two different time steps. Hourly gpp is cumulated over the whole vegetation period, then converted to npp and finally allocated to tree compartments only once at the end of the year. In contrast, there is a close link between phenology and photosynthesis because phenology determines the amount of leaves that can do the photosynthesis but there is no feedback (photosynthesis has no direct impact on leaf development). This has been clarified in the revised version of the first paper (Jonard et al., accepted with major revisions).

3. For the soil water simulation, what's the domain of each individual tree? How does HETEROFOR deal with the spatial heterogeneity of soil water budget? Although the partition of rainwater to interception, throughfall, and stemflow for each tree is one dimensional in the module, the spatial distribution of the individual trees will make the soil water input different under each individual tree. As the soil water availability will have control on foliage conductance (equation 54) and thus on photosynthesis, it is necessary for a clearer description of the 3-dimensional soil water budget.

R:

The default version of HETEROFOR does not consider the spatial heterogeneity of soil water budget. Following the discussions initiated on this subject with the reviewer 1 (see previous comments and responses), we decided to implement a new option performing a water budget at the individual level. We described this new approach in response to the second paragraph of comments of the reviewer 1 (see above).

4. P3L15: I don't see any focus on climate change in this paper

R:

Indeed, a first version of the paper included some simulations considering climate change but this is not the case anymore. This will be removed.

5. P3L16: a temperature increase in which year?

R:

 Between 2071 and 2100 with regards to the 1971-2000 period. This will be added in the introduction.

6. P7L10: what's the endpoint of the gradual loss of photosynthesis and transpiration of the yellowing leaves

R:

Among the different phenological parameters, the user must specify the yellowing threshold under which the green leaf proportion is set to 0 stopping therefore photosynthesis and transpiration. This will be added in the revised version of the manuscript.

7. P8L1: the average budburst date, average of what?

R:

In page 8, we explain how we integrated the within-population variability in the budburst process. As the module was calibrated based on observations carried out on trees representative of the stand, the predicted budburst starting date is expected to be that of an average tree and is consequently called average budburst date. Since, at this date, the leaf expansion of some trees has already started, the model shifts it to obtain the budburst starting date of the earliest trees.

8. P8L16: equals -> reaches

R:

Agreed, it will be changed.

9. P11L4: epsilon in equation 12 is not defined

R:

Epsilon represents the model residuals. This will be stated in the revised version.

10. P11L10: how can the threshold for stemflow appearance is tree size-independent?

R:

In André et al. (2008), from which the equations come from, it is written that "Finally, compensation between increasing stemflow rate and increasing storage capacity as C130 increases explains the non-significant influence of tree size on rainfall thresholds for stemflow."

11. P1123: why equation 16 does not use equation 12 as the stemflow rate?

R:

Equation 16 is derived from Equation 12 and calculates stemflow proportion (stemflow rate divided by the rainfall amount) in order to partition rainfall in throughfall, stemflow and interception. The equation 12 cannot be used directly since it predicts the volume of stemflow (l) during a rainfall event and not the stemflow proportion.

12. P27L9: for the calculation of annual drainage by the chloride mass balance, a monthly calculation won't produce more reliable result?

The chloride mass balance method relies on the assumption that the chloride concentration in rainfall and in the soil are in a steady-state balance. It means that chloride inputs must be equal to outputs and that there is no chloride storage change during the considered timescale. Monthly periods are too short to ensure that the input-output equilibrium assumption is respected. There is indeed a clear seasonal pattern with recharge and discharge periods. The annual timescale seems to be the good trade-off between a small temporal resolution and a perfect verification of the steady-state assumption even though some studies recommend even longer time periods (Alcala and Custodio, 2008; Alcala and Custodio, 2015; Naranjo et al., 2015).